# Diverse values of nature for sustainability

Twenty-five years since foundational publications on valuing ecosystem services for human well-being[1,2], addressing the global biodiversity crisis[3] still implies confronting barriers to incorporating nature's diverse values into decision-making. These barriers include powerful interests supported by current norms and legal rules such as property rights, which determine whose values and which values of nature are acted on. A better understanding of how and why nature is (under)valued is more urgent than ever[4]. Notwithstanding agreements to incorporate nature's values into actions, including the Kunming-Montreal Global Biodiversity Framework (GBF)[5] and the UN Sustainable Development Goals[6], predominant environmental and development policies still prioritize a subset of values, particularly those linked to markets, and ignore other ways people relate to and benefit from nature[7]. Arguably, a 'values crisis' underpins the intertwined crises of biodiversity loss and climate change[8], pandemic emergence[9] and socio-environmental injustices[10]. On the basis of more than 50,000 scientific publications, policy documents and Indigenous and local knowledge sources, the Intergovernmental Platform on Biodiversity and Ecosystem Services (IPBES) assessed knowledge on nature's diverse values and valuation methods to gain insights into their role in policymaking and fuller integration into decisions[7,11]. Applying this evidence, combinations of values-centred approaches are proposed to improve valuation and address barriers to uptake, ultimately leveraging transformative changes towards more just (that is, fair treatment of people and nature, including inter- and intragenerational equity) and sustainable futures.

---

Over millennia and around the world, people have developed myriad ways of understanding and relating to nature and its many values[7]. Although acknowledged in some policy realms, a lot of work remains to consider this diversity in practice (for example, GBF Target 14 regarding "full integration of biodiversity and its multiple values into policies, regulations, planning and development processes … across all levels of government and … sectors"[5]). We assessed diverse evidence sources to synthesize how nature's values are expressed by people and to clarify how nature's values are considered in decisions, including what and whose values are involved or affected. We also introduce a typology that comprises four interrelated meanings of value or its 'layers'[12] (Fig. 1).

The typology's first layer, 'worldviews', encompass the ways people conceive and interact with the world, expressed through 'knowledge systems' (bodies of knowledge, practices and beliefs associated with culture and language)[13]. In the literature, worldviews are frequently classified as anthropocentric (prioritizing human interests) or biocentric and ecocentric (emphasizing living beings or nature's processes as a whole). Pluricentric is also used in the typology to encompass those worldviews with no single 'centre' (focusing on several intertwined relationships among humans, other-than-human beings, nature's components and systemic processes). Next, the second layer, 'broad values' entail the moral principles and life goals held and expressed by individuals, groups and through the institutions (norms and rules) that guide people's interactions with nature and with each other. Certain broad values such as justice, stewardship, unity and responsibility are frequently found to align with sustainability[14]. 'Specific values', the typology's third layer, refer to how judgements regarding the importance of nature and its contributions to people are justified in 'specific' contexts. It is well established that nature's specific values can be instrumental (nature as a means to a desired human end)[15] or intrinsic (value of nature, considered and expressed by people, as an end in itself)[16]. For example, whereas many philosophers interpret intrinsic value in ways that do not relate to the valuer's well-being, economists tend to view intrinsic values as partly connected to a person's well-being, but separate from their own use (a non-use value). The relational category of specific values captures how people express the importance of meaningful relationships between people and nature and among people through nature such as reciprocity and care[17]. Finally, the fourth layer, 'value indicators' are quantitative measures and qualitative descriptors used to denote nature and people–nature relationships and nature's contributions to people (NCP)[18], typically in biophysical, monetary or socio-cultural terms[12].

Cutting across these layers, life frames depict how individuals, institutions or policies might prioritize subsets of values depending on how people–nature relationships are framed[12,19]. For example, 'living from' nature emphasizes instrumental values such as nature's capacity to provide resources for sustaining livelihoods. 'Living in' nature focuses on how people recognize nature's importance as settings for their lives, practices and cultures, particularly supporting relational values. 'Living with' nature centres on nature's life-supporting processes and connections to other-than-human beings, thereby prioritizing both intrinsic and relational values. 'Living as' nature prioritizes embodying and perceiving nature as a physical, mental and spiritual part of oneself, emphasizing broad values of oneness, kinship and interdependence. Different life frames are expressed in varying combinations across time and contexts, but research and policy most frequently align with 'living from' nature.

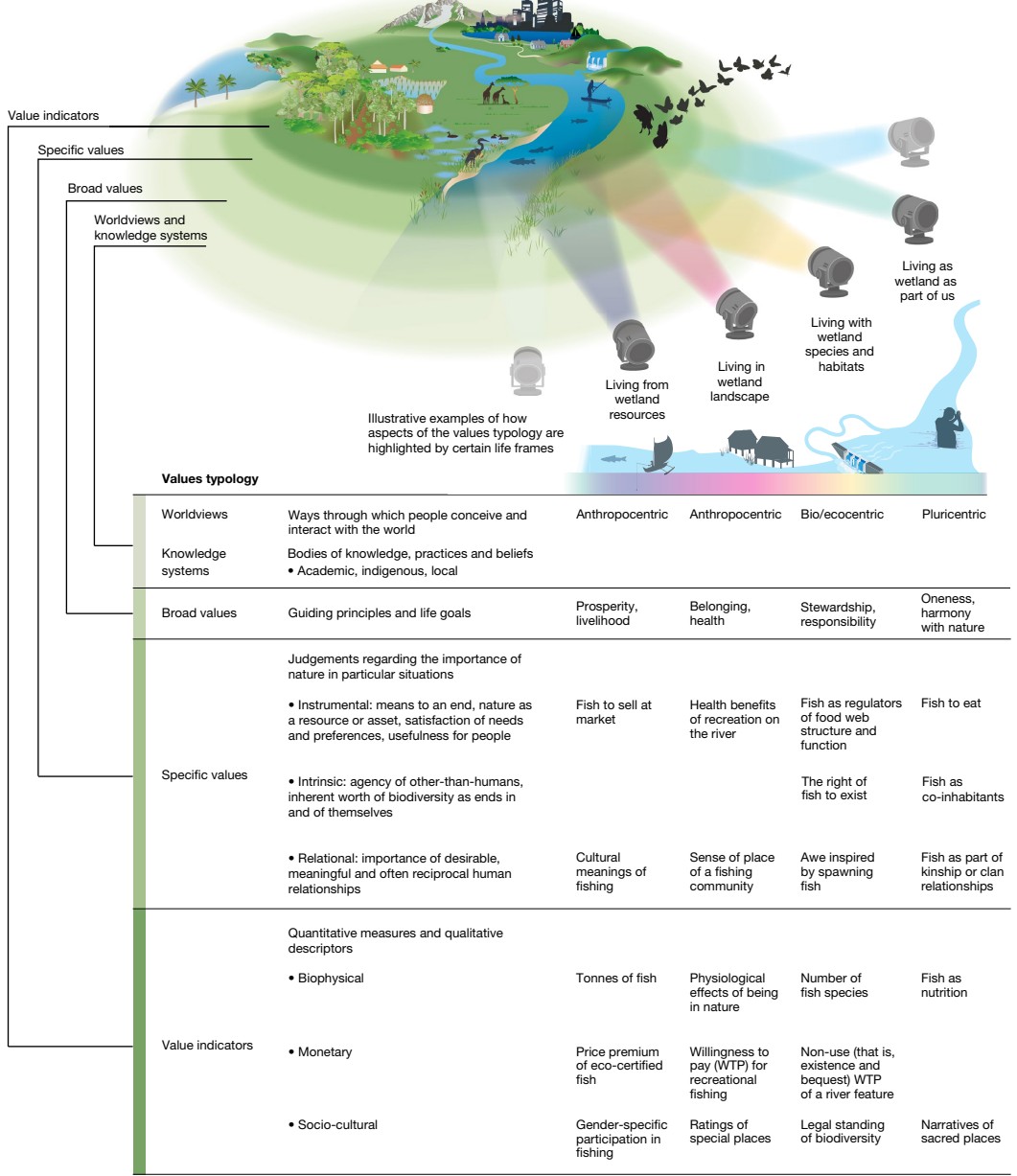

| Values typology | | Living from wetland resources | Living in wetland landscape | Living with wetland species and habitats | Living as wetland as part of us |
|---|---|---|---|---|---|
| Worldviews | Ways through which people conceive and interact with the world | Anthropocentric | Anthropocentric | Bio/ecocentric | Pluricentric |
| Knowledge systems | Bodies of knowledge, practices and beliefs • Academic, indigenous, local | | | | |
| Broad values | Guiding principles and life goals | Prosperity, livelihood | Belonging, health | Stewardship, responsibility | Oneness, harmony with nature |
| Specific values | Judgements regarding the importance of nature in particular situations | | | | |
| | • Instrumental: means to an end, nature as a resource or asset, satisfaction of needs and preferences, usefulness for people | Fish to sell at market | Health benefits of recreation on the river | Fish as regulators of food web structure and function | Fish to eat |
| | • Intrinsic: agency of other-than-humans, inherent worth of biodiversity as ends in and of themselves | | | The right of fish to exist | Fish as co-inhabitants |
| | • Relational: importance of desirable, meaningful and often reciprocal human relationships | Cultural meanings of fishing | Sense of place of a fishing community | Awe inspired by spawning fish | Fish as part of kinship or clan relationships |
| Value indicators | Quantitative measures and qualitative descriptors | | | | |
| | • Biophysical | Tonnes of fish | Physiological effects of being in nature | Number of fish species | Fish as nutrition |
| | • Monetary | Price premium of eco-certified fish | Willingness to pay (WTP) for recreational fishing | Non-use (that is, existence and bequest) WTP of a river feature | |
| | • Socio-cultural | Gender-specific participation in fishing | Ratings of special places | Legal standing of biodiversity | Narratives of sacred places |

Value indicators
Specific values
Broad values
Worldviews and knowledge systems

Illustrative examples of how aspects of the values typology are highlighted by certain life frames

**Fig. 1 | An inclusive typology of the many values of nature.** To clarify and identify different values and their interrelationships, the typology distinguishes four flexible and interconnected layers of what value means: worldviews and knowledge systems, broad values, specific values and value indicators. Life frames (metaphorically shown as light beams) illustrate how some sets of values might be given prominence in the different ways people relate to nature (here a watershed feeding into an estuarine wetland)[7,12].

This plurality of values is found around the world. For example, in postindustrial societies with high levels of material security[20], we see increasing endorsement of relational and intrinsic values such as seeing wildlife as part of one's social community and deserving of rights[21]. In the Global South, where lower levels of livelihood security may favour instrumental values[20], relational value expressions are also prevalent, such as spirituality and cultural identity[22,23]. Similarly, Indigenous and local knowledge are embodied in different philosophies of good living around the world underpinned by relational values as the basis for collective people–nature well-being[12], including through concepts such as *Buen vivir* in South America[24], *Ubuntu* in sub-Saharan Africa[25] and *Satoyama* in Japan[26], among others.

The typology of values facilitates more comprehensive identification of nature's values in complex decision-making contexts. We demonstrate this use to analyse the restoration of India's Chilika Lagoon

(Supplementary Information, Section C). Designated a Ramsar site in 1981, Chilika was listed on the convention's Montreux Record in 1993, due to ecological degradation from human actions (such as aquaculture, development infrastructure) that also harmed numerous stakeholders, including conservationists, traditional fishers, aquaculturists and farmers. In 2001, Chilika became the only Asian wetland to be 'delisted' as an area of concern after a successful restoration effort shifted decision-making from a narrow focus on extractive activities, for instance, those linked to aquaculture's instrumental values, or strict conservation, including those linked to biodiversity's intrinsic values, to a plural-values perspective that balances these with people's important connections to the wetland such as relational values connected to the cultural identify of being fishers. Integrating such specific value types into decisions geared towards wetland restoration required monitoring biophysical indicators (water flow, fish diversity),

economic and financial aspects through monetary measures (income growth and distribution), and socio-cultural descriptors (veneration of religious sites, relationships between fishers and dolphins). A specific decision-making authority was created for the wetland that accounted for diverse stakeholders' worldviews (in this case, conservationists' bio- or ecocentric worldview, as well as fishers' anthropocentric and pluricentric perspectives) to better align actions with broad values underpinning key human–nature relations, exemplified here in the symbiotic relationships between fishers and Irrawaddy dolphins. Identifying these many-value layers and types within the values typology can facilitate their integration into decisions, ultimately enhancing legitimacy, such as through recognition justice, and reconciling value clashes, as can arise between fishers, aquaculturists and environmental managers.

This case also illustrates the potential challenges and approaches to accounting for the many values held by different stakeholders in decisions. For example, the wetland's specific values are directly 'comparable' when they can be accounted for with the same indicators (for example, monetary metrics for cost–benefit analyses of tourism development projects, in which investment costs are compared with the market and non-market economic effects on people and the wetland). Other specific values can be 'compatible' when they share common features such as being measured in spatial units, allowing them to be addressed together, despite using different indicators (for example, spatially overlaying bundles of the wetland's contributions to people proxied with biophysical, monetary and socio-cultural indicators). However, some values are simply 'incommensurable', being neither comparable nor compatible with others[27]. For instance, a wetland could be solely managed for fisheries and commercial tourism to enhance some stakeholder's instrumental values, such as economic benefits or nutritional yield, but erode other intrinsic values, such as keystone species' habitat, and relational values, including fishers' cultural identity. Without a common or compatible denominator, decisions cannot rely on standard trade-off analyses (for example, cost–benefit analysis, multicriteria decision aid); rather, they should treat these values in 'parallel'.

The challenge of incommensurability does not indicate that any particular value by definition should have higher priority than others in decision-making. It does mean that procedures need to recognize values that would otherwise be excluded or lost and ensure reasoned prioritization. Societies create principles to handle such difficult evaluations. For example, several countries have institutionalized the precautionary principle for situations when the consequences of decisions are largely unknown. Similarly, dealing with incommensurability demands further development of agreed principles, which may in turn require deliberative approaches[12] to help decision makers choose among alternative policy options. This can complement evidence based on 'social' preferences (that is, aggregated individual preferences) although it is well understood in social-choice research that aggregation of individual preferences to guide fair societal choices faces dilemmas[28]. The challenges compound when different value types need to be weighed against each other and aggregated. Yet, as participatory processes can be manipulated, power asymmetries should be addressed.

## Valuation methods are diverse and spreading, but uptake remains limited

Valuation generates information that can be used to make nature's values more visible to decision makers, such as a scarcity indicator to protect natural assets that are at the risk of being over-exploited ('living from' nature). Valuation can also be used to reveal the need to protect the ecological systems on which humanity depends ('living with' nature), and to recognize other ways humanity relates to nature ('living in' and 'living as' nature). At present, more than 50 well-established valuation methods from disciplines including anthropology, political science,

economics and conservation biology are available to elicit the diverse values people hold for nature[29]. Several method classifications exist, notably those from environmental economics[30], on which we expand to assess valuation methods grouped into four cross-disciplinary 'method families', based on the source of information about values: (1) nature-based valuation gathers information about the importance of nature and NCP through direct and indirect observation of nature (for example, spatial ecosystem services mapping)[31], (2) statement-based valuation obtains information from people's expressions of their values (such as stated preference surveys[32], deliberative processes[33]), (3) behaviour-based valuation identifies how people value nature by observing what they do in relation to nature (such as hedonic pricing[34], livelihood dependence[35]) and (4) integrated valuation brings together different types of value assessed with diverse information sources (for example, multicriteria decision aid)[36] and also seeks to understand how values, behaviour and environmental outcomes interact in dynamic ways (for example, integrated modelling)[37]. Whereas the valuation field has advanced substantially regarding the first three method families, it has not yet reached maturity regarding its integration potential to understand the dynamic interactions and feedbacks between peoples' values, behaviours and the impact on biophysical indicators[38].

This classification system highlights the diversity of valuation methods across different disciplines and traditions, including those from Indigenous peoples and local communities (IPLCs). Although academic understanding of IPLC valuation methods is limited, existing documentation underscores the important role of worldviews and broad values, including social norms and cultural and spiritual beliefs, in these valuation procedures[29]. IPLC valuation practice can take a different form from the aforementioned categories, often being based on a communal process informed by Indigenous and local knowledge and long-held traditions, and guided by principles such as belonging, stewardship, responsibility and oneness with nature. Bringing together valuation methods associated with diverse worldviews is desirable, but requires giving equal footing to IPLC valuation, and accepting the limitations of knowledge integration[39].

The availability of information influences the valuation method to be used. Likewise, the choice of a valuation method itself influences the information made available for decision-making, such as whether the focus is on instrumental, relational or intrinsic values, what evidence it is based on and the role of those participating in the valuation[40]. Furthermore, the application of any valuation method to real-world decision-making is conditioned by trade-offs between relevance (salience to the decision's context), robustness (reliability and representation[41,42]) and resources (the time, financial, technical and human resources required to design and apply valuation). Therefore, method choice reflects different trade-offs among these '3Rs'; increasing the relevance and robustness of valuation typically requires more resources and determines to a large extent the feasibility of applying any given method[29].

The number of peer-reviewed valuation studies around the world has increased on average by about 10% annually over the last three decades[29]. Nature-based valuation methods are the most commonly applied, followed by statement-based, behaviour-based and integrated valuation methods. Instrumental values are elicited more often than intrinsic and relational values. The ultimate goal of valuation varies, but three main purposes are observed: improving the state of nature, for instance, by assessing changes in ecosystem structure[43]; enhancing people's quality of life, including assessing changes in the provision of NCP[44]; or generating more socially just outcomes by considering and assessing the various dimensions of justice as a broad value. Three widely recognized dimensions of justice include: (1) recognition justice, acknowledging and respecting different worldviews, knowledge systems and values, (2) procedural justice making decisions that are legitimate and inclusive for those holding different values, and (3) distributive justice, ensuring the fair distribution of NCP[45]. Valuation

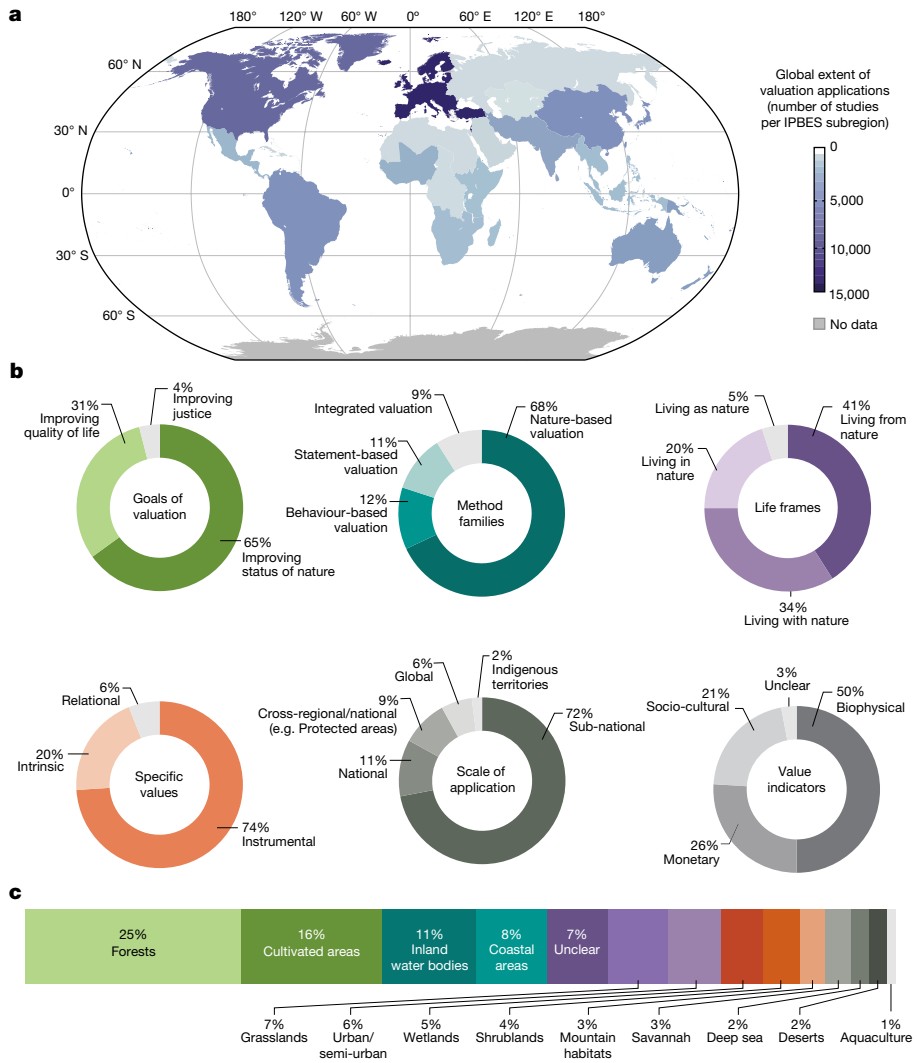

**Fig. 2 | Distribution and characterization of nature valuation studies.**
From 48,781 peer-reviewed studies reported with explicit geo-referenced information (depicted in the map), a stratified random sample of those published between 2010 and 2020 provided 1,163 studies applying specific valuation methods. **a**, Global distribution of valuation studies. **b**, Characterization of nature valuation studies reported. **c**, Habitats in which valuation was applied. These were reviewed in depth and used to develop the figure's statistical graphics[7,29].

to guide decisions that improve the state of nature is prevalent in the literature, whereas a focus on distributive justice is rare (Fig. 2).

Valuations have been mostly undertaken in Europe, the Americas and Asia and the Pacific, and to a lesser extent in Africa and Central Asia. Studies have predominantly targeted forests, cultivated areas and inland water bodies (Fig. 2). Only 10% of valuation studies have been conducted in marine environments (coastal and deep sea) even though oceans cover more than 70% of the planet's surface. Valuations have mainly been performed at subnational scales, rather than national and global scales, which reflects the scale of most decision-making. Very few studies deal with cross-regional or cross-national decision-making or explicitly reference IPLC rights and territories (Fig. 2). The global pattern suggests that the two main factors that increase the amount of valuation studies in a region are the level of threat to biodiversity and environmental quality, and availability of human and financial resources to conduct valuation[29].

The evidence also indicates that most (62%) valuation studies, especially nature-based valuations, do not involve stakeholder participation in the valuation[22]. Furthermore, despite calls for increased use of valuation in policymaking, less than 5% of published peer-reviewed valuation studies document uptake of values information into decisions,

a figure that has not increased over the past three decades[46]. Policy documents from many countries also show limited use of the suite of available valuation methods; only a few methods with certain value perspectives dominate valuation practice. For instance, when countries monitor the values of biodiversity (Aichi Target 2 of the Convention on Biological Diversity), their National Biodiversity Strategies and Action Plans generally use biophysical and to a lesser extent monetary indicators[12,46]. Despite the general perception that policymaking favours economic approaches to the valuation of nature[47], during the last three decades, peer-reviewed economic valuation studies have not documented uptake more often than other (non-economic) approaches[46].

Key barriers to valuation uptake in decision-making are partly due to a perceived lack of robustness and reliability of methods and also insufficient financial and technical resources to commission valuation and integrate it into decision-making processes and management actions[22]. Other barriers include a lack of relevance of valuation results to political jurisdictions, administrative levels, sectoral interests or different stakeholders, and the lag between delivery of study results relative to decision time frames[46]. Even when valuation is commissioned and well communicated in environmental impact assessments, it may be ignored or used to justify decisions to mitigate rather than avoid

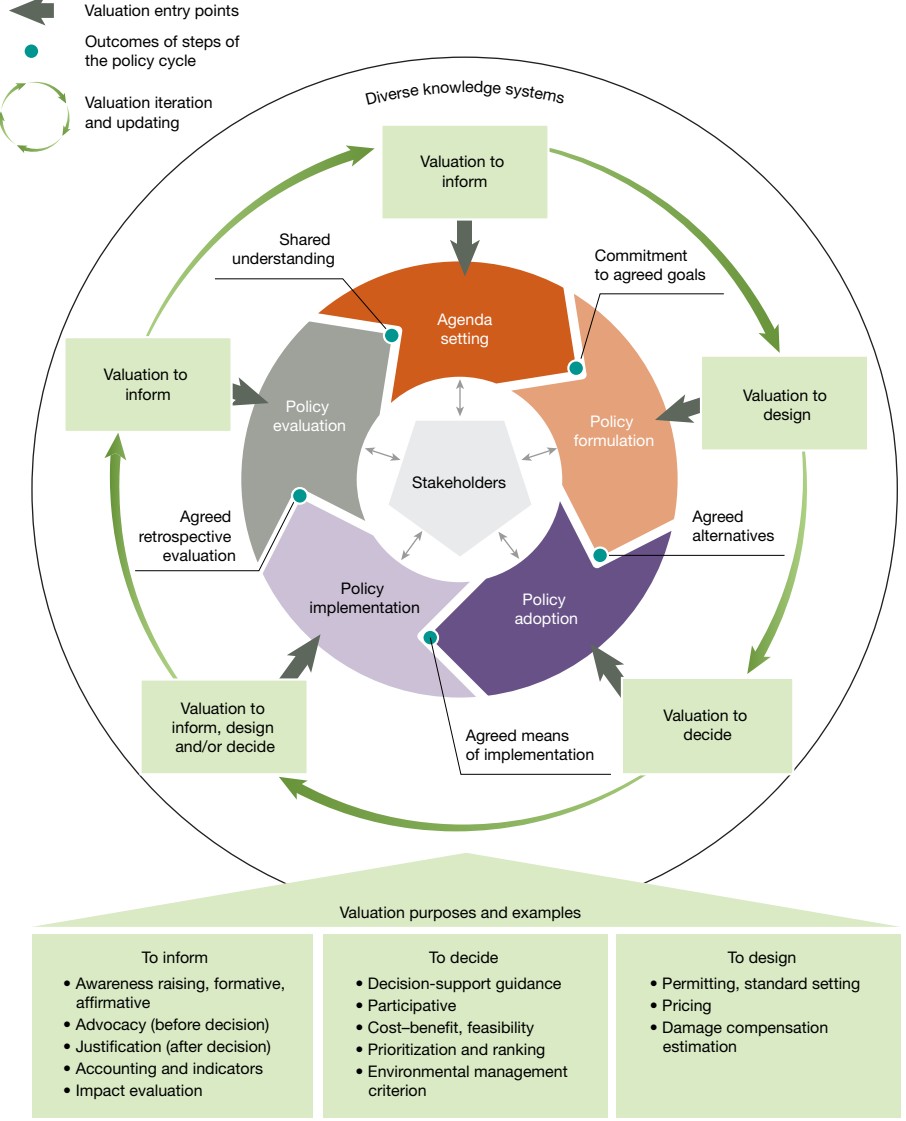

**Fig. 3 | The policy cycle provides different points for entering valuation into decisions.** Valuation activities can support informative, decision-making and policy design purposes by providing different types of value information to policymakers and stakeholders throughout the cycle[7,46].

negative effects[48]. Additionally, powerful stakeholders can hinder the representation of diverse values in decisions by, for example, blocking people's direct representation of their own voices or selectively using valuation methods to only partially represent their values[7]. Whereas guidance exists for improving participation and representation[49], only 12% of assessed valuation studies explicitly consider design choices to improve stakeholder inclusion (for example, efforts to avoid excluding or marginalizing certain values)[22] (Supplementary Information Section B, no. 10).

To increase the likelihood of uptake across the range of valuation approaches and contexts, valuation can be adapted and timed to suit policymaking needs regarding particular purposes and decision-support opportunities[46]. In particular, valuation can be tailored to support different stages of policymaking (often understood as a cycle in which learning from one policy feeds back into the design of the next; Fig. 3). These include: (1) aiding agenda setting and support commitment to agreed goals; (2) providing technical assistance for policy formulation by, for example, agreeing on the alternatives under consideration, or the design of economic incentives, such as payments for ecosystem services (PES); (3) supporting decisions

for policy adoption and assessing cost-effectiveness of alternatives for policy action; (4) facilitating adjustments to implementation measures or budget allocations; and (5) helping undertake retrospective policy evaluation (Fig. 3 and Supplementary Information Section C)[46].

## Engaging diverse values improves decision outcomes

A review of impact evaluation studies on protected areas and an in-depth qualitative examination of case studies from around the world show that when local values such as stewardship are integrated, decision-making delivers more just and sustainable outcomes, especially when these values have been traditionally marginalized[12,46]. Studies have established that community involvement improves management effectiveness (based on an analysis of more than 8,000 assessments from more than 3,000 global protected areas[50]), and that local empowerment and recognition of local values, especially for Indigenous communities, enhances win–wins between ecological and social outcomes of protected areas (demonstrated in a meta-analysis of 171 peer-reviewed studies[51] and a systematic review of 169 publications[52],

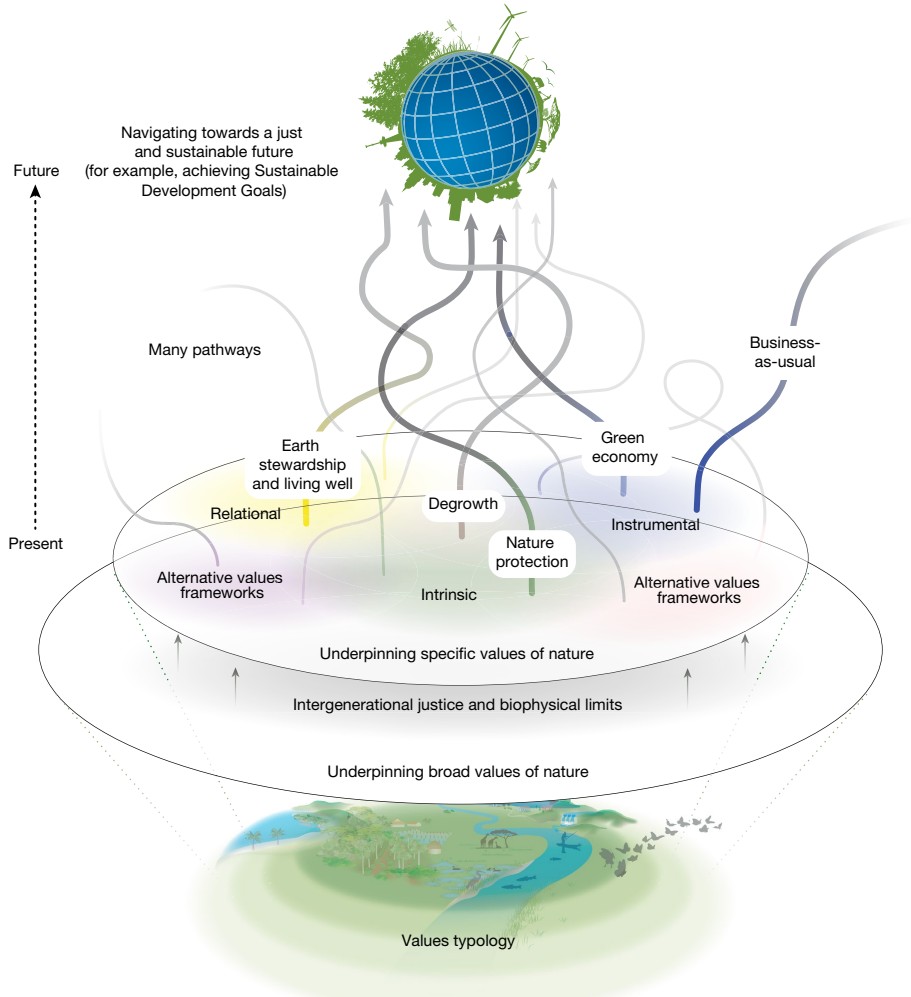

**Fig. 4 | The diverse values of nature underpin different pathways towards sustainability.** Different sustainability pathways, such as 'green economy', (socially and ecologically sustainable) 'degrowth', 'Earth stewardship', 'nature protection' and alternatives arising from diverse worldviews and knowledge systems, including 'Living well' and other philosophies of good living (associated with IPLCs) have varying prioritizations of specific values (instrumental, relational and intrinsic), but share certain broad values, including the imperative to do justice to future generations and respect biophysical limits[7].

as well as in-depth case studies of our review (Supplementary Information Section B, no. 16). Studies also demonstrate that acknowledging nature's many values can help PES programs avoid eroding inherent motivations for conservation or enhance existing pro-environmental behaviours, improving outcomes for people and nature[53]. Similarly, in agroecosystems, being able to recognize a comprehensive suite of small landholder values (including women's) is key to codesigning initiatives that effectively conserve on-farm agro-biodiversity, ensure food security and sovereignty, and maintain place-based livelihoods[46,54].

Further evidence from contextual analysis of case studies (Supplementary Information Section B, nos. 16, 17 and 18) in protected areas, PES programs and major infrastructure projects, such as mining and dams, suggests that power asymmetries among actors involved in or affected by policy decisions can disrupt the representation of the diverse values at stake, especially at the local level[46]. Developers that articulate instrumental values for large development and infrastructure projects often have more discursive and structural power than those local stakeholders negatively affected by these initiatives. For instance, dams are often proposed to enhance market-based instrumental values (such as electricity to urban consumers, irrigation water for agriculture, jobs), whereas the instrumental values of those damaged by the project (such as loss of other farming and fishing livelihoods) are excluded, as are many relational values (such as cultural identity, place attachment)[46,55]. Similarly, the diverse ways that IPLCs conceive and relate to nature's many values are often underrepresented or enter too late into the decision process, such as in the design of protected areas and PES programs[56,57].

Studies incorporating detailed case analysis across all continents have revealed that power asymmetries resulting in imbalanced representation of values in decisions have important consequences for people and nature. For instance, disregarding smallholders' diverse values in the design of sustainability certification programs can lead programs to not account for the barriers these stakeholders face, compromising the intended positive social and environmental benefits[58] (Supplementary Information Section B, no. 15). Likewise, ignoring or marginalizing locally held values in the design and implementation of conservation programs can leave a legacy of mistrust and create conflicts with local communities[59,60], jeopardizing program outcomes over time (Supplementary Information Section B, nos. 16, 17). These failures can be avoided or better addressed when policies align with a more comprehensive suite of local values. This further implies

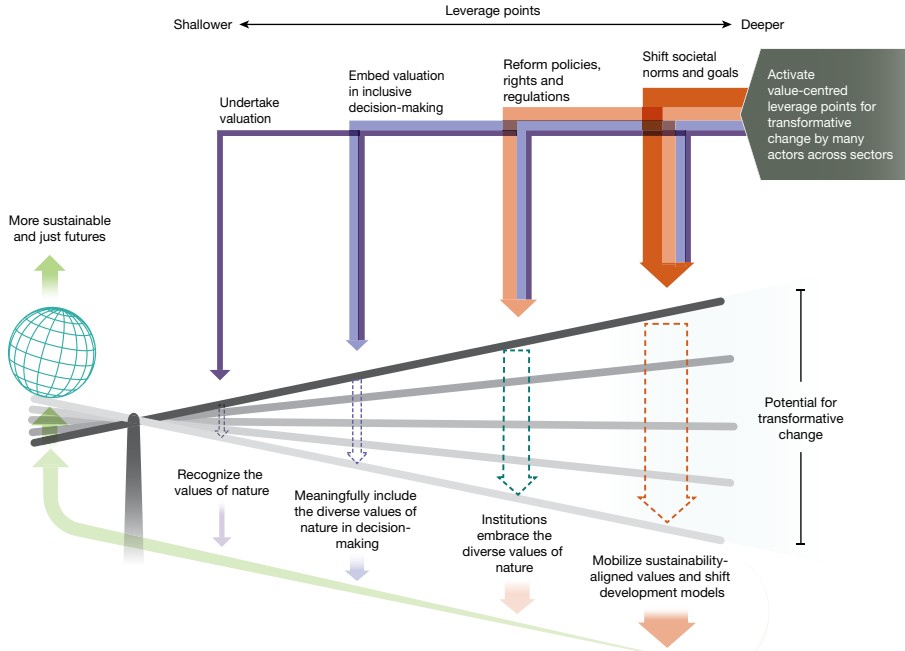

**Fig. 5 | Values-centred leverage points can catalyse transformative change towards more just and sustainable futures.** Transformative change is more likely when interventions engage several values-centred leverage points. The leverage points are interdependent, whereby jointly activating them entails addressing feedbacks among them, adding them up (moving left to right across the lever) or cascading down (moving right to left across the lever)[7].

that power asymmetries must be addressed head-on in the design and development of programs by recognizing the diversity (and potential incommensurability) of values held across all actors and ensuring participatory parity. Doing so can ultimately lead to more equitable distribution of projects' costs and benefits in both conservation and development contexts[46].

## The diversity of values underpins many pathways towards sustainability

Navigating towards more just and sustainable futures entails imagining what this transformation might look like, a process that needs creative and analytical input. Scenario planning integrates thinking across several disciplines and can reveal important insights about values integration into policymaking. Only half of the 460 scenarios reviewed (Supplementary Information Section B, no. 23) mention or explicitly incorporate values. Where scenarios were codeveloped with stakeholders, most (94%) of these were underpinned by instrumental values[14]. The scenarios considered most likely to achieve more just and sustainable futures, however, typically consider instrumental, relational and intrinsic values, and especially emphasize these values regarding NCP[14], although generally scenarios are not codeveloped by taking into account value trade-offs.

Beyond the unsustainable 'business-as-usual' pathway (typically used as a basis for comparison), we identify at least four well-established sustainability pathways[7,14]. The 'green economy' pathway stresses the importance of reforming economic institutions, technologies and performance metrics, while prioritizing instrumental values[4,61]. 'Nature protection' centres around protecting biodiversity for its own sake and expanding protected area networks[62], highlighting intrinsic values (while also acknowledging instrumental values). 'Earth stewardship' emphasizes local sovereignty, solidarity and the promotion of biocultural practices[63], underscoring relational values (and also acknowledging intrinsic values). Socially and ecologically sustainable 'degrowth' focuses on reducing overconsumption and overproduction,

and redistributing wealth[64,65], engaging instrumental, intrinsic and relational values. All these sustainability pathways are founded on broad values associated with intergenerational justice and respect for biophysical limits[14] (Fig. 4).

## Transformative change involves leveraging nature's values

Achieving more just and sustainable futures calls for reforming societal structures to address asymmetric power relations underpinning the allocation of property rights, including legal decisions about who holds rights to degrade or be protected from environmental harm and who or what is a subject of rights (for example, a river, Mother Earth and so on). These reforms need to be complemented by the use of policy instruments to internalize negative environmental externalities that arise from the rift between private and public values, reducing overconsumption and overproduction, and by applying indicators of progress that include social and ecological sustainability criteria[4]. Achieving these actions also implies confronting the contradictions evidenced by the historical and current prioritization of a narrow suite of nature's values. For example, governments and private enterprises frequently make decisions grounded in market-based instrumental values. Similarly, conservation policies have frequently prioritized nature's intrinsic values, despite increasing advocacy regarding the instrumental and relational values held by those living within and around protected areas who rely on biodiversity for their livelihoods[12,66]. As a consequence, a system-wide strategy is needed across technological, economic and social domains, including profound changes that address the worldviews and broad values that underlie the direct and indirect drivers of biodiversity loss[8,67]. We identify four values-centred leverage points, ranging from short-term, easier to achieve actions to longer-term, harder to achieve efforts that, in combination, can catalyse system-wide transformative changes (Fig. 5).

The first leverage point involves improved valuation by identifying more diverse values of nature[12] and ensuring there are methods

and procedures to describe, record and report them[22,46]. Such recognition and accounting is still not widely done, but is an essential step for harnessing knowledge(s) and motivations to protect nature, including mobilizing a more inclusive set of specific values of nature and sustainability-aligned broad values[14]. Yet, although enhancing recognition of nature's values and undertaking valuation are necessary, these efforts alone are insufficient to ensure pro-environmental decisions and behaviour[14].

Therefore, the second leverage point involves enabling value information generated through valuation approaches to be embedded into decision-making[46,68]. Actions here may include using existing legal and economic policy measures (for example, green taxes) to make production and consumption decisions more sustainable or establishing guidelines for planning decisions that require consideration for the many values of nature. Whereas many theories explain causal relationships between values and behaviour, broader contexts partially determine people's capacity and ability to act on their values[69]. Hence, interventions should be tailored accordingly, as illustrated by choice of transport mode being affected by availability of public transportation infrastructure. Furthermore, integrating values into policy decisions is more likely to occur when valuation is tailored for a specific policy purpose[46,70]. For instance, at a national level, development of standardized, high spatial resolution ecosystem accounts[71] can provide the biophysical indicators to inform policy design. Some countries (for example, Portugal, Germany) already use biophysical indicators of conservation effort to then redistribute tax revenues (known as ecological fiscal transfers) to local and regional administrations as compensation for lost revenue and extra costs due to establishing and managing protected areas[72]. Likewise, using valuation as part of incentives for pro-environmental behaviour in production and consumption practices (including certification, tax rebates, PES and so on) offers opportunities for strengthening people's sustainability-aligned values. In addition, embedding valuation into environmental and social safeguards (including land tenure rights, equitable access and benefits sharing and procedural justice) can promote conservation in IPLC territories[52,57]. To enable the conditions for embedding valuation into decisions, it is particularly important to implement inclusive and legitimate processes that meaningfully represent stakeholders' values[46].

The third leverage point involves reconfiguration of societal structures, especially with regard to the decision-making architecture to normalize and scale-up the incorporation of diverse values in decisions. This requires reforms to core legal, economic and political institutions (for example, property rights, trade rules, parliamentary systems) in ways that change what and whose values gain decision-making power in society[73]. Moderating the impetus towards short-term political decisions tied to electoral cycles (for example, instituting procedural rules that protect the interests of future generations) would also be an important structural reform. Another would be to enhance businesses' capacity to care for nature's values by broadening responsibility beyond shareholder interests (for example, instituting rules that preclude biodiversity loss throughout value chains). Similarly, reforming and complementing macroeconomic indicators (for example, gross domestic product) to include values that encompass social and ecological well-being could change both the design and intent of the economic system[4]. In the context of IPLCs, institutional reforms to secure territorial property rights and recognize the rights of natural entities (for example, rivers) have demonstrated potential to be highly transformative[68,74]. Similarly, embracing rights-based approaches would legitimize many IPLCs' customary rules that already recognize and embed diverse values and valuation in their conservation decisions. All such institutional changes across sectors would alter predominant societal rules to better ensure recognition for diverse worldviews and broad values of nature. In turn, these actions could support broader reforms towards comanagement regimes and foster further institutional changes throughout political and economic systems, helping to overcome current resistance to the worldviews and values held by IPLCs. For example, revisiting the wetland case study (Supplementary Information Section C), the Chilika Development Authority was created to implement socially legitimate wetland restoration by ensuring dialogue that embraces the many values of its diverse stakeholder representatives.

Whereas the first three leverage points act on largely existing values, the fourth leverage point involves modifying underlying social norms and goals to reflect the links between justice and sustainability. Examples of fundamental changes in social norms include how a society views 'progress' or a 'good life' in terms of relationships with nature[14]. These tasks are complex, but inherently transformative. They accompany many institutional reforms contemplated in the previous leverage point (for example, changing macroeconomic indicators of 'progress' beyond gross domestic product) and could powerfully go beyond the goal of some sectors to continue increasing material and energy consumption in already affluent societies. Whereas environmental responsibility norms can be nurtured throughout the lever, strategies for wider socialization can aid larger-scale sustainability outcomes. For instance, empowering civil society's role through new participative fora such as citizen assemblies could be a way to form new shared values or surface latent sustainability-aligned values, fostering a counter-force to dominant ways of conceiving the values of nature and shifting current hegemonic societal norms through more open dialogue[75].

Transformative change is, thus, a multifaceted process involving engagement of the four values-centred leverage points. Fortunately, opportunities for synergies arise, as leverage points are not static categories; instead, there are interdependencies along the lever's action gradient. Leverage points may be activated in a cumulative way (from left to right across the lever), such as when a policy change (for example, introducing a green tax) triggers a change in social norms over time (for example, recycling). Values-centred leverage points can also be triggered in the opposite direction (cascading down the lever). For example, in Europe a deep leverage point involved a shift in vision about the role of agriculture, driven by the wider societal goal of sustainability and epitomized through a political agreement underpinning the European Common Agricultural Policy. In the early 1990s, this involved a change from supporting the agricultural sector to ensure self-sufficiency to recognizing the need for mitigating the negative externalities harming wildlife and people's health (a new social norm and goal). Since that time, a series of reforms and the associated political effort has increased the environmental components of the agricultural policy framework (the third leverage point). First, policy instruments and tools were implemented towards compliance with minimum environmental standards to justify income support to farmers. More recently, the reform has introduced environmentally targeted payments for adopting sustainable agricultural practices (the second leverage point)[76]. The designs of these policy instruments are being aided by the valuation of the externalities for which different methods and decision-support tools are used (for example, shadow pricing, choice experiments and cost–benefit analysis)[77,78]. This example illustrates how shifting societal norms and goals can trigger the activation of all other values-centred leverage points. Clearly, power relations must be confronted, such as between citizens and agri-business, that ultimately influence whose and which values get priority in decisions.

## Conclusion

The transformative changes needed to achieve ambitious biodiversity and development goals require confronting the status quo and associated vested interests tightly tied to current institutions (norms and legal rules), including the allocation of property rights over nature. Such transformation also demands recognizing and integrating the diverse values of nature into political, economic and other day-to-day

decision-making that informs environmental management. Given the current allocation of property rights in predominant political systems, the primacy of market-based instrumental values in many decisions is at the core of underlying direct and indirect drivers of today's global biodiversity crisis[67]. These market-based values need to be balanced with the relational, intrinsic and non-market instrumental values that are also part-and-parcel of the reasons nature matters to people. Achieving more just and sustainable futures entails mobilizing and nurturing broad values and new societal norms and goals to trigger changes in the current institutional fabric of society. Fortunately, by using an inclusive typology of nature's values and the extensive portfolio of existing valuation methods, decision-making can be better informed to reflect nature's many values. Furthermore, value-centred leverage points can be triggered to achieve tangible, multiscale outcomes.

Nature's values are expressed in and shaped by worldviews and knowledge systems, but also by power relations that underpin institutional structures in societies[79]. Thus, enabling solutions to the global biodiversity crisis implies identifying and navigating these issues of values diversity and associated potential conflicts. To scope the limitations and opportunities with respect to better integration of nature's many values, we have contextualized our findings in light of real-world policy-making needs such as implementing the GBF's new targets. Specifically, we propose (1) using an inclusive typology of values, (2) accessing the extensive portfolio of available valuation methods, and (3) engaging a range of leverage points that can be acted on for transformative change. To ignore the diversity of nature's values in science and policy would be to continue to sell nature short to the detriment of all life on Earth.

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

Unai Pascual[1,2,3 ✉], Patricia Balvanera[4], Christopher B. Anderson[5,6], Rebecca Chaplin-Kramer[7,8], Michael Christie[9], David González-Jiménez[4,10], Adrian Martin[11], Christopher M. Raymond[12,13,14], Mette Termansen[15], Arild Vatn[16], Simone Athayde[17], Brigitte Baptiste[18], David N. Barton[19], Sander Jacobs[20,21], Eszter Kelemen[22], Ritesh Kumar[23], Elena Lazos[24], Tuyeni H. Mwampamba[4,25], Barbara Nakangu[26], Patrick O'Farrell[27,28], Suneetha M. Subramanian[30,31,32], Meine van Noordwijk[30,31], SoEun Ahn[33], Sacha Amaruzaman[30], Ariane M. Amin[34,35], Paola Arias-Arévalo[36], Gabriela Arroyo-Robles[4], Mariana Cantú-Fernández[4], Antonio J. Castro[37], Victoria Contreras[4], Alta De Vos[38,39], Nicolas Dendoncker[40], Stefanie Engel[41], Uta Eser[42], Daniel P. Faith[43], Anna Filyushkina[44,45], Houda Ghazi[46], Erik Gómez-Baggethun[16,19], Rachelle K. Gould[47], Louise Guibrunet[48], Haripriya Gundimeda[49], Thomas Hahn[50], Zuzana V. Harmáčková[50,51], Marcello Hernández-Blanco[52], Andra-Ioana Horcea-Milcu[53,54], Mariaelena Huambachano[55], Natalia Lutti Hummel Wicher[56], Cem İskender Aydın[57], Mine Islar[58], Ann-Kathrin Koessler[40,59], Jasper O. Kenter[9,60,61], Marina Kosmus[62], Heera Lee[63,64], Beria Leimona[30], Sharachchandra Lele[65,66,67], Dominic Lenzi[68], Bosco Lliso[1,69], Lelani M. Mannetti[70], Juliana Merçon[71], Ana Sofía Monroy-Sais[72], Nibedita Mukherjee[73], Barbara Muraca[74], Roldan Muradian[75], Ranjini Murali[76,77], Sara H. Nelson[78], Gabriel R. Nemogá-Soto[79,80], Jonas Ngouhouo-Poufoun[81,82], Aidin Niamir[83], Emmanuel Nuesiri[84], Tobias O. Nyumba[61,85], Begüm Özkaynak[86], Ignacio Palomo[87], Ram Pandit[88,89], Agnieszka Pawłowska-Mainville[90,91], Luciana Porter-Bolland[92], Martin Quaas[93], Julian Rode[94], Ricardo Rozzi[95,96], Sonya Sachdeva[97], Aibek Samakov[98], Marije Schaafsma[44,99], Nadia Sitas[39], Paula Ungar[100], Evonne Yiu[101], Yuki Yoshida[102] & Eglee Zent[103]

[1]Basque Centre for Climate Change (BC3), Scientific Campus of the University of the Basque Country, Leioa, Spain. [2]Ikerbasque Basque Foundation for Science, Bilbao, Spain. [3]Centre for Development and Environment, University of Bern, Bern, Switzerland. [4]Instituto de Investigaciones en Ecosistemas y Sustentabilidad, Universidad Nacional Autónoma de México (UNAM), Morelia, México. [5]Instituto de Ciencias Polares, Ambiente y Recursos Naturales, Universidad Nacional de Tierra del Fuego (ICPA-UNTDF), Ushuaia, Argentina. [6]Centro Austral de Investigaciones Científicas, Consejo Nacional de Investigaciones Científicas y Técnicas (CADIC-CONICET), Ushuaia, Argentina. [7]Global Science, WWF, San Francisco, CA, USA. [8]Institute on the Environment, University of Minnesota, St. Paul, MN, USA. [9]Aberystwyth Business School, Aberystwyth University, Aberystwyth, UK. [10]Global Resilience Partnership, Cape Town, South Africa. [11]School of International Development, University of East Anglia, Norwich, UK. [12]Helsinki Institute of Sustainability Science, University of Helsinki, Helsinki, Finland. [13]Ecosystems and Environment Research Program, Faculty of Biological and Environmental Sciences, University of Helsinki, Helsinki, Finland. [14]Department of Economics and Management, University of Helsinki, Helsinki, Finland. [15]Department of Food and Resource Economics, University of Copenhagen, Copenhagen, Denmark. [16]Department of International Environment and Development Studies, Norwegian University of Life Sciences, Ås, Norway. [17]Department of Global and Sociocultural Studies and Kimberly Green Latin American and Caribbean Center, Florida International University, Miami, FL, USA. [18]University of EAN, Bogotá, Colombia. [19]Norwegian Institute for Nature Research (NINA), Oslo, Norway. [20]Research Institute for Nature and Forest INBO, Brussels, Belgium. [21]Belgian Biodiversity Platform, Brussels, Belgium. [22]ESSRG Nonprofit Kft., Budapest, Hungary. [23]Wetlands International South Asia, New Delhi, India. [24]Instituto de Investigaciones Sociales, Universidad Nacional Autónoma de México (UNAM), Mexico City, México. [25]Department of Ecosystems and Conservation, College of Forestry, Wildlife and Tourism, Sokoine University of Agriculture, Morogoro, Tanzania. [26]World Wide Fund for Nature (WWF), Culemborg, The Netherlands. [27]Department of Biodiversity and Conservation Biology, Faculty of Natural Sciences, University of the Western Cape, Cape Town, South Africa. [28]Institute for Integrated Management of Material Fluxes and of Resources, UNU-FLORES, United Nations University, Dresden, Germany. [29]Institute for the Advanced Study of Sustainability, United Nations University, Tokyo, Japan. [30]International Centre for Forestry Research and World Agroforestry (CIFOR-ICRAF), Bogor, Indonesia. [31]Plant Production Systems, Wageningen University and Research, Wageningen, The Netherlands. [32]Agroforestry Research Group, Brawijaya University, Malang, Indonesia. [33]Korea Environment Institute, Sejong, Republic of Korea. [34]Université Felix Houphouët-Boigny, Abidjan, Côte d'Ivoire. [35]Centre Suisse de Recherche Scientifique, Abidjan, Côte d'Ivoire. [36]Departamento de Economía, Facultad de Ciencias Sociales y Económicas, Universidad del Valle, Cali, Colombia. [37]Departamento de Biología y Geología, Centro Andaluz de Evaluación y Seguimiento del Cambio Global (CAESCG), Universidad de Almería, Almería, Spain. [38]Department of Environmental Science, Rhodes University, Grahamstown, South Africa. [39]Centre for Sustainability Transitions, Stellenbosch University, Stellenbosch, South Africa. [40]Department of Geography, Institute of Life Earth and Environment, University of Namur, Namur, Belgium. [41]School of Business Administration and Economics & Institute for Environmental Systems Research, Osnabrück University, Osnabrück, Germany. [42]Office for Environmental Ethics, Tübingen, Germany. [43]Charles Perkins Centre, The University of Sydney, Sydney, New South Wales, Australia. [44]Department of Ecology, Swedish University of Agricultural Sciences, Uppsala, Sweden. [45]Institute for Environmental Studies, Vrije University Amsterdam, Amsterdam, The Netherlands. [46]OCP Foundation, Casablanca, Morocco. [47]Rubenstein School of Environment and Natural Resources, University of Vermont, Burlington, VT, USA. [48]Institute of Geography, Universidad Nacional Autónoma de México (UNAM), Mexico City, México. [49]Department of Economics, Indian Institute of Technology Bombay, Mumbai, India. [50]Stockholm Resilience Centre, Stockholm University, Stockholm, Sweden. [51]Global Change Research Institute of the Czech Academy of Sciences, Brno, Czech Republic. [52]Independent scholar, San José, Costa Rica. [53]Kassel Institute for Sustainability, University of Kassel, Kassel, Germany. [54]Faculty of Humanities and Cultural Studies, University of Kassel, Kassel, Germany. [55]Center for Global Indigenous Cultures and Environmental Justice Center, Syracuse University, New York, NY, USA. [56]Escola de Administração de Empresas de São Paulo da Fundação Getúlio Vargas, São Paulo, Brazil. [57]Institute of Environmental Sciences, Boğaziçi University, Istanbul, Turkey. [58]Center for Sustainability Studies, Lund University, Lund, Sweden. [59]Institute of Environmental Planning, Leibniz University Hannover, Hannover, Germany. [60]Ecologos Research Ltd, Aberystwyth, UK. [61]Department of Environment and Geography, University of York, York, UK. [62]Deutsche Gesellschaft für Internationale Zusammenarbeit GIZ, Bonn, Germany. [63]Department of Forestry and Landscape Architecture, Konkuk University, Seoul, Republic of Korea. [64]Karlsruhe Institute of Technology (KIT), Institute of Meteorology and Climate Research, Atmospheric Environmental Research (IMK-IFU), Garmisch-Partenkirchen, Germany. [65]Centre for Environment & Development, ATREE, Bengaluru, India. [66]Indian Institute of Science Education & Research, Pune, India. [67]Shiv Nadar University, Delhi, India.

[68]Department of Philosophy, University of Twente, Enschede, The Netherlands. [69]World Benchmarking Alliance, Amsterdam, The Netherlands. [70]Urban Studies Institute, Georgia State University, Atlanta, GA, USA. [71]Instituto de Investigaciones en Educación, Universidad Veracruzana, Xalapa, México. [72]Centro de Investigaciones en Geografía Ambiental, Universidad Nacional Autónoma de México (UNAM), Morelia, México. [73]Division of Anthropology, Geography and Development, Department of Social and Political Sciences, Brunel University, London, UK. [74]Department of Philosophy and Environmental Studies Program, University of Oregon, Eugene, OR, USA. [75]Faculdade de Economia, Universidade Federal Fluminense, Niterói, Brazil. [76]The Snow Leopard Trust, Seattle, WA, USA. [77]Geography Department, Humboldt Universität zu Berlin, Berlin, Germany. [78]Centre for Climate Justice, University of British Columbia, Vancouver, British Columbia, Canada. [79]University of Winnipeg, Winnipeg, Manitoba, Canada. [80]Universidad Nacional de Colombia, Bogotá, Colombia. [81]International Institute of Tropical Agriculture (IITA), Nkolbisson Yaoundé, Cameroon. [82]Congo Basin Institute (CBI), Nkolbisson Yaoundé, Cameroon. [83]Senckenberg Biodiversity and Climate Research Institute, Frankfurt, Germany. [84]African Leadership College (ALC), Pamplemousses, Mauritius. [85]African Conservation Centre, Nairobi, Kenya. [86]Department of Economics, Boğaziçi University, Istanbul, Turkey. [87]University of Grenoble Alpes, IRD, CNRS, INRAE, Grenoble, France. [88]Centre for Environmental Economics and Policy, School of Agriculture and Environment, University of Western Australia, Perth, Western Australia, Australia. [89]Global Center for Food, Land and Water Resources, Research Faculty of Agriculture, Hokkaido University, Sapporo, Japan. [90]Global and International Studies, University of Northern British Columbia, Prince George, British Columbia, Canada. [91]Nicholaus Copernicus University, Toruń, Poland. [92]Red de Ecología Funcional, Instituto de Ecología, A.C., Xalapa, México. [93]German Centre for Integrative Biodiversity Research (iDiv), Leipzig, Germany. [94]Helmholtz-Centre for Environmental Research (UFZ), Leipzig, Germany. [95]Cape Horn International Center (CHIC), Universidad de Magallanes, Santiago, Chile. [96]Department of Biological Sciences and Department of Philosophy and Religion, University of North Texas, Denton, TX, USA. [97]Northern Research Station, US Forest Service, Evanston, IL, USA. [98]Aigine Cultural Research Center, Bishkek, Kyrgyz Republic. [99]School of Geography and Environmental Science, University of Southampton, Southampton, UK. [100]The Field Museum of Natural History, Chicago, IL, USA. [101]Ernst & Young ShinNihon LLC, Tokyo, Japan. [102]National Institute for Environmental Studies, Tsukuba, Japan. [103]Laboratorio Ecología Humana, Instituto Venezolano de Investigaciones Científicas, Altos de Pipe, Venezuela. ✉e-mail: unai.pascual@bc3research.org

## Methods

This paper's findings are supported by reviews undertaken for the IPBES Values Assessment[11]. These literature reviews came from 15 different disciplines, including social sciences, life sciences, the humanities and interdisciplinary mixed approaches. Review protocols were developed collaboratively by the author team and refined through two open review calls, three Indigenous and local knowledge dialogues and several stakeholder workshops. Different analytical approaches appropriate to different academic perspectives and knowledge traditions reflect various understandings of what makes evidence relevant (contextually appropriate or salient to the issue) and robust (reliable and valid information). Findings are, consequently, presented using qualitative and quantitative approaches.

In total, 29 different assessment protocols (Supplementary Information Sections A and B) were used, focused on five broad topics of nature's values and valuation: (1) the diverse conceptualizations of the values of nature, (2) the ways values can be elicited and made visible through valuation methods and approaches, (3) how values and valuation can be integrated into decision-making processes, (4) the outcomes from such decisions on nature and people, and (5) the role that values play in future sustainability pathways. The reviews encompassed many evidence sources identified using diverse strategies including keyword searches, and natural language processing of 48,781 peer-reviewed papers on nature valuation. The evidence reviewed in depth included 1,163 valuation studies, 1,270 study-site units reporting on values-based outcomes for 217 case studies, 838 documents from the 'grey literature' of environmental and development policy (for example, reports from governmental, non-governmental organizations and valuation initiatives), 26 specific contributions from Indigenous and local knowledge holders and experts, 460 futures scenarios, 37 policy instruments, 217 country-specific datasets (for example, Aichi target 2 progress and UN System of Environmental-Economic Accounting−Ecosystem Accounting implementation) and 134 values-based behavioural theories (Supplementary Information Section B). This evidence was analysed in depth following quantitative and/or qualitative approaches, which were supported by discipline-specific standards. Most evidence sources were in English (96%) and covered 1981–2020, with a greater focus on 2000–2020. Most assessed information was from Western Europe, the USA, Canada and Australia (73%); a smaller share was produced in Asia Pacific (8%), Latin America and the Caribbean (5%), Africa (4%) and Eastern Europe (1%), and 5% had no clear origin.

Assessment protocols followed five general strategies: (1) comprehensive structured reviews using search strings and search terms that defined the review's scope, the different filtering iterations, as well as defined parameters for the selection of the documents to review (Supplementary Information Section B, nos. 4, 9, 10, 19, 29); (2) semistructured reviews relying partially on expert-based search criteria (Supplementary Information Section B, nos. 3, 5, 6, 7, 11, 14, 15, 22, 23, 24); (3) non-structured reviews, fully based on expert criteria (Supplementary Information Section B, nos. 25, 27); (4) invited contributions from external experts and stakeholders through sources such as reports, news articles and art (Supplementary Information Section B, nos. 8, 18); and (5) combinations of the above (Supplementary Information Section B, nos. 1, 2, 12, 13, 16, 17, 20, 21, 26, 28). A diversity of analytics was applied among the 29 reviews, ranging from artificial intelligence-based automated text analysis to in-depth expert assessment of case studies. Furthermore, mixed-methods were used to systematize evidence, including quantitative (for example, frequency, correlation, cluster, geographical analyses) and qualitative approaches (for example, content analysis, analysis of constructs, identification of archetypes). Both inductive (for example, generalization and synthesis) and deductive (for example, hypothesis testing) approaches were applied to evaluate evidence. All protocols, data analyses and results were subjected to formal IPBES assessment procedures. Limitations concerning the accessibility to academic and particularly non-academic literature in diverse languages and the heterogeneous analytical approaches appropriate across disciplines were recognized and explicitly addressed.

### Ethics and inclusion statement

The authors were experts who contributed to the IPBES Values Assessment, selected by IPBES following its formal rules and procedures, following an open call for nominations with consideration for balancing gender, geographical region and expertise to the extent possible. All the experts had specific roles in the production of the Values Assessment in accordance to the IPBES guide on the production of assessments (https://www.ipbes.net/documents/policies-procedures).

### Reporting summary

Further information on research design is available in the Nature Portfolio Reporting Summary linked to this article.

## Data availability

All the data are freely available online. The supplementary information provides links to Zenodo with specific DOIs where the data are stored for free use.

**Acknowledgements** We are grateful to the IPBES, whose 139-member states commissioned the Values Assessment and approved its Summary for Policymakers. We are also grateful for the contributions to the assessment's review editors: S. Anderson, S. Baker, J. Camilo Cardenas, J. Cariño, K. Chan, J. Farley, C. Okereke, L. Pereira, E. Raez, H. Vessuri and R. Watson; the members of the management committee: B. Vilá, A. Díaz-de-León, C. Diaw, M. Avdibegovic, J. Marton-Lefevre and R. Allahverdiyev, and the more than 200 contributing authors who provided specific input to the full report. We express our gratitude to IPBES Executive Secretary A. Larigauderie and IPBES Chair A. M. Hernández for their strategic vision and continued advice. We received no specific funding for this work; all authors involved in IPBES do so on a voluntary basis. The IPBES Values Assessment was made possible thanks to many generous contributions, including non-earmarked contributions to the IPBES trust fund from governments. All donors are listed on the IPBES website www.ipbes.net/donors. U.P. acknowledges BC3's Maria de Maeztu excellence accreditation 2023–2026 (reference no. CEX2021-001201-M) provided by grant no. MCIN/AEI/10.13039/501100011033.

**Author contributions** U.P. led the writing (original draft) of the manuscript and the revisions. U.P., P.B., M.C., B.B. and D.G.-J. coordinated the work of whole the team during the IPBES Values Assessment (2018–2022). P.B., U.P., D.G.-J. and M.C. coordinated the methods section. U.P., P.B., C.B.A., R.C.-K., M.C., D.G.-J., A.M., C.M.R., M.T., A.V., S.A., D.N.B., S.J., E.K., R.K., E.L., T.H.M., B.N., P.O.F., S.M.S. and M.v.N. coordinated the analysis of the evidence and contributed to writing the manuscript. The rest of the coauthors collected and analysed the assessed data.

**Competing interests** The authors declare no competing interests.

**Additional information**
**Correspondence and requests for materials** should be addressed to Unai Pascual.

# Reporting Summary

## Statistics

For all statistical analyses, confirm that the following items are present in the figure legend, table legend, main text, or Methods section.

| n/a | Confirmed | |
|---|---|---|
| ☐ | ☒ | The exact sample size (*n*) for each experimental group/condition, given as a discrete number and unit of measurement |
| ☒ | ☐ | A statement on whether measurements were taken from distinct samples or whether the same sample was measured repeatedly |
| ☒ | ☐ | The statistical test(s) used AND whether they are one- or two-sided<br>*Only common tests should be described solely by name; describe more complex techniques in the Methods section.* |
| ☒ | ☐ | A description of all covariates tested |
| ☒ | ☐ | A description of any assumptions or corrections, such as tests of normality and adjustment for multiple comparisons |
| ☒ | ☐ | A full description of the statistical parameters including central tendency (e.g. means) or other basic estimates (e.g. regression coefficient) AND variation (e.g. standard deviation) or associated estimates of uncertainty (e.g. confidence intervals) |
| ☒ | ☐ | For null hypothesis testing, the test statistic (e.g. $F$, $t$, $r$) with confidence intervals, effect sizes, degrees of freedom and $P$ value noted<br>*Give P values as exact values whenever suitable.* |
| ☒ | ☐ | For Bayesian analysis, information on the choice of priors and Markov chain Monte Carlo settings |
| ☒ | ☐ | For hierarchical and complex designs, identification of the appropriate level for tests and full reporting of outcomes |
| ☒ | ☐ | Estimates of effect sizes (e.g. Cohen's *d*, Pearson's *r*), indicating how they were calculated |

*Our web collection on statistics for biologists contains articles on many of the points above.*

## Software and code

Policy information about availability of computer code

| Data collection | The description of the protocols for the literature reviews undertaken are available in the supplementary information with appropriate DOIs |
|---|---|
| Data analysis | The paper does not rely on any specific algorithm or data software. as it is mainly based on the assessment of the existing literature on the topics covered. |

For manuscripts utilizing custom algorithms or software that are central to the research but not yet described in published literature, software must be made available to editors and reviewers. We strongly encourage code deposition in a community repository (e.g. GitHub). See the Nature Portfolio guidelines for submitting code & software for further information.

## Data

Policy information about availability of data

All manuscripts must include a data availability statement. This statement should provide the following information, where applicable:
- Accession codes, unique identifiers, or web links for publicly available datasets
- A description of any restrictions on data availability
- For clinical datasets or third party data, please ensure that the statement adheres to our policy

All the data used is described in the supplementary information. The supplementary information includes hyperlinks and persistent identifiers (e.g. DOI or accession number) for the data and can be accessed through Zenodo

# Research involving human participants, their data, or biological material

Policy information about studies with human participants or human data. See also policy information about sex, gender (identity/presentation), and sexual orientation and race, ethnicity and racism.

| | |
|---|---|
| Reporting on sex and gender | The paper does not reserach with human participants or human data. |
| Reporting on race, ethnicity, or other socially relevant groupings | The social categorizations are general and they do not refer to ethnicity, race, culture, gender or religion. |
| Population characteristics | See above |
| Recruitment | The study is not experimental so no recruitment approaches need to be described. |
| Ethics oversight | IPBES official rules of procedure were used to conduct the Values Assessment from which the information for the paper was used. |

Note that full information on the approval of the study protocol must also be provided in the manuscript.

# Field-specific reporting

Please select the one below that is the best fit for your research. If you are not sure, read the appropriate sections before making your selection.

☐ Life sciences ☒ Behavioural & social sciences ☐ Ecological, evolutionary & environmental sciences

For a reference copy of the document with all sections, see nature.com/documents/nr-reporting-summary-flat.pdf

# Life sciences study design

All studies must disclose on these points even when the disclosure is negative.

| | |
|---|---|
| Sample size | *Describe how sample size was determined, detailing any statistical methods used to predetermine sample size OR if no sample-size calculation was performed, describe how sample sizes were chosen and provide a rationale for why these sample sizes are sufficient.* |
| Data exclusions | *Describe any data exclusions. If no data were excluded from the analyses, state so OR if data were excluded, describe the exclusions and the rationale behind them, indicating whether exclusion criteria were pre-established.* |
| Replication | *Describe the measures taken to verify the reproducibility of the experimental findings. If all attempts at replication were successful, confirm this OR if there are any findings that were not replicated or cannot be reproduced, note this and describe why.* |
| Randomization | *Describe how samples/organisms/participants were allocated into experimental groups. If allocation was not random, describe how covariates were controlled OR if this is not relevant to your study, explain why.* |
| Blinding | *Describe whether the investigators were blinded to group allocation during data collection and/or analysis. If blinding was not possible, describe why OR explain why blinding was not relevant to your study.* |

# Behavioural & social sciences study design

All studies must disclose on these points even when the disclosure is negative.

| | |
|---|---|
| Study description | The study assesses the literature both theoretical and empirical about the role of nature's values for sustainability |
| Research sample | Literature on 39 different topics (all described in the supp. information) |
| Sampling strategy | Each of the 39 reviews had their own sampling strategy (see supp. information for details) |
| Data collection | Each of the 39 reviews had their own data collection strategy (see supp. information for details) |
| Timing | Study conducted between 2018-2022. |
| Data exclusions | No general exclusion strategies were applied. Each literature review relies on data exclusion and inclusion strategies (see supp. information) |
| Non-participation | THi sis not an experimental study. Participation (or lack thereof) is not an issue. |
| Randomization | No general randomization strategy applied for literature reivews. |

# Ecological, evolutionary & environmental sciences study design

All studies must disclose on these points even when the disclosure is negative.

| | |
|---|---|
| Study description | *Briefly describe the study. For quantitative data include treatment factors and interactions, design structure (e.g. factorial, nested, hierarchical), nature and number of experimental units and replicates.* |
| Research sample | *Describe the research sample (e.g. a group of tagged Passer domesticus, all Stenocereus thurberi within Organ Pipe Cactus National Monument), and provide a rationale for the sample choice. When relevant, describe the organism taxa, source, sex, age range and any manipulations. State what population the sample is meant to represent when applicable. For studies involving existing datasets, describe the data and its source.* |
| Sampling strategy | *Note the sampling procedure. Describe the statistical methods that were used to predetermine sample size OR if no sample-size calculation was performed, describe how sample sizes were chosen and provide a rationale for why these sample sizes are sufficient.* |
| Data collection | *Describe the data collection procedure, including who recorded the data and how.* |
| Timing and spatial scale | *Indicate the start and stop dates of data collection, noting the frequency and periodicity of sampling and providing a rationale for these choices. If there is a gap between collection periods, state the dates for each sample cohort. Specify the spatial scale from which the data are taken* |
| Data exclusions | *If no data were excluded from the analyses, state so OR if data were excluded, describe the exclusions and the rationale behind them, indicating whether exclusion criteria were pre-established.* |
| Reproducibility | *Describe the measures taken to verify the reproducibility of experimental findings. For each experiment, note whether any attempts to repeat the experiment failed OR state that all attempts to repeat the experiment were successful.* |
| Randomization | *Describe how samples/organisms/participants were allocated into groups. If allocation was not random, describe how covariates were controlled. If this is not relevant to your study, explain why.* |
| Blinding | *Describe the extent of blinding used during data acquisition and analysis. If blinding was not possible, describe why OR explain why blinding was not relevant to your study.* |

Did the study involve field work?  ☐ Yes   ☒ No

## Field work, collection and transport

| | |
|---|---|
| Field conditions | *Describe the study conditions for field work, providing relevant parameters (e.g. temperature, rainfall).* |
| Location | *State the location of the sampling or experiment, providing relevant parameters (e.g. latitude and longitude, elevation, water depth).* |
| Access & import/export | *Describe the efforts you have made to access habitats and to collect and import/export your samples in a responsible manner and in compliance with local, national and international laws, noting any permits that were obtained (give the name of the issuing authority, the date of issue, and any identifying information).* |
| Disturbance | *Describe any disturbance caused by the study and how it was minimized.* |

# Reporting for specific materials, systems and methods

We require information from authors about some types of materials, experimental systems and methods used in many studies. Here, indicate whether each material, system or method listed is relevant to your study. If you are not sure if a list item applies to your research, read the appropriate section before selecting a response.

## Materials & experimental systems

| n/a | Involved in the study |
|---|---|
| ☒ | Antibodies |
| ☒ | Eukaryotic cell lines |
| ☒ | Palaeontology and archaeology |
| ☒ | Animals and other organisms |
| ☒ | Clinical data |
| ☒ | Dual use research of concern |
| ☒ | Plants |

## Methods

| n/a | Involved in the study |
|---|---|
| ☒ | ChIP-seq |
| ☒ | Flow cytometry |
| ☒ | MRI-based neuroimaging |

# Antibodies

Antibodies used | *Describe all antibodies used in the study; as applicable, provide supplier name, catalog number, clone name, and lot number.*

Validation | *Describe the validation of each primary antibody for the species and application, noting any validation statements on the manufacturer's website, relevant citations, antibody profiles in online databases, or data provided in the manuscript.*

# Eukaryotic cell lines

Policy information about <u>cell lines and Sex and Gender in Research</u>

Cell line source(s) | *State the source of each cell line used and the sex of all primary cell lines and cells derived from human participants or vertebrate models.*

Authentication | *Describe the authentication procedures for each cell line used OR declare that none of the cell lines used were authenticated.*

Mycoplasma contamination | *Confirm that all cell lines tested negative for mycoplasma contamination OR describe the results of the testing for mycoplasma contamination OR declare that the cell lines were not tested for mycoplasma contamination.*

Commonly misidentified lines (See <u>ICLAC</u> register) | *Name any commonly misidentified cell lines used in the study and provide a rationale for their use.*

# Palaeontology and Archaeology

Specimen provenance | *Provide provenance information for specimens and describe permits that were obtained for the work (including the name of the issuing authority, the date of issue, and any identifying information). Permits should encompass collection and, where applicable, export.*

Specimen deposition | *Indicate where the specimens have been deposited to permit free access by other researchers.*

Dating methods | *If new dates are provided, describe how they were obtained (e.g. collection, storage, sample pretreatment and measurement), where they were obtained (i.e. lab name), the calibration program and the protocol for quality assurance OR state that no new dates are provided.*

☐ Tick this box to confirm that the raw and calibrated dates are available in the paper or in Supplementary Information.

Ethics oversight | *Identify the organization(s) that approved or provided guidance on the study protocol, OR state that no ethical approval or guidance was required and explain why not.*

Note that full information on the approval of the study protocol must also be provided in the manuscript.

# Animals and other research organisms

Policy information about <u>studies involving animals</u>; <u>ARRIVE guidelines</u> recommended for reporting animal research, and <u>Sex and Gender in Research</u>

Laboratory animals | *For laboratory animals, report species, strain and age OR state that the study did not involve laboratory animals.*

Wild animals | *Provide details on animals observed in or captured in the field; report species and age where possible. Describe how animals were caught and transported and what happened to captive animals after the study (if killed, explain why and describe method; if released, say where and when) OR state that the study did not involve wild animals.*

Reporting on sex | *Indicate if findings apply to only one sex; describe whether sex was considered in study design, methods used for assigning sex. Provide data disaggregated for sex where this information has been collected in the source data as appropriate; provide overall numbers in this Reporting Summary. Please state if this information has not been collected. Report sex-based analyses where performed, justify reasons for lack of sex-based analysis.*

Field-collected samples | *For laboratory work with field-collected samples, describe all relevant parameters such as housing, maintenance, temperature, photoperiod and end-of-experiment protocol OR state that the study did not involve samples collected from the field.*

Ethics oversight | *Identify the organization(s) that approved or provided guidance on the study protocol, OR state that no ethical approval or guidance was required and explain why not.*

Note that full information on the approval of the study protocol must also be provided in the manuscript.

# Clinical data

Policy information about clinical studies

All manuscripts should comply with the ICMJE guidelines for publication of clinical research and a completed CONSORT checklist must be included with all submissions.

| | |
|---|---|
| Clinical trial registration | *Provide the trial registration number from ClinicalTrials.gov or an equivalent agency.* |
| Study protocol | *Note where the full trial protocol can be accessed OR if not available, explain why.* |
| Data collection | *Describe the settings and locales of data collection, noting the time periods of recruitment and data collection.* |
| Outcomes | *Describe how you pre-defined primary and secondary outcome measures and how you assessed these measures.* |

# Dual use research of concern

Policy information about dual use research of concern

## Hazards

Could the accidental, deliberate or reckless misuse of agents or technologies generated in the work, or the application of information presented in the manuscript, pose a threat to:

No | Yes
- ☒ ☐ Public health
- ☒ ☐ National security
- ☒ ☐ Crops and/or livestock
- ☒ ☐ Ecosystems
- ☒ ☐ Any other significant area

## Experiments of concern

Does the work involve any of these experiments of concern:

No | Yes
- ☒ ☐ Demonstrate how to render a vaccine ineffective
- ☒ ☐ Confer resistance to therapeutically useful antibiotics or antiviral agents
- ☒ ☐ Enhance the virulence of a pathogen or render a nonpathogen virulent
- ☒ ☐ Increase transmissibility of a pathogen
- ☒ ☐ Alter the host range of a pathogen
- ☒ ☐ Enable evasion of diagnostic/detection modalities
- ☒ ☐ Enable the weaponization of a biological agent or toxin
- ☒ ☐ Any other potentially harmful combination of experiments and agents

# Plants

| | |
|---|---|
| Seed stocks | *Report on the source of all seed stocks or other plant material used. If applicable, state the seed stock centre and catalogue number. If plant specimens were collected from the field, describe the collection location, date and sampling procedures.* |
| Novel plant genotypes | *Describe the methods by which all novel plant genotypes were produced. This includes those generated by transgenic approaches, gene editing, chemical/radiation-based mutagenesis and hybridization. For transgenic lines, describe the transformation method, the number of independent lines analyzed and the generation upon which experiments were performed. For gene-edited lines, describe the editor used, the endogenous sequence targeted for editing, the targeting guide RNA sequence (if applicable) and how the editor was applied.* |
| Authentication | *Describe any authentication procedures for each seed stock used or novel genotype generated. Describe any experiments used to assess the effect of a mutation and, where applicable, how potential secondary effects (e.g. second site T-DNA insertions, mosiacism, off-target gene editing) were examined.* |

# ChIP-seq

## Data deposition

☐ Confirm that both raw and final processed data have been deposited in a public database such as GEO.

☐ Confirm that you have deposited or provided access to graph files (e.g. BED files) for the called peaks.

| | |
|---|---|
| **Data access links**<br>*May remain private before publication.* | *For "Initial submission" or "Revised version" documents, provide reviewer access links. For your "Final submission" document, provide a link to the deposited data.* |
| **Files in database submission** | *Provide a list of all files available in the database submission.* |
| **Genome browser session**<br>(e.g. UCSC) | *Provide a link to an anonymized genome browser session for "Initial submission" and "Revised version" documents only, to enable peer review. Write "no longer applicable" for "Final submission" documents.* |

## Methodology

| | |
|---|---|
| **Replicates** | *Describe the experimental replicates, specifying number, type and replicate agreement.* |
| **Sequencing depth** | *Describe the sequencing depth for each experiment, providing the total number of reads, uniquely mapped reads, length of reads and whether they were paired- or single-end.* |
| **Antibodies** | *Describe the antibodies used for the ChIP-seq experiments; as applicable, provide supplier name, catalog number, clone name, and lot number.* |
| **Peak calling parameters** | *Specify the command line program and parameters used for read mapping and peak calling, including the ChIP, control and index files used.* |
| **Data quality** | *Describe the methods used to ensure data quality in full detail, including how many peaks are at FDR 5% and above 5-fold enrichment.* |
| **Software** | *Describe the software used to collect and analyze the ChIP-seq data. For custom code that has been deposited into a community repository, provide accession details.* |

# Flow Cytometry

## Plots

Confirm that:

☐ The axis labels state the marker and fluorochrome used (e.g. CD4-FITC).

☐ The axis scales are clearly visible. Include numbers along axes only for bottom left plot of group (a 'group' is an analysis of identical markers).

☐ All plots are contour plots with outliers or pseudocolor plots.

☐ A numerical value for number of cells or percentage (with statistics) is provided.

## Methodology

| | |
|---|---|
| **Sample preparation** | *Describe the sample preparation, detailing the biological source of the cells and any tissue processing steps used.* |
| **Instrument** | *Identify the instrument used for data collection, specifying make and model number.* |
| **Software** | *Describe the software used to collect and analyze the flow cytometry data. For custom code that has been deposited into a community repository, provide accession details.* |
| **Cell population abundance** | *Describe the abundance of the relevant cell populations within post-sort fractions, providing details on the purity of the samples and how it was determined.* |
| **Gating strategy** | *Describe the gating strategy used for all relevant experiments, specifying the preliminary FSC/SSC gates of the starting cell population, indicating where boundaries between "positive" and "negative" staining cell populations are defined.* |

☐ Tick this box to confirm that a figure exemplifying the gating strategy is provided in the Supplementary Information.

# Magnetic resonance imaging

## Experimental design

| | |
|---|---|
| **Design type** | *Indicate task or resting state; event-related or block design.* |

| Design specifications | *Specify the number of blocks, trials or experimental units per session and/or subject, and specify the length of each trial or block (if trials are blocked) and interval between trials.* |
| Behavioral performance measures | *State number and/or type of variables recorded (e.g. correct button press, response time) and what statistics were used to establish that the subjects were performing the task as expected (e.g. mean, range, and/or standard deviation across subjects).* |

## Acquisition

| Imaging type(s) | *Specify: functional, structural, diffusion, perfusion.* |
| Field strength | *Specify in Tesla* |
| Sequence & imaging parameters | *Specify the pulse sequence type (gradient echo, spin echo, etc.), imaging type (EPI, spiral, etc.), field of view, matrix size, slice thickness, orientation and TE/TR/flip angle.* |
| Area of acquisition | *State whether a whole brain scan was used OR define the area of acquisition, describing how the region was determined.* |

Diffusion MRI ☐ Used ☐ Not used

## Preprocessing

| Preprocessing software | *Provide detail on software version and revision number and on specific parameters (model/functions, brain extraction, segmentation, smoothing kernel size, etc.).* |
| Normalization | *If data were normalized/standardized, describe the approach(es): specify linear or non-linear and define image types used for transformation OR indicate that data were not normalized and explain rationale for lack of normalization.* |
| Normalization template | *Describe the template used for normalization/transformation, specifying subject space or group standardized space (e.g. original Talairach, MNI305, ICBM152) OR indicate that the data were not normalized.* |
| Noise and artifact removal | *Describe your procedure(s) for artifact and structured noise removal, specifying motion parameters, tissue signals and physiological signals (heart rate, respiration).* |
| Volume censoring | *Define your software and/or method and criteria for volume censoring, and state the extent of such censoring.* |

## Statistical modeling & inference

| Model type and settings | *Specify type (mass univariate, multivariate, RSA, predictive, etc.) and describe essential details of the model at the first and second levels (e.g. fixed, random or mixed effects; drift or auto-correlation).* |
| Effect(s) tested | *Define precise effect in terms of the task or stimulus conditions instead of psychological concepts and indicate whether ANOVA or factorial designs were used.* |

Specify type of analysis: ☐ Whole brain ☐ ROI-based ☐ Both

| Statistic type for inference<br>(See Eklund et al. 2016) | *Specify voxel-wise or cluster-wise and report all relevant parameters for cluster-wise methods.* |

| Correction | *Describe the type of correction and how it is obtained for multiple comparisons (e.g. FWE, FDR, permutation or Monte Carlo).* |

## Models & analysis

| n/a | Involved in the study |
|-----|------------------------|
| ☒ | ☐ Functional and/or effective connectivity |
| ☒ | ☐ Graph analysis |
| ☒ | ☐ Multivariate modeling or predictive analysis |

| Functional and/or effective connectivity | *Report the measures of dependence used and the model details (e.g. Pearson correlation, partial correlation, mutual information).* |
| Graph analysis | *Report the dependent variable and connectivity measure, specifying weighted graph or binarized graph, subject- or group-level, and the global and/or node summaries used (e.g. clustering coefficient, efficiency, etc.).* |
| Multivariate modeling and predictive analysis | *Specify independent variables, features extraction and dimension reduction, model, training and evaluation metrics.* |

