## [Peer Review File · Nature]

Manuscript Title: Diverse values of nature for sustainability

Reviewer Reports on the Initial Version:

Referees' comments:

Referee #1 (Remarks to the Author):

Review of: Nature manuscript 2022-07-11504A
Diverse values of nature underpin just and sustainable futures

PLEASE SEE UPLOADED REPORT

{ Please see page No.10}

Referee #2 (Remarks to the Author):

A. Summary of the key results

The article summarizes some of the key results of the recently published IPBES Methodological assessment regarding the diverse conceptualization of multiple values of nature and its benefits, including biodiversity and ecosystem functions and services. It leverages the idea of values of nature to a language understandable and accessible to all – academics, policy makers, activists. Based on the analysis of more than 50,000 selected publications and thanks to the work of more than 200 experts, the Value Assessment represents a key milestone to influence political and economic decisions to address the current environmental crisis.

The study provides clear and straight-forward information on the topic and proposes a shared language and classification to deal with the heterogeneity of the values of nature, of the methodologies to study them, and of the way people's lives, decisions, principles, and behaviors are influenced by nature's values.

Moreover, the paper shows the current gaps in the valuation processes and methodologies, and their struggles to influence decision making. In particular, the clarifications and typologies proposed help identify which values (and hence stakeholders' worldviews) are excluded from decision making and underline the need to use different methodologies and indicators to fully take in consideration diverse and sometimes incommensurable values to promote change at different "levels" of the typology (worldviews, broad values, specific values). It also discloses the lack of uptake documentation in peer reviewed literature and the need for more engagement with relevant stakeholders.

The paper also interestingly proposes a classification of the different value-assessment methodologies (counted to be more than 50 types) in 4 families distinguished on the basis of the source of value information (nature-based – the most commonly applied –, statement-based, behavior-based, integrated). It also provides comprehensive information of the different types of goals of valuation studies, ranging from improving the state of nature, and enhancing people's quality of life, to generating more socially just outcomes.

Importantly, the paper stresses the need to better assess and integrate in decision-making IPLC's values because they are often guided by principles that promote positive interactions with nature. More broadly, the paper underlines the need to better integrate value assessments in decision making

practice through the increasing of diversity, giving more attention to non-anthropocentric worldviews and non-monetary indicators, and paying more attention to power asymmetries. It is interesting to see how the paper explains which are the main barriers to valuation uptake in policy making – so giving inputs on how such limits may be overcome – and describes the 4 leverage points for transformative change (recognizing the values of nature; embedding valuation into decision-making; reforming policies; shifting underlying societal goals and norms). Furthermore, the paper shows how real transformative change can only be undertaken if market-based instrumental values stop being at the center of political and economic decisions and other types of value – relational, intrinsic and combinations thereof – are incorporated in decision making and promoted among people.

Last but not least, the infographic that is used is astonishing. It incorporates a great amount of information while remaining very clear and direct and – also – beautiful.

B. Originality and significance

The conclusions – based on an incredibly large amount of literature – are novel and unique. No such comprehensive study has ever been done – to my knowledge – on the topic of nature's value assessment and their relationship with decision making. The paper leverages to policy makers very useful, clear and important information that may truly help better integrate nature's values assessments into decision making procedures.

The paper is also interesting for academics studying IPLC TK and worldviews, Payments for Ecosystem Services, Nature Contribution to People, rights-based and community-based biodiversity management and protection, as well as other alternative nature-based models that aim at incorporating people's values, interests and needs in nature protection and conservation.

C. Data & methodology

The data used is very large and was selected through an attentive and well explained (in the SI) methodology. The methodology for analysis is also very robust and well presented in the methodology section which explains in detail how the 29 reviews were prepared, and the methodology used by the involved experts.

SI: the authors have included the doi of each of the 29 protocols produced by IPBES. Adding the title of each protocol could be useful to gain, at first sight, a clearer picture of their structure, content and purpose (for example, the fact that many of them are named after the Chapter of the IPBES Assessment they were used for is a potentially useful information).

The paper is also very well written with simple but precise language.

D. Appropriate use of statistics and treatment of uncertainties (if applicable)

N/A

E. Conclusions

They appear to me to be very robust, clearly explained and relying on a very large body of literature as well as on the expertise of more than 200 experts.

F. Suggested improvements

351: Unsure of why the authors used the term “similarly”, as I do not understand what it refers to.

455: Unsure of how more resources may increase the relevance of a valuation method to a real world decision making procedure.

503: “the vast majority of valuation studies DOES not engage with...”

577: Unsure why the authors refer to fixing market and institutional failures. Given the scope of the other leverage points identified, reference to markets seems too specific. This broader scope is reflected in Figure 4 that refers to “reform policies, rights and regulations”. I would delete reference to markets and add reference to rights.

Pathways of sustainability: Earth stewardship pathways are considered by the authors as relying on the perception of the relational value of nature. While it is surely more symmetric to state so (considering that Green economy is presented as relying on instrumental value, Nature protection on intrinsic value, and Degrowth on the 3 together), Earth stewardship pathways – that I interpret as those closer to so called Earth Jurisprudence – also strongly rely on the recognition of intrinsic value. (On Earth Jurisprudence see, among the other: BURDON P.D. (ed.), *Exploring Wild Law. The Philosophy of Earth Jurisprudence*, Wakefield Press; CULLINAN C. 2002. *Wild Law*, Siber Ink; BURDON P.D. 2015. *Earth Jurisprudence: Private Property and the Environment*, Routledge; BERRY T. 2006. *Evening Thoughts: Reflecting on Earth as Sacred Community*, in TUCKER M.E. (ed.), *Sierra Club Books*.)

G. References: does this manuscript reference previous literature appropriately?

Yes, it does.

H. Clarity and context: Is the abstract clear, accessible? Are abstract, introduction and conclusions appropriate?

Yes, they are.

Giulia Sajeve

Referee #3 (Remarks to the Author):

I am an evidence synthesist (especially focusing on quantitative syntheses) and ecologist, and I am certainly not an expert in valuation studies. However, I should be able to provide my view on the article as a general reader as well as a synthesis expert. I note that this review seems to be a summary of the longer report, “The assessment report on the diverse values and valuation of nature” by IPBES. And this report is based on the results of many systematic maps (scoping reviews).

One: in terms of synthesis methods, it is well-organized and well-done. The method is very transparent, and related documents are archived online, which is excellent. I make two points about the method. I did not see any mention of limitations of their method, which is usually reported for this kind of systematic mapping. It does not need to be part of the main text (or some important limitations can be), but I certainly want to see some discussions on limitations.

Two: One of the important items for systematic maps is stakeholder engagement (e.g. ROSES –

Haddaway et al. 2018). But I did not see any of this, which is surprising as such stakeholder engagements would have been conducted.

Haddaway NR, Macura B, Whaley P, Pullin AS. ROSES RepOrting standards for Systematic Evidence Syntheses: pro forma, flow-diagram and descriptive summary of the plan and conduct of environmental systematic reviews and systematic maps. *Environmental Evidence*. 2018 Dec;7(1):1-8.

Three: I got the main message of this article "we historically focused too much on single values e.g., economic values. We now need to embrace and practice diverse types of valuations", which is great, but it is not news to anybody. Before reading this article, I was hoping to see more quantitative evidence of such. Although this is claimed to be evidence-based, readers may not be quite sure what is evidence-based. It felt like authors' opinions on the trends they observed in the literature.

Four: This is related to my point 3. Although they seem to have some descriptive statistics from the 50,000 documents they reviewed for the study, their narrative does not include statistics. I note that Fig 2 has some stats, but these are not well integrated into the text; I also note "5%" is mentioned in the abstract). I suggest they use more precise language quoting % in the main text.

Five: The repeated and excessive use of acronyms makes it impossible for me to read this article without going back and forth (e.g. NCP, IPLC, LK, PES). And this has prevented me from understanding many seemingly good points the authors are making.

Six: Relating to the point above, The MS used a lot of jargon and introduced a lot of concepts without explaining what they meant. A glossary may be helpful. Due to this and the excessive use of acronyms, the MS reads like impenetrable government reports or social science articles (therefore, I did not find it exciting or informative, at least in the current form, although this is an important study). I consider myself having a good understanding of general things, so for me to find it difficult to understand is a bad sign. Thus, it requires some re-writing and editing to make it a lot more accessible.

Seven: I was wondering why I found this article so difficult to read and understand apart from the points already mentioned. It is probably due to the complete lack of concrete examples (there are no concrete stories people can relate to, even though they must have so many examples). Yes, Fig 1 tries to do this to a certain extent, but a more concrete example would have done better (e.g., a specific river and real people there; I am not saying Fig 1 is bad, but I need some accompanying story to make it relatable). Also, concrete examples related to Fig 3 and 4 will make these concepts more tangible. Also, some recommendations and future directions are suggested, but again without seeing some success stories, it is impossible to see how these things would play out.

Eight: I was also surprised not to see any concrete suggestions for what different parties can do to increase uptake of different valuations in the literature, policy documents and practices (e.g. academics, NGO, politicians, local communities etc). In some sense, they have done this, but it is hard to see it in the current form.

Referee #4 (Remarks to the Author):

A. "Diverse values of nature underpin just and sustainable futures" is a welcome review of and addition to the literature on sustainability because it attempts to deal with two relative neglected and complex realms that underlie human-environmental relations at the national and international levels: human values and knowledge diversity. The key results, based on a review of more than 50,000

scientific publications, policy documents and Indigenous and local knowledge sources by experts involved in the Intergovernmental Science-Policy Platform on Biodiversity and Ecosystem Services (IPBES) Values Assessment, include a basic typology and assessment of nature values and valuation methods and a framework for understanding and applying them within a more rigorous and inclusive sustainability science. The article also offers a (perhaps too-veiled) critique of national and UN Sustainable Development Goals (SDGs), which despite a broad nation-state consensus, “still prioritize a narrow subset of nature’s values, ignoring many of the ways people interact with, care about, and benefit from nature.” Although not a new critique, the case for what is lost by homogenizing nature values—especially as assets and services to humans—is well developed and compelling.

B. The paper is original and significant in that it provides a novel framework for conceptualizing and integrating values and valuation into nature assessments such as the IPBES. It posits that there is a “values crises...at the core of the intertwined crises of biodiversity loss and climate change, pandemic emergence, cultural erosion, and social and political polarization, and social-environmental injustice.” Therefore, values cannot be neglected or taken for granted (e.g., as homogenous, universal, or inherent in markets). The typology of values is also an original and useful heuristic but some categories/distinctions, such as “sociocultural” under “values indicators” beg for further subdivision or specification, as they cover just about everything (arguably even monetary values are sociocultural). Contrastingly, the term worldview, borrowed from anthropology, is a catchall term that seems over-essentialized when subdivided. In other words, cultures that are “cosmocentric” can still be anthropocentric in relation to the cosmos or aspects of it, at least at certain times (indeed this may unavoidable), or be transactional in “living with” nature. While there is value in these distinctions, perhaps, as ideal types (a la Weber), it important to caution the reader that they may not represent any particular culture or group but rather are points along a continuum.

C. The data and methodology are clear enough and follow an established standard for review or systematic review. The >50,000 sources consulted is impressive but how such things as coding the results actually worked, and to what extent it may have involved those with diverse knowledge systems, could have been further developed. The paper is largely conceptual synthesis based the existing literature (published and gray, including ILK of the environment), stressing not just major themes, but key gaps in the literature and how these gaps may be undermining efforts such as those of the IPBES. This approach is laudable and the quality of presentation is generally excellent, especially as concerns the general critique and (counter) proposition and the figures used to support it. The underlying data is there in the copious sources cited, but the analytical procedures themselves could be slightly more elaborated in the body of the paper or notes.

D. As noted above, the reader may wonder about coding procedures and whether statistical analysis of such things as how many cultural groups are classified as possessing a particular worldview in the values typology would be useful (if only to understand the likelihood of getting to a cosmocentric worldview). Nevertheless, the descriptive statistical analyses presented, particularly on what valuation methods toward nature have been used and where, were compelling—and, again, nicely supported with clear, nuanced figures.

E. The conclusions are generally robust, though validity and reliability in some cases seem may rest on the replicability of the methods used to classify the literature. A “values turn” in sustainability and conservation science is overdue and critical to success in organizing the diversity of cultural knowledge systems and environmental values to support the diversity and sustainability of various types of environments and their constituent species. A weakness of the paper is that it does not say much about the causality (or cultural models) leading from values and knowledge to action (pro or anti-environmental in character), though the findings and conclusions do underscore key correlations such as that “Considering locally held or place-based values, for instance through meaningful community

involvement, can lead to more equitable and sustainable outcomes” as well as effective ways of framing engagement and communication with constituent groups about environmental sustainability/conservation aims and objectives. At the bottom of all this, too, lies power and dominance, and whose cultural model of the environment gets operationalized when competing models (and/or the worldviews that underlie them) may be incommensurate.

F. This last point comprises my main suggestion for revision, which is to consider causality beyond worldview a little more carefully. The cultural models framework may be useful in this respect because most of the world's biodiversity and cultural diversity is now subsumed under nation-states and their aggregates (the UN, the EU, etc.) which seem to be, with a few exceptions perhaps, driven by cultural model of sustainable development (epitomized by the SDGs) stemming from a Western notion of 'progress' that, through globalization, has been projected onto other diverse cultures. Elsewhere, one finds very different imagined futures under that globalized projection. Yet cultural models, as shared mental models, do not essentialize the way worldviews do; rather they aggregate mental models which show variation, if not significant divergence, both within and across populations, and can be calibrated as such. This work is also relevant to communication, for as Kempton et al's (1995) seminal work *Environmental Values in American Culture* showed that the unity and diversity of values within nation-states can be important to understand when developing and communicating about environmental change and policy. In fact Americans share important biocentric values which can unite people behind sustainability values, while in other respects they diverge (see also Thornton, T. F. et al. 2019. *Cultural models of and for urban sustainability... Climatic Change* (<https://doi.org/10.1007/s10584-019-02518-2>)) for a more recent review.

G. Beyond those works mentioned above, there are also useful works that critique Western/"WEIRD" (Western Educated Industrialized Rich Democratic) models of ecosystem services (as in the Millennium Ecosystem Assessment) by interpreting/critiquing them through other cultural values systems (e.g., Combetti, et al. 2015. *Ecosystem services or services to ecosystems? Valuing cultivation and reciprocal relationships between humans and ecosystems. Global Environmental Change*, 34, pp.247-262, (<https://doi.org/10.1016/j.gloenvcha.2015.07.007>)). Other useful work on biocentric and biophilia values also exists (e.g., Kellert and Wilson's 1993 *The Biophilia Hypothesis*) and seems worth referencing, especially as it considers the unity and diversity of environmental perception and values in ways not considered in this article. In addition, while the broadly anthropological (and philosophical) perspective on worldview and cultural-environmental values is welcome, including a few seminal references (e.g., Ingold 2000), there is much more that has been done in this field and environmental social science more generally, which might be constructively incorporated, including a number of recent works on Indigenous Knowledge Systems with lots of case examples and references (e.g., Thornton and Bhagwat, eds., *Routledge Handbook on Indigenous Environmental Knowledge*, 2020). Many of these show how different values systems lead to different human-environmental perspectives, relations, and outcomes.

H. Overall the paper is clearly written, contextualized (although could be broader and deeper in places as suggested in specific comments above), and lucid. The figures enhance the argument and presentation in illuminating and useful ways. In addition to the above suggested revisions, I would like to see a slightly stronger conclusion, too, going beyond recognizing the diversity of environmental values and valuations, and prescribing how best to organize and balance the diversity in a way that supports both sustainable biodiversity and cultural diversity at present or restorative levels. That's a tall order given most nation-states' power and continuing preferences for growth and development in unsustainable ways, and requires much more social science and political work around values and diversity that could flow from this investigation.

Referee #5 (Remarks to the Author):

This paper applies a values framework to map out what the global sustainable futures might look like if a diverse range of values was considered in potential pathways towards sustainability. The paper has arisen from the work carried out under the aegis of IPBES and meets the criteria of rigour and global significance. As a review, it is not particularly novel, but the heuristic framework proposed is likely to generate interest and provoke conversations about sustainability pathways. The only area where the paper perhaps falls short is the 'how' question. The paper talks about what needs to be done and this is well argued and justified, but what steps need to be taken to get there in the eight years to 2030 could be covered more directly to trigger change in policy and practice.

Referee #6 (Remarks to the Author):

The authors present a summary of the findings of the IPBES Values Assessment report. These results suggest (paraphrasing) that more diversity in valuation approaches, and a better incorporation of this diversity of approaches into policymaking, can leverage transformative change towards more just and sustainable planetary futures.

I enjoyed reading the paper as it was well-written, engaging, and thought-provoking, and no doubt summarizing a massive volume of work in such a short space was an incredible challenge. The fact this is a summary of a pre-existing report does make it a difficult piece to review, since there isn't really the potential for making suggestions that could lead to additional data collection, analyses, etc. As such I focus my comments on how existing information could be better presented to make the piece more compelling. These detailed suggestions are below, but they boil down to two main areas:

1. Empirical evidence for value systems. The authors suggest that there are a diverse range of values for nature found around the world, but that only a small subset of these values is typically represented in decision making and communication (paraphrasing: instrumental values, anthropocentric worldviews predominate). The authors suggest that this is due to asymmetries in power dynamics, and suggest or imply that many (most?) people hold other latent, pro-environmental values that are waiting to be unleashed. But no empirical evidence – at least here – is presented to justify this take? I completely agree that power dynamics are present and no doubt skewing policies and decision-making in this domain as they do in all others. But an alternative hypothesis is simply that the vast majority of humanity holds anthropocentric, instrumentalist values for nature. And that the other worldviews/values are held by a tiny minority of people, meaning there isn't a great, untapped pool of pro-environmental values just waiting to be unleashed. Do you have empirical evidence to bolster your view that globally it is the former rather than the latter that explains the predominance of the anthropocentric/utilitarian worldview in decision-making?

2. Even if we take at face value the supposition that it is societal power asymmetries and flawed institutions that are excluding the full diversity of nature's values from decision making, and that correcting this can unleash transformational change, the question remains: how exactly can this be done? I understand and appreciate the leverage points that are described, but I believe that a fair reading of the text suggests very little specific actions or pathways are given that could provide a blueprint for how to achieve this. E.g., how exactly can the balance of institutions be changed? How can the influence of powerful, often malign stakeholders be blunted when 'transformational change' is not in their interest? See additional points in specific comments below. There are some key threads from political economy and behavioural psychology that could perhaps be drawn into this discussion, but as it stands, much of the 'how' is left undescribed. And yet this would be a very useful contribution of this piece: to dig into the 'how' in much more detail.

Specific comments:

425: Define 'valuation' here for readers.

468: Perhaps I am misunderstanding what you mean, but I would have thought the motivation for valuation in many/most cases would have been academic exploration for knowledge generation and improved understanding of a particular system? Rather than having some desired end goal in sight?

477: Per above, is this simply because the large majority of the world holds these framings?

503-504: Understandable, as this isn't typically the point of a valuation study itself; rather as you say above, studies can be harnessed by decision-makers for policy purposes.

506: Documenting uptake of a particular valuation study is not necessarily the job of the scientists behind it, and in any case can only happen (much) after the original study is published...perhaps I am misunderstanding but suggests rewording/clarification is necessary.

516-518: Environmental economists will not be surprised at this result!

526-529: This starts to get at the crux of my point above, re: power dynamics / political economy.

545-549: Completely agree.

564-565: How does 'recognizing the diversity of values held across all actors through participatory assessments' actually address asymmetries in power dynamics? On the face of it, not at all. Please elaborate.

589-590: What incentives do powerful decision makers have to 'acknowledge and respect' diverse values of nature when this may threaten existing power structures that favour them?

610-613: Totally agree and this is the crux of the matter...but how exactly can this be done?

615:617: And the balance of these two types of institutions is very clear given the current state of the world. How specifically can this balance be changed?

618-623: Same question as above: how can these underlying goals and norms be changed? Particularly as they are deep and slow moving, which means they will take time to change...and time is of the essence.

665-667: Certainly not true over the past decade-plus of conservation. Take a look at any conservation organization and see how things are often framed around ecosystem services, local communities, indigenous peoples, etc. Or the way the post-2020 CBD agreement is wording things. Suggest this statement needs editing or indeed removal.

669-671: Yet surely this statement could be applied to any particular policy issue. I do not think you need a massive, multi-year values assessment by dozens of global experts to come to this conclusion; it's simply common sense (similarly to a point made above on how policies and interventions work better in areas when people who actually live in that area are engaged in their design). And yet despite this, most often this ideal is not achieved. Why not? It would be useful to draw lessons from

other arenas and see what has worked in instances where this has in fact been achieved. If the authors have already one this, very useful to present here.

Review of: Nature manuscript 2022-07-11504A

Diverse values of nature underpin just and sustainable futures

General Comments

The paper provides a review and appraisal of the literature concerning the diverse values of nature and the extent to which a fuller recognition of those values might deliver a more just and sustainable future. It is in the main a more concise version of Reference 15.

The paper puts forward a series of frameworks for viewing the literature. However, that literature itself contains a prior series of frameworks and it is not obvious that those suggested here are clearly superior, nor that they would lead to an improvement in the incorporation of nature's diverse values within decision making.

The standpoint of the review is academic and pays relatively little attention to some of the key practicalities which have prevented the incorporation of those diverse values to date. For example, little weight is placed on the importance of property ownership as a very significant barrier to such incorporation. If valuation does not result in real world change then it is of little practical use. We could greatly enhance the assessment of values and find that this has no impact upon the decision taken by government or the actions of businesses because of these property rights. As a further example, it is very likely that the intact value of the Amazon greatly exceeds the value of its exploitation for timber and agriculture – but that has not prevented the long term and ongoing loss of the greatest rainforest we have in the world. More consideration of these practical barriers would greatly enhance the usefulness of this review, showing how valuation can play a role here would be a significant contribution.

Specific comments are as follows

Thank you for the opportunity to review your work.

Review of: Nature manuscript 2022-07-11504A

Diverse values of nature underpin just and sustainable futures

Specific comments

Line number	Comment
315-317	The ms. states: “Consequently, a ‘values crisis’ is at the core of the intertwined crises of biodiversity loss and climate change, pandemic emergence, cultural erosion and social and political polarization, and social-environmental injustice”. It would be equally valid (and arguably more useful) to argue that there is a ‘property rights crisis’ at the core of these challenges. We have known that these various values exist for decades, but it has not led to a resolution of these challenges. That is not surprising as naming or even estimating these values won’t lead to them being any more respected than they have been in the past. The conflict between the public values highlighted here and the private values underlying the actions of land-owners or those that pillage natural resources will still persist. How does this paper contribute to addressing that problem?
344 – 350:	The ms. states: “Worldviews are frequently classified in the literature as anthropocentric (i.e. prioritizing human interests or needs²³); biocentric and ecocentric (i.e. emphasizing nature’s inherent or intrinsic value, placing animals, plants and other beings, ecosystems and ecological processes at the centre²⁴); and pluricentric (i.e. a category that recognises worldviews that have no ‘centre’ and instead focus on relationships among human and ‘other-than-human’ beings, as well as nature’s components and systemic processes²⁵).” This perpetuates the (admittedly widespread) misnomer that humans understand and can articulate nature’s intrinsic value. Nature may well have an intrinsic value, but we as humans can never know what that is. All we can ever articulate are human values for the environment. So

	humans can decide that they want certain species to thrive, or that a given environment should continue unchanged. But this is not an intrinsic value for nature; it is an expression of human values – and human values change. What is now seen as pristine wetland was once viewed as an awful bog to be eradicated. Aside from the fact that ‘bringing nature’s intrinsic value into decisions’ mangles the dictionary and proposes an impossible action, such claims confer an entirely spurious moral superiority to such assessments, intended to trump any critique. If we abandon the scientific method in favour of unverifiable claims to being able to measure the unmeasurable (by definition I cannot measure a non-human value; non-human entities such as wild animals or even trees cannot articulate their values in ways we humans understand) then there is no rational basis left for decision making. The environment has to be brought into decision making if we are to avoid global collapse, but abandoning science is not the way to do that. I would object to this paper being published without a very clear statement that the true intrinsic value of nature is by definition unknowable. An honest line would be to accept that humans make the decisions which are dominating the planet and those same humans have very clear preferences for sustainability (and other objectives as well such as improving equity) which need to be respected in decision making – but we cannot know nature’s intrinsic value, only humans value for nature.
356-358	The ms. states: “Broad values that align most with sustainability emphasize principles like justice, stewardship, unity and responsibility³¹⁻³³” The word “most” is almost certainly wrong here. The values that align most strongly with environmental sustainability are likely to be rather repugnant extreme ownership rules where we exclude humans from the use of key natural capital so as to move the world back within planetary boundaries. For example. huge cuts in fertiliser use could significantly help stabilise the climate and water environment. Probably the broad values that would most strongly deliver sustainability are those that advocate massive depopulation of the earth.

	Please rephrase this to recognise that there are some pretty repugnant values which would be highly conducive to sustainability (e.g., slavery and, if we are solely interested in true sustainability of the intrinsic environment, genocide). What you really need to be arguing for is a nexus between environmental sustainability and a variety of other social objectives, which not everyone holds, and which may not be the most conducive to that sustainability. Trade-offs are very likely.
342 - 364	The worldviews vs. broad values vs. specific values categorisation reminds me a lot of the regulating vs. supporting vs. provisioning vs. cultural services paradigm. While the latter caused great excitement amongst academics, I still remain unconvinced that it delivered any substantial contribution to improving real world decision making (and note that this nomenclature has waned markedly from policy documents with no discernible effect en-route). I can see a nice cottage industry of academics classifying values into these various headings; it generates much heat but little light within the decision-making process. What is the rationale for this? It makes for a great graphic, but will it contribute to better decision making? Does it even allow us to understand the world more clearly?
368	The ms. states: “The way people frame their relationships with nature (i.e. their ‘life frames’)” If someone asked me what my ‘life frame’ was that might trigger quite a lot of guesses, but my ‘relationship with nature’ would not be one of them. I don’t see the usefulness of terms like this; they seem very likely to create confusion rather than understanding.
381-390	Figure 1 An inclusive typology of the values of nature clarifies key concepts and their inter-relationships. There is some good work here and the authors are trying to bring a lot of ideas together. I am loath to try and add further complexity to an already challenging graphic but I’m not sure that the different worldviews are as separate as

	this diagram makes them seem. There is evidence within both psychology and sociology that individuals can hold multiple viewpoints at the same time. So, using the example shown, I can view fish as a commodity to be sold for a market price while at the same time seeing fish as regulators of food webs and enjoying eating fish. Using the labels as shown in the diagram this would make me Anthropocentric, Bio/ecocentric and Pluricentric all at the same time. Given this then how useful are these labels? I am interested to note that the that the intrinsic value label is not applied to the first two columns. Given my comments previously I would argue that it cannot be applied to any of these columns. If humans want to preserve wild species then that is a reflection of human values, not non-human intrinsic values. My previous comment on the term ‘life frames’ applies here also.
402-408	The ms. states: However, some values are incommensurable, i.e. they are neither comparable nor compatible with other values. For example, while development projects are associated with instrumental values (e.g. economic and health benefits), they may also affect relational values (e.g. loss of sense of place). Even though these values can be incommensurable, decisions can consider them in parallel, such as through deliberation with affected parties or granting autonomous decision-making to Indigenous peoples and local communities (IPLCs) within their territories to maintain their own mix of instrumental, intrinsic and relational values⁴⁹⁻⁵¹. The problem of comparability is one I fully acknowledge; measurement is challenging. However, the above text also states that some values are not ‘compatible’ with other values. While this term is not defined it suggests that there are some values which, in principle (as opposed to in practice), cannot be traded off with others. Now sustainability requires that we maintain certain stocks of natural assets above given levels, below which

vital earth processes are threatened. But that is quite different from a value which is truly incompatible with trade-offs.

Consider even the atmospheric services of a climate operating within planetary boundaries. This might not be the situation we are in now, but if we really did reverse global heating then we would again allow humanity the safe operating space to allow for trade-offs between other sources of wellbeing and greenhouse gas emissions. So even the atmosphere is perfectly compatible with other values once we are within that safe operating space – its simply that we are at present a long way from that position.

There is a real danger in declaring that multiple different values are incompatible with other values. We will then have a profusion of special cases. This leads to two practical problems:

First, every special case trumps all other value assessments. Because it contains values that, according to the above definition, are not compatible with any other values then there can be no trade-off with any other value. So that case HAS to be funded under these rules. This of course will lead to an explosion of such special cases and any assessment which does not have one will find that all available resources have already been allocated elsewhere.

Second, the above situation would of course quickly prove untenable and the ‘special case’ rule would be ignored.

The result is that we are back at square one – except that we have forfeited the credibility of assessments.

This is not a reasonable approach for real world decision making. The appeal to

“...deliberation with affected parties or granting autonomous decision-making to Indigenous peoples and local communities (IPLCs) within their territories to maintain their own mix of instrumental, intrinsic and relational values⁴⁹⁻⁵¹”

is merely a call to respect property rights (which I fully endorse; see previous comments to this effect) dressed up as values assessment. It is not and should not be presented as such.

428-441

The ms. states:

These methods can be organized into four cross-disciplinary ‘method families’, based on their source of value information: 1) nature-based valuation gathers information about the importance of nature and NCP^{65,66} through direct and indirect observation of nature (e.g. spatial mapping of ecosystem services⁶⁷), 2) statement-based valuation obtains information from people’s expressions of their values (e.g. stated preference surveys⁶⁸; or deliberative processes⁶⁹), 3) behaviour-based valuation identifies how people value nature by observing what they do in relation to nature (e.g. hedonic pricing^{70,71}; or livelihood dependence⁷²), and 4) integrated valuation brings together different types of values assessed with different information sources (e.g. participatory rural appraisal⁷³; integrated modeling⁷⁴).

And subsequently:

IPLC valuation methods

There is a very well established set of terms for valuation methods, why has it been rejected?

The existing typology is:

- Market based
- Non-market: Revealed preference
- Non-market: Stated preference

The paper rejects this terminology. Instead it provides its own typology which seems very likely to cause confusion. Terms like “integrated valuation” suggest that all other types of valuation are not integrated. Furthermore, the definition of mapping as ‘nature based valuation’ is simply misleading; mapping is not valuation.

This seems like unnecessary invention of new terms for their own sake. Decision makers are confused enough without further loading such as this.

441-447 and 462-467	The discussion of how to incorporate the values of Indigenous peoples and local communities within assessments is interesting and topical. However, the link to property rights challenges needs to be made. There is little point in incorporating IPLC values into assessments if those will be ignored on the ground because of property right absences or violations.
468-471	Valuations carried out for conventional business sector decision making are ignored here. While I understand that, as they are likely to be the majority of all assessments then the authors need to note this demarcation of their analysis. More complex though will be the large and growing number of assessments carried out by the private sector for mixed purposes of both regular investment appraisals and analysis of the environmental and social impacts of decisions. These appear to have been omitted both from this text and Figure 2 and it is therefore somewhat unclear what the criteria were for inclusion within this analysis (see next comment).
468-471 and 479-488 and Supplementary Info	The text reports that 48,781 studies were reviewed. While this at first seems quite remarkable, a review of the Supplementary Information (SI) document shows that this is the number of papers that were delivered by a web browser search of terms principally on four databases: Web of Science; Scopus; Google Scholar; and EBSCOhost (Academic Search Premier). This very large number of studies was then reduced to some 1163 studies based on a series of rules set out in the SI. It is not possible to assess the defensibility of the precise rules used for inclusion of papers as applied across the 12 documents contained in the SI. For example, in Information Document 3 papers are accepted into the review according to an unspecified criterion based on the paper citation score normalised for publication date. These rules vary across the individual Information Documents depending on the ‘Specific topics supported’ and ‘Type of review’ criteria.
479-488	The importance of understanding and validating the criterion used for selecting studies is crucial to the interpretation of the results presented in Figure 2. Figure 2. Global distribution and characterization of nature valuation studies reported in the literature.

	If the selection of studies is defensible then the world map of studies presented here is both expected and interesting as it provides a quantification of the focus on Europe within valuation studies. Similarly the categorisation of studies by habitat type was an interesting finding. A comparison of this to the physical distribution of land use and globally would be interesting as it will reveal that the marine environment is massively under-investigated. The authors might repeat this just for terrestrial land use types and studies. Doing this with or without Antarctica will reveal that we ignore the latter as well. A comparison of remaining land use distribution with study habitat might then show the focus decisions of researchers. The very clear dominance of ‘Nature-based valuation’ studies in the selected literature was of some concern. Given that this category includes mapping studies which, I would argue, are not valuation assessments then the fact that these represent 68% of the selected studies suggests that this decision by the authors has had a huge impact upon analyses. A similar concern arises regarding the dominance of biophysical measures within the Value Indicator assessment.
490-501	I found Figure 3 both interesting and useful. However, its relation to the studies assessed is not made obvious, indeed this looks rather stand-alone.
503-504	The ms. states: “the literature suggests that the vast majority of valuation studies do not engage with relevant stakeholders⁶⁴” This is a very sweeping statement to make from just a single reference (a further review by the same authors which may be the source document for much of this paper). If this is true, and it is likely to be contended, then it is very likely to also apply to non-valuation studies which in other respects adopt similar survey or experimental methods – which would suggest that labelling this as a problem of valuation research is misplaced.
505-506	The ms. states: “during the last three decades the share of peer-reviewed studies documenting uptake has not increased”

	It is not clear what the phrase “ documenting uptake ” means.
526-528	The ms. states: “the power of stakeholders with more resources can hinder the representation of diverse values in decisions” I strongly agree with this statement – but note that it comes after a long section advocating greater use of stakeholder perspectives. The solution to this conundrum is not obvious here.
532-533	The ms. states: “Considering locally held or place-based values, for instance through meaningful community involvement, can lead to more equitable and sustainable outcomes” The difference between “can” and “does” is crucial here and relates back to the previous issue. There is a strong assertion here that ‘local’ and ‘stakeholder’ will enhance equity and sustainability. Of course this can be the case – but the opposite can also hold. Local decision making can lead to domination by local power-bases. In such cases, more remote decision making might well prove more impartial. The local = good equation offered here is too simple. What are the designs and criteria needed to deliver more equitable and sustainable outcomes? The inference that local stakeholders will deliver such results is not a sufficient argument on its own. This criticism applies to most of this paragraph.
531-567	While I accept that the points made in Section 3 are referenced I found it very difficult to judge the weight of evidence. The section makes a long series of statements: that local is better than national, stakeholder better than population, indigenous better than other, power is in league with development, etc. This may all be true but I could not judge the strength of evidence regarding these statements
570-579	This is a nice summary of the IPBES report produced by the authors.

580-587	Is this figure from the IPBES report? I think it is and it should be clearly marked as the source.
570-587 and 618-623	The framework presented here is nice and clear and the continuum developed through the IPBES report is useful. There is a fundamental difference between stages 1-3 and stage 4 and I feel that needs highlighting. Stages 1-3 are different levels of recognising the world as it is and incorporating the values it generates within decision making. However, stage 4 is very different; it is an attempt to change those values. There is an existing literature on this (e.g. the Arrow et al ‘social norms’; paper) that needs acknowledging. More importantly I feel it would be useful to give the reader some idea as to how these changes are to be delivered. The GDP example is great – but a discussion of how ordinary peoples’ preferences and values can be influenced would be very helpful here. I would suggest a two pronged approach should be highlighted. First existing preferences can be used to modify behaviour – for example via carbon taxes on food, fuel, etc. This I feel fits in with earlier stages in the continuum but clarifies the difference with the second approach. The second approach focusses on the modification and outright change of preferences. Again you can highlight extant literature here – general preferences regarding climate change and biodiversity loss have been the focus of preference altering information for years; some of this has been high profile (for example the impact of the David Attenborough Blue Planet programmes on plastics pollution). Value change can be very effective – e.g. the social norms regarding smoking indoors and drunk driving have changed radically. However, with all of this there should be acknowledgement that there is no single golden bullet and all of these levers across the full continuum need to be applied to deliver a sustainable world. As a final aside, which you may or may not wish to use; note that in principle there is no difference between trying to manipulate values to move in a pro-sustainability direction than the manipulation of preferences to deliver outcomes which we now consider repugnant. The Nazi regime realised the power of such manipulation and used it with great effectiveness to change preferences in disgusting ways. This raises issues of morality and power which I feel would be an interesting and

	honest insight. Who decides on the goals of such exercises? What is the moral basis of determining that a “just and sustainable future” should be that goal? What is the balance of risks associated with developing mechanisms to alter mass-preferences?
626-659	I will flag up to the Editor that I am not a fan of the vast majority of scenario analyses and that she/he should take that into account and might wish to dismiss this point as reflecting my own opinions and not the majority of the literature. I feel that most scenario analyses are of extremely limited value and some are simply misleading. The major failing that nearly all of them have in common is the absence of a supporting analysis to assess, in quantified terms, the trade-offs associated with moving between scenarios. In my experience the typical scenario study (including some very high profile cases) are basically exercises in policy persuasion dressed up as analysis. They contain a scenario where unbridled expansion of industry and land use intensification is contrasted with a Business as Usual ‘baseline’ and a couple of pro-nature alternatives. The conclusion is inevitable: pro-nature is best for nature and delivers sustainability and may improve distribution. I feel such exercises are simply unscientific. That does not mean they do not have a policy message – there are alternatives to the status quo. That’s a useful story and could well have real world impact. But this is typically not an academically sound undertaking. Unless scenario analyses are backed by rigorous analysis of the trade-offs each scenario entails then they are policy briefs, not academic research. As I say, the Editor might decide this should be ignored on the grounds that scenario analyses are prevalent and (as I acknowledge) may have policy impact. But I feel that most of them should be confined to the realms of political science.
667	Who are these “local people”? Is everyone in the world a ‘local person’?
671-674	The mention of rights is important and highlighted in my earlier comments. However, the authors here are referring to moral rights rather

	than property rights – and very frequently in the world the latter trumps the former, indeed this is the root of many of the challenges facing indigenous people. Their property rights are often not recognised in law and/or respected in practice. I feel the authors needed to acknowledge this and highlight how important this issue is throughout the paper. Put simply, it often doesn't matter what indigenous values are, or whether or not they are incorporated in assessments, if property rights problems mean that in practice those values are not respected.
683-696	The latter part of this paragraph begins to approach the property rights issue – but then shies away and retreats into a discussion of citizens assemblies. These will remain ineffective if property rights are not well defined and respected in practice. Given that a lot of this paper seems motivated by a desire to see indigenous preferences (and ‘citizens’ values – but see my comment about ‘local’ – everyone is a ‘local’ ‘citizen’ so your terminology includes the whole world) I feel the omission of a serious discussion of property rights running throughout the paper is a problem.
722-760	Please see my comments on the SI – above
827	Ref: 16. IPBES. Methodological assessment of the diverse values and valuation of nature of the Intergovernmental Science-Policy Platform on Biodiversity and Ecosystem Services. P. Balvanera, U. Pascual, M. Christie, B. Baptiste, D. González-Jiménez (eds.). IPBES secretariat, Bonn, Germany (2022). https://doi.org/10.5281/zenodo.6522522. This is an important element of the review as it should provide details of the methodology. The link leads to the appropriate IPBES report however the file itself when downloaded is blank.

Author Rebuttals to Initial Comments:

	Line num. (original ms)	Reviewers' comments	Responses to reviewers' comments
Editor			
1	General (editor's comment)	Please focus more on the practical limitations to the consideration of the values of nature, including property rights	Rather than listing the many potential limitations of our work, we have focused mostly on what can be done, which also addresses our scope and reach (e.g., applying the values typology all the way to activating leverage points, and have brought in new examples to highlight these mostly untapped possibilities). Given the important point about property rights, we have included reference to this (not as a limitation, but as a further issue that needs to be addressed in conjunction with activating or forming values that need to be integrated into decision making), and we have further clarified the ways considering nature's values could trigger transformative changes towards sustainability (see new section 5, where we mostly focus on the 'how' issue). We provide detailed information regarding the way we have revised the paper trying to address head on each of the comments received from the six referees.
2	General (editor's comment)	Please examine/re-examine underlying assumptions and categorisations (and adjust/better justify where necessary, or acknowledging parallels to other extant categorisations).	We have made more explicit how this assessment presents the findings of 29 separate reviews that have brought together evidence from diverse sources and synthesised categories to facilitate comprehension and application of different conceptualizations of value, valuation and actions upon values-centred leverage points for transformative change. Inherently, this exercise implies making decisions regarding categories throughout (e.g., the values typology, the methods families, classifying leverage points). We have now more explicitly clarified the basis for each of these categorizations and also demonstrated their utility. While there are limitations to any analytical approach, we do not claim any of these are the only way of approaching any issue from a disciplinary or specific perspective. Instead we do claim they are a singularly integrative and inclusive approach to dialogue between different analytical perspectives and are helpful to present a unified assessment (not just of concepts, but also ways to use them in practical applications).

3	General (editor's comment)	If it's not possible to render the piece more quantitative (referee 3), please make it clearer that this is in part opinion-based.	We have added more information on the sources of evidence throughout the paper to make clear where assertions are supported by quantitative vs. qualitative evidence. We also emphasise that both forms of evidence are valid, based on the standards of different disciplines/knowledge traditions. Since it is our intention to dialogue between disciplines and knowledge perspectives to communicate with the entire community of scholarship and practice involved in the study and management of biodiversity and nature(s), we do not refer to qualitative assessments as opinion-based, but rather make attempt to make explicit the limitations and opportunities that different approaches provide to the understanding and governance of biodiversity and nature. We have also added some more quantitative results, taking into account the word limit.
4	General (editor's comment)	Please include specific examples to illustrate the various points.	We have added examples throughout the paper. Besides general illustrations for various points that may be seen as more theoretical (e.g., illustrating tangible factors that affect behaviour like motivation, capacity, and ability or different dimensions of power), we continue to use various specific policy instrument examples, such as information related to protected area management and payment for ecosystem services. In the new version of the paper, we now highlight one specific illustrative case of a relatively successful wetland management and restoration from India (see new supplementary information section C on the case of the Chilika Lagoon/wetland). This case is first referenced in Section 1 and then echoed in section 5 to provide a better narrative and storyline. We hope that this example will help the readers translate some key ideas with a real world case and thus connect those ideas to other cases in their own contexts (see also response to Comment #60). Furthermore, this issue is connected to addressing the next point, as we have now included specific examples to help address 'how' values can be clarified with the typology of values, 'how' values enter into the policy cycle, and what options (via examples) there exist to catalyse values-centred leverage points for transformative change.
5	General (editor's comment)	Please consider more closely the 'how', when it comes to integrating the manifold values of nature into decision-	We have paid special attention to this comment raised by various reviewers. Besides including new examples (see previous response), we have also restructured the manuscript to increase the focus on the 'how' issue. The

		making. This would also better set this apart from the summary for policymakers.	original sections 4 and 6 have been merged into new section 5, which keeps the original points associated with the activation of values-based leverage points to catalyse transformative change in decision-making and adds examples for each. This has also given us the opportunity to improve the way we explain the interconnections and overlaps among the four leverage points, which we did not mention explicitly in the original version. Furthermore, we have paid special attention to the core argument raised by reviewer 1 regarding the role of property rights, which also accompanies the values perspective in transformative change.
6	General (editor's comment)	And something to consider, is the word 'just' okay? In the social sciences in particular this can have different definitions. Would it be better to keep to 'sustainable', which would include this?	The Values Assessment focused on the dual objectives of justice and sustainability, as the two are interrelated. Furthermore, we employ the expression 'just and sustainable futures' precisely to emphasise the importance of considering both social and environmental goals together in sustainable development discourse and practice (as in some scholarly and practical traditions they are addressed separately). As evidence for the relevance of this approach, one can observe the recent increase in the use of 'just and sustainable futures' in the context of UNESCO and more generally the UN to emphasise the importance of ensuring that environmental and development policies (e.g., climate change policies connected to REDD+ or Green Economy schemes) are also fully inclusive of the principle of justice.
Reviewer 1			
7	General	The paper provides a review and appraisal of the literature concerning the diverse values of nature and the extent to which a fuller recognition of those values might deliver a more just and sustainable future. It is in the main a more concise version of Reference 15. The paper puts forward a series of frameworks for viewing the literature. However, that literature itself contains a prior series of frameworks and it is not obvious that those suggested here are clearly superior, nor that they would lead to an improvement in the incorporation of nature's diverse values within decision making. The standpoint of the review is academic and pays	Thank you for this comment, We note that the reviewer raises an important issue regarding property rights. This topic is now dealt with more clearly in the broader treatment of 'how' to use values into decisions, which was also raised by other reviewers. Instead of making changes 'here and there', we have concentrated our responses to the 'how' issue - including referring to the aspect of property rights mentioned by reviewer 1 - in a new section 5 that combines and adds to previous content from the original sections 4 and 6. This way, the responses are linked systematically to each leverage point (Figure 4). More specifically regarding property rights, this issue is part of a more general problem regarding the needed changes to goals, policies, and measures (now explicitly mentioned) at the outset of the manuscript (i.e. in

		relatively little attention to some of the key practicalities which have prevented the incorporation of those diverse values to date. For example, little weight is placed on the importance of property ownership as a very significant barrier to such incorporation. If valuation does not result in real world change then it is of little practical use. We could greatly enhance the assessment of values and find that this has no impact upon the decision taken by government or the actions of businesses because of these property rights. As a further example, it is very likely that the intact value of the Amazon greatly exceeds the value of its exploitation for timber and agriculture – but that has not prevented the long term and ongoing loss of the greatest rainforest we have in the world. More consideration of these practical barriers would greatly enhance the usefulness of this review, showing how valuation can play a role here would be a significant contribution.	the ‘bold’ paragraph). We include specific comments on rights and property rights, but have not modified our assessment to make this ‘the fundamental issue’, which one could interpret to be implied from this comment. To clarify, this is a paper on the fundamental role that a value focus could play in environmental scholarship and decision-making. Hence, that is still its core, while political implications, including property rights, are now more clearly spelled out and illustrated across the sections. Please see further responses on the use of frameworks.
8	315-317	The ms. states: “Consequently, a ‘values crisis’ is at the core of the intertwined crises of biodiversity loss and climate change, pandemic emergence, cultural erosion and social and political polarization, and social-environmental injustice”. It would be equally valid (and arguably more useful) to argue that there is a ‘property rights crisis’ at the core of these challenges. We have known that these various values exist for decades, but it has not led to a resolution of these challenges. That is not surprising as naming or even estimating these values won’t lead to them being any more respected than they have been in the past. The conflict between the public values highlighted here and the private values underlying the actions of landowners or those that pillage natural resources will still persist. How does this paper contribute to addressing that problem?	Thank you also for this comment. We have rewritten the text. We maintain that we are facing a ‘values crisis’ associated with a series of ‘environmental crises’ - of which biodiversity loss is a key one. We, therefore, retain the argument that the basic challenge regards the limited set of values underpinning present policies and decisions. We agree that failed property rights are among the present deficiencies of policies, goals, and measures. We note, however, that it is not the only one. As stated in our response to Comment #7, the property rights concern raised by Reviewer 1 now is covered more specifically in the new section 5.
9	344–350	The ms. states: “Worldviews are frequently classified in the literature as anthropocentric (i.e. prioritizing human	Thank you for sharing this concern (this response is also part of how we addressed Comment #13). We do acknowledge there is a long, extensive

	interests or needs²³); biocentric and ecocentric (i.e. emphasizing nature's inherent or intrinsic value, placing animals, plants and other beings, ecosystems and ecological processes at the centre²⁴); and pluricentric (i.e. a category that recognises worldviews that have no 'centre' and instead focus on relationships among human and 'other-than-human' beings, as well as nature's components and systemic processes²⁵)." This perpetuates the (admittedly widespread) misnomer that humans understand and can articulate nature's intrinsic value. Nature may well have an intrinsic value, but we as humans can never know what that is. All we can ever articulate are human values for the environment. So humans can decide that they want certain species to thrive, or that a given environment should continue unchanged. But this is not an intrinsic value for nature; it is an expression of human values – and human values change. What is now seen as pristine wetland was once viewed as an awful bog to be eradicated. Aside from the fact that 'bringing nature's intrinsic value into decisions' mangles the dictionary and proposes an impossible action, such claims confer an entirely spurious moral superiority to such assessments, intended to trump any critique. If we abandon the scientific method in favour of unverifiable claims to being able to measure the unmeasurable (by definition I cannot measure a non-human value; non-human entities such as wild animals or even trees cannot articulate their values in ways we humans understand) then there is no rational basis left for decision making. The environment has to be brought into decision making if we are to avoid global collapse, but abandoning science is not the way to do that. I would object to this paper being published without a very clear statement that the true intrinsic value of nature is by definition unknowable. An honest line would be to accept that humans make the decisions which are dominating the	debate in the literature regarding intrinsic values (indeed, there are important nuanced philosophical debates regarding most of these value types, which this article cannot fully develop). However, we have attempted to be inclusive of different schools of thought and intellectual traditions on intrinsic values. Upon careful and extensive reflection, we decided to take the position that aims to bridge these, whereby one can value something 'intrinsically' for its inherent properties (e.g., being a subject of a life) or as others would argue, in itself and not for 'what it does to me nor for how it benefits me'. Therefore, the way we have chosen to use the concept of intrinsic value emphasises the value that something has regardless of any reference to people (as valuers). This does not mean that people are not the valuers, but the reason why something is valued (note that specific values are defined as the 'justification of value') does not depend on them being the valuers. This approach is epistemically anthropocentric and anthropogenic (i.e. people are the valuers), but not morally anthropocentric (i.e. people are superior to other beings or central to the reason why something is valued). Protecting nature for its intrinsic worth is thus a reflection of (broad) human values. In the IPBES Values Assessment, we specifically focused on how people express/embody/articulate values – many people consider intrinsic values of nature important as motivation or for moral reasons. We seek to reflect this in the typology of values and analysis without entering the extensive debate about whether intrinsic means objective or subjective. We hope this clarifies our position and the reasons why and how we use the concept of intrinsic values.
--	---	---

		planet and those same humans have very clear preferences for sustainability (and other objectives as well such as improving equity) which need to be respected in decision making – but we cannot know nature’s intrinsic value, only humans value for nature.	
10	356-358	The ms. states: “Broad values that align most with sustainability emphasise principles like justice, stewardship, unity and responsibility 31– 33” The word “most” is almost certainly wrong here. The values that align most strongly with environmental sustainability are likely to be rather repugnant extreme ownership rules where we exclude humans from the use of key natural capital so as to move the world back within planetary boundaries. For example, huge cuts in fertiliser use could significantly help stabilise the climate and water environment. Probably the broad values that would most strongly deliver sustainability are those that advocate massive depopulation of the earth. Please rephrase this to recognise that there are some pretty repugnant values which would be highly conducive to sustainability (e.g., slavery and, if we are solely interested in true sustainability of the intrinsic environment, genocide). What you really need to be arguing for is a nexus between environmental sustainability and a variety of other social objectives, which not everyone holds, and which may not be the most conducive to that sustainability. Trade-offs are very likely.	We appreciate the reviewer’s comment, although we do not fully agree with the premise that social injustice and associated repugnant values can be aligned with sustainability. This difference is clearly a matter of definition/conception of sustainability itself, which we cannot enter into in the paper. However, we and the general scholarship and policy on sustainability comes from a position that it is both social and ecological (i.e. not compatible with eliminating people from the earth e.g. for efficiency goals) and long-term (and thus dependent on cooperation and mutuality in the long run to be effective). We appreciate there can be other ways of conceiving sustainability in which it may not necessarily be compatible with social justice (e.g., sustainability in terms of natural resource optimization and intergenerational distribution). Rather than get into this discussion here, since there is no real debate among the policy documents nor scholarship we have reviewed, we opt for a simple remedy which is to clarify that the ‘alignment’ of justice etc. with sustainability is something we find frequently in the literature and it is at the core of the Values Assessment position. Specifically, our review of 460 futures visions/scenarios clearly found that ‘sustainable’ scenarios were most often characterised by the kinds of broad values mentioned here. Also, this is in keeping with the broad values expressed in major policy documents, like the SDGs and the CBD’s GBF. We have also rewritten the sentence in the second para of section 1: “... certain broad values like justice, stewardship, unity, and responsibility are frequently found to align with sustainability (Martin et al. 2022).”
11	342 - 364	The worldviews vs. broad values vs. specific values categorisation reminds me a lot of the regulating vs. supporting vs. provisioning vs. cultural services paradigm. While the latter caused great excitement amongst	We thank the reviewer for this constructive feedback. We have substantially revised the manuscript’s section 1 to better clarify the relevance of the typology of values for decision-making. Specifically, we highlight the applied relevance of the typology with a real world example

		academics, I still remain unconvinced that it delivered any substantial contribution to improving real world decision making (and note that this nomenclature has waned markedly from policy documents with no discernible effect en-route). I can see a nice cottage industry of academics classifying values into these various headings; it generates much heat but little light within the decision-making process. What is the rationale for this? It makes for a great graphic, but will it contribute to better decision making? Does it even allow us to understand the world more clearly?	from a wetland case (as requested in Comment #60 to provide tangible storylines so readers can relate the typology as an analytical tool useful for decision-making). Given that we have had to address many comments, we also had to condense the section.
12	368	The ms. states: “The way people frame their relationships with nature (i.e. their ‘life frames’)” If someone asked me what my ‘life frame’ was that might trigger quite a lot of guesses, but my ‘relationship with nature’ would not be one of them. I don’t see the usefulness of terms like this; they seem very likely to create confusion rather than understanding.	The four life frames are different ways of ‘framing’ relationships to nature (in the broadest sense - i.e. including ecosystems, particular natural entities, other-than-human species, etc.). The four ways presented here represent nature as a resource (living from nature), nature as a place or setting for people’s lives (living in nature), nature as cycles, processes, species etc. other than the human world (living with nature), and nature as self (living as nature), such as through kinship, spirituality, embodiment, etc. The first three frames were presented by O’Neill et al. (2008), with the fourth frame emerging from the Values Assessment’s work and O’Connor and Kenter (2019) in recognition of other (particularly non-Western) ways of being/living that do not recognize a nature/culture dichotomy. The Values Assessment used these four generalised life frames as a heuristic way to help organise and communicate different sets of values. The word ‘life’ is the noun for the adjective living that helps term each of the frames, hence ‘life frames’, but life can also be read to refer to nature in shorthand without suggesting a separation between people and nature, which the framework seeks to overcome. Nonetheless, these are clearly heuristic devices, rather than static categories. We clearly show how they can be used to see how different peoples or policies may prioritise certain values in different contexts, making them useful as well as conceptually rigorous, based on their applicability to both academic and policy literature. See full reviews in Anderson et al. (2022). References: O’Connor and Kenter (2019) Sustain. Sci. 14; O’Neill et al (2008). “Environmental Values”. Routledge.

13	381-390	Figure 1 An inclusive typology of the values of nature clarifies key concepts and their inter-relationships. There is some good work here and the authors are trying to bring a lot of ideas together. I am loath to try and add further complexity to an already challenging graphic but I'm not sure that the different worldviews are as separate as this diagram makes them seem. There is evidence within both psychology and sociology that individuals can hold multiple viewpoints at the same time. So, using the example shown, I can view fish as a commodity to be sold for a market price while at the same time seeing fish as regulators of food webs and enjoying eating fish. Using the labels as shown in the diagram this would make me Anthropocentric, Bio/ecocentric and Pluricentric all at the same time. Given this then how useful are these labels? I am interested to note that the that the intrinsic value label is not applied to the first two columns. Given my comments previously I would argue that it cannot be applied to any of these columns. If humans want to preserve wild species then that is a reflection of human values, not nonhuman intrinsic values. My previous comment on the term 'life frames' applies here also.	We have responded to this comment by addressing the two issues that the reviewer raises: a) Categories are not 100%, and therefore what is their utility: We thank the reviewer for this comment regarding the utility/limitations of labels. Indeed, there was much discussion regarding the fact that values of both individuals and groups/cultures cannot be boxed into static categories. For that reason, this values typology explicitly is not a list of 'the values of nature', as some have sought or expected. Rather the typology highlights the key meanings and types of values that have been distinguished across disciplines and policy realms. In this way, scholars or practitioners are better equipped to 'navigate' the diversity of what value means in different contexts. However, the reviewer is also correct to note that categories are analytic devices and not monolithic. Instead, they are dynamic. To re-enforce this fact, we note the following: i) Visually, Figure 1 intentionally includes a gradient between worldviews in terms of colours to show that they are not separate silos. ii) Plus, the columns are separated with dashed lines, rather than solid ones. iii) Finally, the 'spotlights' of each life frame intentionally overlap, as they are not mutually exclusive, as noted in the caption. To further attend this concern, we have reinforced with text in various parts of section 1 the fact that no categories are static and indeed part of the typology's utility is to 'navigate' across the typology's value layers and types. Also see response to Comment #11. In conclusion, these labels were not invented for this paper, but have been synthesised from academic and policy sources to contribute to the real world situations where nature's multiple values must be confronted in complex decision-making processes. For example, the life frames were developed based on an extensive cross-disciplinary systematic review of the environmental values literature and testing with an ethnographic local knowledge dataset. The four life frames were distinctly recognisable in the literature and were able to encompass the vast majority (>90%) of framings found in these sources. At the same time, they are not equitably represented in publications or policies (see also Fig. 2), which has clear implications for promoting more plural values in environmental management and decision-making. Therefore, while one may (and does) hold multiple frames, it is important to consider that they
----	---------	---	---

			are not being reflected as such in extant research or decisions, which allows their better incorporation in the future based on this analytical tool. b) Considerations of intrinsic values in the Values Assessment: Please see response the Comment #9 on intrinsic values.
14	402-408	The ms. states: "However, some values are incommensurable, i.e. they are neither comparable nor compatible with other values. For example, while development projects are associated with instrumental values (e.g. economic and health benefits), they may also affect relational values (e.g. loss of sense of place). Even though these values can be incommensurable, decisions can consider them in parallel, such as through deliberation with affected parties or granting autonomous decision-making to Indigenous peoples and local communities (IPLCs) within their territories to maintain their own mix of instrumental, intrinsic and relational values^{49–51}." The problem of comparability is one I fully acknowledge; measurement is challenging. However, the above text also states that some values are not 'compatible' with other values. While this term is not defined it suggests that there are some values which, in principle (as opposed to in practice), cannot be traded off with others. Now sustainability requires that we maintain certain stocks of natural assets above given levels, below which vital earth processes are threatened. But that is quite different from a value which is truly incompatible with trade-offs. Consider even the atmospheric services of a climate operating within planetary boundaries. This might not be the situation we are in now, but if we really did reverse global heating then we would again allow humanity the safe operating space to allow for trade-offs between other sources of wellbeing and greenhouse gas emissions. So even the atmosphere is perfectly compatible with other values once we are within that safe operating space – its simply that we are at present a long way from that position. There is a real danger in declaring that multiple different	We thank the reviewer for these very insightful comments that draw out several points. We outline our response to each below. First, we recognize that we do not define 'compatible', but instead give a description in section 1 of what we mean by saying "specific values can be compatible when they share common features like being measured in spatial units, allowing them to be addressed together despite using different indicators (e.g., spatially overlaying bundles of the wetland's contributions to people proxied with biophysical, monetary and socio-cultural indicators)." This also includes cases where different types of values can be included in the same theoretical framework, such as a choice experiment or a multi-criteria analysis, even if the individual components are not directly comparable. We believe that the example given on spatial alignment of very different aspects is a good example of how valuation procedures can bring multiple values of nature into a common analysis, which is useful for decision-making and does not require them being placed into the same value indicators or units. The second points relate to incommensurable values where the reviewer raises some important points. We agree that including incommensurable values in decision-making in parallel through deliberative processes does not guarantee that all affected parties find the decisions reached reasonable or even that decisions can be reached. Furthermore, we agree that values that are incommensurable with instrumental economic values are not necessarily more important and should not trump other values just because they are hard to measure. We also recognize that giving autonomy to Indigenous peoples and local communities does not necessarily make incommensurable values more comparable or compatible. To better clarify, we suggest deleting here the sentence that relates to property rights for Indigenous peoples and local communities as this question is now addressed in other parts of the paper where we touch upon property rights (i.e.. section 5). We, however, also suggest keeping the reference to deliberation as a useful process when diverse worldviews and value

		values are incompatible with other values. We will then have a profusion of special cases. This leads to two practical problems: First, every special case trumps all other value assessments. Because it contains values that, according to the above definition, are not compatible with any other values then there can be no trade-off with any other value. So that case HAS to be funded under these rules. This of course will lead to an explosion of such special cases and any assessment which does not have one will find that all available resources have already been allocated elsewhere. Second, the above situation would of course quickly prove untenable and the ‘special case’ rule would be ignored. The result is that we are back at square one – except that we have forfeited the credibility of assessments. This is not a reasonable approach for real world decision making. The appeal to “...deliberation with affected parties or granting autonomous decision-making to Indigenous peoples and local communities (IPLCs) within their territories to maintain their own mix of instrumental, intrinsic and relational values^{49–51}” is merely a call to respect property rights (which I fully endorse; see previous comments to this effect) dressed up as values assessment. It is not and should not be presented as such.	perspectives play a central part of the issues at stake in a decision-making initiative, as this is substantiated in the valuation literature. We have rewritten the last paragraph of section 1 to accommodate this comment about the challenge of incommensurability. See also response to comment #28.
15	428-441	The ms. states: These methods can be organized into four crossdisciplinary ‘method families’, based on their source of value information: 1) nature-based valuation gathers information about the importance of nature and NCP 65,66 through direct and indirect observation of nature (e.g. spatial mapping of ecosystem services⁶⁷), 2) statement based valuation obtains information from people’s expressions of their values (e.g. stated preference surveys⁶⁸; or deliberative processes⁶⁹), 3) behaviour-based valuation identifies how people value nature by observing what they do in relation to nature (e.g. hedonic pricing ^{70,71}; or livelihood dependence⁷²), and 4) integrated	We realise that there are several well-established classification logics, especially from environmental valuation in economics, that are useful for highlighting strengths and limitations of different approaches. Furthermore, introducing new ways of classifying has the risk of confusing policymakers and researchers alike. We do not reject the standard classification in economic valuation; it has been very helpful for decades, as a means to further develop valuation and improve valuation practice. Therefore, we do not make suggestions for changes to this classification. However, the classification used in environmental economics suggested by the reviewer matches economic theory and economic value definitions, and the Values Assessment is broader. To assess the strengths and weaknesses of diverse methods to inform decision-making on values of nature and human-nature

		valuation brings together different types of values assessed with different information sources (e.g. participatory rural appraisal⁷³; integrated modeling⁷⁴). And subsequently: IPLC valuation methods There is a very well established set of terms for valuation methods, why has it been rejected? The existing typology is:  • Market based • Non-market: Revealed preference • Non-market: Stated preference The paper rejects this terminology. Instead it provides its own typology which seems very likely to cause confusion. Terms like “integrated valuation” suggest that all other types of valuation are not integrated. Furthermore, the definition of mapping as ‘nature based valuation’ is simply misleading; mapping is not valuation. This seems like unnecessary invention of new terms for their own sake. Decision makers are confused enough without further loading such as this.	relationships, we have chosen to use a more interdisciplinary framework (i.e. beyond economics). We also argue that the ‘method family’ classification is useful as a tool to highlight strengths and weaknesses of diverse methods. Having said this, we want to reiterate that many of the lessons from economic valuation are mirrored in the method family classification, but our classification also introduces other strengths and weaknesses than those revealed from the environmental economics literature. In particular, the ecosystem service mapping (as highlighted by the reviewer), is not a valuation method in an economic sense. However, by picking which services to map, and which service indicators to use, the mapping experts essentially become the valuers. This means that mapping is not value-free. We find that this is an important point to make. Mapping is not the provision of an objective truth and decision-makers should not treat it as such. The reviewer also argues that we are saying that integrated valuation is the only form of valuation that is ‘integrated’, whereas they argue that most valuation is integrated to some extent. The way the Values Assessment defines integrated valuation methods is as those that explicitly aim to integrate. This is a useful distinction because it allows us to highlight the challenges involved in aggregation (i.e. how to compare values, how to aggregate values, how to decide what a good outcome is, and how to identify it). This is critical in valuation and the choice of valuation methods. The method families have, therefore, been a useful tool to enable the authors to bring their different expertise together in the assessment. We have now further emphasised that the methods family classification does in some ways expand on the traditional valuation method classification in economics (first para. of section 2).
16	441-447 and 462-467	The discussion of how to incorporate the values of Indigenous peoples and local communities within assessments is interesting and topical. However, the link to property rights challenges needs to be made. There is little point in incorporating IPLC values into assessments if those will be ignored on the ground because of property right absences or violations	Thank you again for this point. We have now mentioned the role of territorial property rights of IPLCs in the new section 5.

17	468-471	Valuations carried out for conventional business sector decision making are ignored here. While I understand that, as they are likely to be the majority of all assessments then the authors need to note this demarcation of their analysis. More complex though will be the large and growing number of assessments carried out by the private sector for mixed purposes of both regular investment appraisals and analysis of the environmental and social impacts of decisions. These appear to have been omitted both from this text and Figure 2 and it is therefore somewhat unclear what the criteria were for inclusion within this analysis (see next comment).	We agree with the reviewer. We have clarified in the text referring to Figure 2 that the analysis concerns ‘peer-reviewed’ publications (fourth para. section 2). Our review does not include publications not referenced in Web of Science. Further clarification of the scope of the literature is provided in response to Comment #18 below.
18	468-471 and 479-488 and Supplementary Info	The text reports that 48,781 studies were reviewed. While this at first seems quite remarkable, a review of the Supplementary Information (SI) document shows that this is the number of papers that were delivered by a web browser search of terms principally on four databases: Web of Science; Scopus; Google Scholar; and EBSCOhost (Academic Search Premier). This very large number of studies was then reduced to some 1163 studies based on a series of rules set out in the SI. It is not possible to assess the defensibility of the precise rules used for inclusion of papers as applied across the 12 documents contained in the SI. For example, in Information Document 3 papers are accepted into the review according to an unspecified criterion based on the paper citation score normalised for publication date. These rules vary across the individual Information Documents depending on the ‘Specific topics supported’ and ‘Type of review’ criteria.	The selection of the 48,781 papers is described in information document 12. They are identified through keyword searches in Web of Science. The details on the search terms and the web search to generate the database can be found in 10.5281/zenodo.6468906 and 10.5281/zenodo.6468906 (specified in footnote 17). This search identified 79,040 papers. The selection of the 48,781 papers from the keyword search is based on geographical mapping of the papers to reveal where the studies have been conducted (i.e. in which socio-political-ecological context). This has been done through country codes. The rationale for mapping the studies like this is that mapping based on author affiliations on institutional affiliation would be misleading with regards to where valuation studies are being undertaken.
19	479-488	The importance of understanding and validating the criterion used for selecting studies is crucial to the interpretation of the results presented in Figure 2. Figure 2. Global distribution and characterization of nature valuation studies reported in the literature. If the selection of studies	This is a good point, and we find it a good idea to make it more obvious what information the reader can draw from the map. See suggested edits below.

	is defensible then the world map of studies presented here is both expected and interesting as it provides a quantification of the focus on Europe within valuation studies. Similarly, the categorisation of studies by habitat type was an interesting finding. A comparison of this to the physical distribution of land use and globally would be interesting as it will reveal that the marine environment is massively under-investigated. The authors might repeat this just for terrestrial land use types and studies. Doing this with or without Antarctica will reveal that we ignore the latter as well. A comparison of remaining land use distribution with study habitat might then show the focus decisions of researchers. The very clear dominance of ‘Nature-based valuation’ studies in the selected literature was of some concern. Given that this category includes mapping studies which, I would argue, are not valuation assessments then the fact that these represent 68% of the selected studies suggests that this decision by the authors has had a huge impact upon analyses. A similar concern arises regarding the dominance of biophysical measures within the Value Indicator assessment.	Antarctica is shown as ‘no data’ to make clear from the map that Antarctica is underrepresented, compared to the acreage coverage. An additional analysis would not generate additional information. We agree that the doughnuts in the figure show the choices of researchers, not necessarily which habitats are most abundant, which places on the globe are in most need of valuation studies, or what impacts are most important. This is precisely the contribution of the map. It provides food for thought about what evidence research is contributing to decision-making on values of nature. It is also correct that the choice of including nature-based valuation in the review has had a significant impact on the selected papers. This was a choice by the authors based on the scoping document of the IPBES Values Assessment. We would argue that it was a good choice. Nature-based valuation is de facto a valuation method. Research on decision support for area protection is based on mapping of, for example, red listed species without assessment of the impact on people. The reviewer might not find this appropriate, and the authors agree, but including nature-based valuation helps to highlight what values currently form the basis for decisions on the management of nature and whose values are being included. One of the limitations of nature-based valuation is (as argued in the Values Assessment), that it is largely based on expert assessment of what is important and largely fails to include values of those living close to the areas in question, depending on the natural resources from the area or in other ways value nature protection. The Values Assessment argues that behaviour-based or statement-based valuations are needed to understand such impacts. Therefore, the dominance of nature-based valuation and biophysical measures is illustrating the point that most of the research on how to inform and enable decision-making on nature are neither economic valuation nor assessments of socio-cultural influences of decisions on nature. We find that this is important to highlight. Excluding nature-based valuation would portray a misleading picture of the literature on the methods used to directly or indirectly represent values of nature in decisions. Lastly, as for the reviewer’s comment on marine environments, we now include the point in section 2 (fifth para.) that just about 10% of valuation
--	--	---

			studies are conducted in marine environments (coastal and deep sea) even though oceans cover >70% of the planet's surface.
20	490-501	I found Figure 3 both interesting and useful. However, its relation to the studies assessed is not made obvious, indeed this looks rather stand-alone.	We agree with the reviewer here as well. Figure 3 visually summarises a recommendation to explicitly design valuation studies for specific purposes and opportunities in project/policy cycles. This springs from the review of the peer-reviewed valuation literature, and the finding that most valuation studies make only cursory reference to how valuation information has been or could be used. We have clarified this in the text by stating that "...to increase the likelihood of uptake across the range of approaches and contexts, valuation can be adapted and timed to suit policymaking needs regarding particular purposes and decision-support opportunities". We have also moved Figure 3 to the end of section 2.
21	503-504	The ms. states: "the literature suggests that the vast majority of valuation studies do not engage with relevant stakeholders ⁶⁴ " This is a very sweeping statement to make from just a single reference (a further review by the same authors which may be the source document for much of this paper). If this is true, and it is likely to be contended, then it is very likely to also apply to non-valuation studies which in other respects adopt similar survey or experimental methods – which would suggest that labelling this as a problem of valuation research is misplaced.	We agree with the reviewer that this statement can be misunderstood. We have identified different types of stakeholder engagement, from no mentioning of stakeholders to engagement of stakeholders in each step of the valuation process, and from scoping to final reporting and communication of results. While our systematic review has shown that less than 6% of valuation studies do engage with stakeholders in every step of the valuation process, it has equally shown that it is mainly approaches based on nature-based valuation methods that do not report on any interaction with stakeholders. We suggest making this clear by linking the statement to nature-based valuation and only report on the percentage of valuation studies that do not involve stakeholders (for example through interviews or surveys). Many valuation studies obviously do involve interactions with people that may be affected by e.g. the proposed policy proposal being studied. We suggest to change the sentence to "The evidence also suggests that the majority (62%) of valuation studies, especially nature-based valuations, do not involve stakeholders in the valuation" (para after Fig 2). We hope that this makes the statement clearer and more relevant. It is not possible from our review of peer reviewed papers to evaluate whether the stakeholders involved in a study are the relevant stakeholders, and we have removed the reference to "relevant stakeholders".

22	505-506	The ms. states: “during the last three decades the share of peer-reviewed studies documenting uptake has not increased” It is not clear what the phrase “documenting uptake” means.	Documented uptake refers to the article specifying the stakeholder commissioning and/or their use of the study, and the purpose for which it is used. "documented" does not therefore correspond necessarily to "actual" uptake, because it may happen after publication, go unobserved by the researcher, or not be the primary reporting concern of the researcher to peers.
23	526-528	The ms. states: “the power of stakeholders with more resources can hinder the representation of diverse values in decisions” I strongly agree with this statement – but note that it comes after a long section advocating greater use of stakeholder perspectives. The solution to this conundrum is not obvious here.	The statement is included to highlight that there are no silver bullets in the choice of valuation methods. The Values Assessment has reviewed the participation literature which highlights that participation processes need to be conscious of power dynamics. The review of how valuation studies consider participation (including power dynamics) has been designed based on Bryson et al (2013), now included in the reference list. An example of how this can be done in practice is shifting from formal public hearings, which tend to be dominated by a small number of individuals comfortable with that format, to one-on-one interactions or smaller group discussions. The review shows that very few valuation studies explicitly consider power in their design choices. In addition, we now include a sentence making the point that only 12 % of the reviewed valuation studies explicitly consider design choices to improve inclusion of stakeholders including efforts to reduce marginalization (previous to last para in section 2). This relates to Information Document #10 in the supplementary information (section B).
24	532-533	The ms. states: “Considering locally held or place-based values, for instance through meaningful community involvement, can lead to more equitable and sustainable outcomes” The difference between “can” and “does” is crucial here and relates back to the previous issue. There is a strong assertion here that ‘local’ and ‘stakeholder’ will enhance equity and sustainability. Of course this can be the case – but the opposite can also hold. Local decision making can lead to domination by local power-bases. In such cases, more remote decision making might well prove more impartial. The local good equation offered here is too simple. What are the designs and criteria needed to deliver more equitable and sustainable outcomes? The inference	It is an excellent point that not all "local" values are compatible with stewardship, and we have adjusted the text at the beginning of section 3 to reflect that. However, there is substantial evidence (that we provide subsequently in section 3) that disregarding local input leads to disenfranchisement and can undermine program goals.

		that local stakeholders will deliver such results is not a sufficient argument on its own. This criticism applies to most of this paragraph.	
25	531-567	While I accept that the points made in Section 3 are referenced I found it very difficult to judge the weight of evidence. The section makes a long series of statements: that local is better than national, stakeholder better than population, indigenous better than other, power is in league with development, etc. This may all be true but I could not judge the strength of evidence regarding these statements.	The reviewer makes a good point that we had previously reported only on the findings rather putting in context how the evidence was assembled. We have now rewritten the text to acknowledge where the evidence was more quantitative vs. qualitative, and the number of studies supporting these assertions.
26	570-579	This is a nice summary of the IPBES report produced by the authors.	Thank you.
27	580-587	Is this figure from the IPBES report? I think it is and it should be clearly marked as the source.	The leverage points figure is central in the new section 5 (merger of original sections 4 and 6). Its purpose is to show that actions need to be undertaken to achieve a values-centered transformative change approach. The new section 5 provides examples of each leverage point. The figure helps identify those leverage points and their position with regard to being a ‘shallower’ (easier to achieve but limited impact) or ‘deeper’ (harder to achieve but larger impact). See also response to Comment #28.
28	570-587 and 618-623	The framework presented here is nice and clear and the continuum developed through the IPBES report is useful. There is a fundamental difference between stages 1-3 and stage 4 and I feel that needs highlighting. Stages 1-3 are different levels of recognising the world as it is and incorporating the values it generates within decision making. However, stage 4 is very different; it is an attempt to change those values. There is an existing literature on this (e.g. the Arrow et al ‘social norms’; paper) that needs acknowledging. More importantly I feel it would be useful to give the reader some idea as to how these changes are to be delivered. The GDP example is great – but a discussion of how ordinary peoples’ preferences and values can be influenced would be very helpful here. I would suggest a two pronged approach should be highlighted. First existing	We largely agree with the reviewer, and the paper now benefits from acknowledging the difference between leverage points 1-3 and 4. This is specified whilst also adding the nuance that institutional and policy change often serves a normative function (Chapin III et al 2022 reference now included). This is an important contribution to the ‘how’ part of the comment in the revised section 5. Finally, we have amended the Fig 4 description to clarify the key message that transformative change requires interventions across multiple leverage points and also (in light of the normative function of policies/institutions) that alignment (joined up) of multiple interventions is possible. We have extensively rewritten the new section 5 to deal with other points made by the reviewer in this overall comment, including issues of (property) rights, power and dominance. We also agree that Arrow has made multiple contributions to how values of nature can be included in decision making. The manuscript already

		preferences can be used to modify behaviour – for example via carbon taxes on food, fuel, etc. This I feel fits in with earlier stages in the continuum but clarifies the difference with the second approach. The second approach focuses on the modification and outright change of preferences. Again you can highlight extant literature here – general preferences regarding climate change and biodiversity loss have been the focus of preference altering information for years; some of this has been high profile (for example the impact of the David Attenborough Blue Planet programmes on plastics pollution). Value change can be very effective – e.g. the social norms regarding smoking indoors and drunk driving have changed radically. However, with all of this there should be acknowledgement that there is no single golden bullet and all of these levers across the full continuum need to be applied to deliver a sustainable world. As a final aside, which you may or may not wish to use; note that in principle there is no difference between trying to manipulate values to move in a pro-sustainability direction than the manipulation of preferences to deliver outcomes which we now consider repugnant. The Nazi regime realised the power of such manipulation and used it with great effectiveness to change preferences in disgusting ways. This raises issues of morality and power which I feel would be an interesting and honest insight. Who decides on the goals of such exercises? What is the moral basis of determining that a “just and sustainable future” should be that goal? What is the balance of risks associated with developing mechanisms to alter mass-preferences?	acknowledged the contribution of Arrow on the measurement of sustainability as proposed in the Dasgupta Review. We have kept the point about the need for adjusting GDP and agree that this is a good example of how nature’s values can be included in decision making. We have also now included a reference to social choice research (Arrow, 2012) at the end of section 1. This new text acknowledges the literature on social choice pioneered by Arrow’s seminal work from 1951 on the challenges involved in making social choices in a democracy. We do not have the space to unpack these challenges in the present paper, but agree that the literature should be acknowledged, as it now is.
29	626-659	I will flag up to the Editor that I am not a fan of the vast majority of scenario analyses and that she/he should take that into account and might wish to dismiss this point as reflecting my own opinions and not the majority of the literature.	We appreciate the reviewer’s perspective about scenario analysis. The text has been adapted to acknowledge the nature of scenario documents, as highlighted by the reviewer. The intention of engaging across a range of literature sources has also been clarified in section 4 with the sentence

		I feel that most scenario analyses are of extremely limited value and some are simply misleading. The major failing that nearly all of them have in common is the absence of a supporting analysis to assess, in quantified terms, the trade-offs associated with moving between scenarios. In my experience the typical scenario study (including some very high profile cases) are basically exercises in policy persuasion dressed up as analysis. They contain a scenario where unbridled expansion of industry and land use intensification is contrasted with a Business as Usual 'baseline' and a couple of pro-nature alternatives. The conclusion is inevitable: pro-nature is best for nature and delivers sustainability and may improve distribution. I feel such exercises are simply unscientific. That does not mean they do not have a policy message – there are alternatives to the status quo. That's a useful story and could well have real world impact. But this is typically not an academically sound undertaking. Unless scenario analyses are backed by rigorous analysis of the trade-offs each scenario entails then they are policy briefs, not academic research. As I say, the Editor might decide this should be ignored on the grounds that scenario analyses are prevalent and (as I acknowledge) may have policy impact. But I feel that most of them should be confined to the realms of political science.	“Scenario planning integrates thinking across multiple disciplines and can reveal important insights about values integration into policymaking”.
30	667	Who are these “local people”? Is everyone in the world a ‘local person’?	We thank the reviewer for pointing out that our language was too vague here; indeed, everyone is ‘local’ to somewhere. The point we were making here, however, was that people who are locally impacted by conservation decisions (because of their proximity to and even dependence on these areas) often are disregarded in these decisions, and the values of people external to these areas (who live far away and are not materially impacted by restriction of access to these areas) are often privileged. We have now adjusted the text to make this distinction clearer.

31	671-674	The mention of rights is important and highlighted in my earlier comments. However, the authors here are referring to moral rights rather than property rights – and very frequently in the world the latter trumps the former, indeed this is the root of many of the challenges facing indigenous people. Their property rights are often not recognised in law and/or respected in practice. I feel the authors needed to acknowledge this and highlight how important this issue is throughout the paper. Put simply, it often doesn't matter what indigenous values are, or whether or not they are incorporated in assessments, if property rights problems mean that in practice those values are not respected.	Thank you. The intention of the original formulation was to refer to property rights - note the formulation 'formal recognition of rights'. As noted above, the old section 6 is completely rewritten (now section 5), and we refer explicitly to property rights when we talk about rights of Indigenous peoples and local communities.
32	683-696	The latter part of this paragraph begins to approach the property rights issue – but then shies away and retreats into a discussion of citizens assemblies. These will remain ineffective if property rights are not well defined and respected in practice. Given that a lot of this paper seems motivated by a desire to see indigenous preferences (and 'citizens' values – but see my comment about 'local' – everyone is a 'local' 'citizen' so your terminology includes the whole world) I feel the omission of a serious discussion of property rights running throughout the paper is a problem.	The whole section has been restructured and rewritten; hence this sequence appears differently now.
33	722-760	Please see my comments on the SI – above	We thank the reviewer for this comment. Indeed, in general, we were not clear enough on what was the total magnitude of the evidence analysed. We have now rephrased the main text to make this clear and have also added a new section at the beginning of the supplementary materials to provide further details. Such a clarification applies to all reviews, not just the one referenced here.
34	827	Ref: 16. IPBES. Methodological assessment of the diverse values and valuation of nature of the Intergovernmental Science-Policy Platform on Biodiversity and Ecosystem Services. P. Balvanera, U. Pascual, M. Christie, B. Baptiste, D. González-Jiménez (eds.). IPBES secretariat, Bonn,	The full report will appear in this link once its editing is finalised. In the meantime, we have requested the IPBES Secretariat add the links to all the chapters and Summary for Policy Makers within this web entry.

		Germany (2022). https://doi.org/10.5281/zenodo.6522522 . This is an important element of the review as it should provide details of the methodology. The link leads to the appropriate IPBES report however the file itself when downloaded is blank.	
Reviewer 2			
35	General	Summary of the key results: The article summarizes some of the key results of the recently published IPBES Methodological assessment regarding the diverse conceptualization of multiple values of nature and its benefits, including biodiversity and ecosystem functions and services. It leverages the idea of values of nature to a language understandable and accessible to all – academics, policy makers, activists. Based on the analysis of more than 50,000 selected publications and thanks to the work of more than 200 experts, the Value Assessment represents a key milestone to influence political and economic decisions to address the current environmental crisis.:	Thank you.
36	General	The study provides clear and straight-forward information on the topic and proposes a shared language and classification to deal with the heterogeneity of the values of nature, of the methodologies to study them, and of the way people's lives, decisions, principles, and behaviors are influenced by nature's values.	Thank you.
37	General	Moreover, the paper shows the current gaps in the valuation processes and methodologies, and their struggles to influence decision making. In particular, the clarifications and typologies proposed help identify which values (and hence stakeholders' worldviews) are excluded from decision making and underline the need to use different methodologies and indicators to fully take in consideration diverse and sometimes incommensurable values to promote change at different "levels" of the typology (worldviews,	Thank you.

		board values, specific values). It also discloses the lack of uptake documentation in peer reviewed literature and the need for more engagement with relevant stakeholders.	
38	General	The paper also interestingly proposes a classification of the different value-assessment methodologies (counted to be more than 50 types) in 4 families distinguished on the basis of the source of value information (nature-based – the most commonly applied –, statement-based, behavior-based, integrated). It also provides comprehensive information of the different types of goals of valuation studies, ranging from improving the state of nature, and enhancing people’s quality of life, to generating more socially just outcomes.	Thank you.
39	General	Importantly, the paper stresses the need to better assess and integrate in decision-making IPLC’s values because they are often guided by principles that promote positive interactions with nature. More broadly, the paper underlines the need to better integrate value assessments in decision making practice through the increasing of diversity, giving more attention to non-anthropocentric worldviews and non-monetary indicators, and paying more attention to power asymmetries. It is interesting to see how the paper explains which are the main barriers to valuation uptake in policy making – so giving inputs on how such limits may be overcome – and describes the 4 leverage points for transformative change (recognizing the values of nature; embedding valuation into decision-making; reforming policies; shifting underlying societal goals and norms). Furthermore, the paper shows how real transformative change can only be undertaken if marked-based instrumental values stop being at the center of political and economic decisions and other types of value – relational, intrinsic and combinations thereof – are incorporated in decision making and promoted among people.	Thank you.

40	General	Last but not least, the infographic that is used is astonishing. It incorporates a great amount of information while remaining very clear and direct and – also – beautiful.	Thank you.
41	General	Originality and significance: The conclusions – based on an incredibly large amount of literature – are novel and unique. No such comprehensive study has ever been done – to my knowledge – on the topic of nature’s value assessment and their relationship with decision making. The paper leverages to policy makers very useful, clear and important information that may truly help better integrate nature’s values assessments into decision making procedures. The paper is also interesting for academics studying IPLC TK and worldviews, Payments for Ecosystem Services, Nature Contribution to People, rights-based and community-based biodiversity management and protection, as well as other alternative nature-based models that aim at incorporating people’s values, interests and needs in nature protection and conservation.	Thank you.
42	General	Data & methodology: The data used is very large and was selected through on attentive and well explained (in the SI) methodology. The methodology for analysis is also very robust and well presented in the methodology section which explains in detail how the 29 reviews were prepared, and the methodology used by the involved experts.	Thank you.
43	General	SI: the authors have included the doi of each of the 29 protocols produced by IPBES. Adding the title of each protocol could be useful to gain, at first sight, a clearer picture of their structure, content and purpose (for example, the fact that many of them are named after the Chapter of the IPBES Assessment they were used for is a potentially useful information).	Thank you for the suggestion. We have now done so.
44	General	The paper is also very well written with simple but precise language.	Thank you.

45	General	Conclusions: They appear to me to be very robust, clearly explained and relying on a very large body of literature as well as on the expertise of more than 200 experts	Thank you.
46	General	References: does this manuscript reference previous literature appropriately? Yes, it does	Thank you.
47	General	Clarity and context: Is the abstract clear, accessible? Are abstract, introduction and conclusions appropriate? Yes, they are.	Thank you.
48	351	Unsure of why the authors used the term “similarly”, as I do not understand what it refers to.	Edited.
49	455	Unsure of how more resources may increase the relevance of a valuation method to a real word decision making procedure.	Relevance means that the valuation methods can provide information about the values that matter to people. The use of ‘benefit transfer’ methods can be suitable in resource-poor situations, but is rarely the most robust method and can also exclude values that are relevant to stakeholders. This implies that when more resources are available other methods (that are more resource intensive) that can be better targeted to the local context become more advantageous.
50	503	“the vast majority of valuation studies DOES not engage with...”	Edited.
51	577	Unsure why the authors refer to fixing market and institutional failures. Given the scope of the other leverage points identified, reference to markets seems too specific. This broader scope is reflected in Figure 4 that refers to “reform policies, rights and regulations”. I would delete reference to markets and add reference to rights.	Thank you. The entire section is rewritten, and the reference to markets is deleted
52	Section 4	Earth stewardship pathways are considered by the authors as relying on the perception of the relational value of nature. While it is surely more symmetric to state so (considering that Green economy is presented as relying on instrumental value, Nature protection on intrinsic value, and Degrowth on the 3 together), Earth stewardship pathways –	We agree with this point, and the use of ‘prioritise’ was intended to communicate that other types of specific value are not completely absent. Similarly, we would say that the nature protection pathway from a multiple-values perspective is not only about intrinsic values, but also acknowledges instrumental values, such as through attention to ecosystem services. We have amended these accordingly. We have also made a minor change to the

		that I interpret as those closer to so called Earth Jurisprudence – also strongly rely on the recognition of intrinsic value. (On Earth Jurisprudence see, among the other: BURDON P.D. (ed.), Exploring Wild Law. The Philosophy of Earth Jurisprudence, Wakefield Press; CULLINAN C. 2002. Wild Law, Siber Ink; BURDON P.D. 2015. Earth Jurisprudence: Private Property and the Environment, Routledge; BERRY T. 2006. Evening Thoughts: Reflecting on Earth as Sacred Community, in TUCKER M.E. (ed.), Sierra Club Books.)	pathways Figure so that the Earth Stewardship pathway does not arise directly from the ‘relational’ sphere but from a point close to relational but also in between relational and intrinsic.
Reviewer 3			
53	General	I am an evidence synthesist (especially focusing on quantitative syntheses) and ecologist, and I am certainly not an expert in valuation studies. However, I should be able to provide my view on the article as a general reader as well as a synthesis expert. I note that this review seems to be a summary of the longer report, “The assessment report on the diverse values and valuation of nature” by IPBES. And this report is based on the results of many systematic maps (scoping reviews).	We thank the reviewer for pointing out that some of the description of evidence was not accessible to non-social science audiences. We have reframed the methods section to emphasise that to represent the full breadth of the diverse values of nature, our evidence gathering needed to take many different forms to encompass the different knowledge systems and ways of knowing about these values. Quantitative evidence in this review is highlighted where it was collected, but many qualitative methods exist for gathering evidence that are equally important (though less well known or understood by many natural scientists). Part of our goal with this paper is to present these many forms of evidence and diverse ways of knowing alongside each other, which is something that needs to be done equitably when the values of nature are considered in decisions. To respond to these concerns, we have redrafted the methods section in the main text to allow readers from different disciplines to better understand our approach. Also, we have expanded quite significantly on this issue in a new introduction of the supplementary information (section A).
54	General	One: in terms of synthesis methods, it is well-organized and well-done. The method is very transparent, and related documents are archived online, which is excellent. I make two points about the method. I did not see any mention of limitations of their method, which is usually reported for this kind of systematic mapping. It does not need to be part	We thank the reviewer for this important observation. We have now added a sentence on these limitations to the main text and a whole paragraph with an introductory section to the supplementary materials (Section A) in response to this and other excellent comments from reviewers

		of the main text (or some important limitations can be), but I certainly want to see some discussions on limitations.	
55	General	Two: One of the important items for systematic maps is stakeholder engagement (e.g. ROSES – Haddaway et al. 2018). But I did not see any of this, which is surprising as such stakeholder engagements would have been conducted. Haddaway NR, Macura B, Whaley P, Pullin AS. ROSES RepOrting standards for Systematic Evidence Syntheses: pro forma, flow-diagram and descriptive summary of the plan and conduct of environmental systematic reviews and systematic maps. Environmental Evidence. 2018 Dec;7(1):1-8.	We fully understand the reviewer’s concerns and agree that the text was not clear enough on this issue. We have now edited the methods section and added several paragraphs to the new section at the beginning of the supplementary material on this aspect. The IPBES assessment process involves relevant stakeholders at all stages, including the identification of the need for a specific assessment by the member states who commission the report; the reviews to the scoping document conducted by experts and approved by IPBES members states to define the questions and bounds of assessment; two rounds of open, external reviews by a wide range of stakeholders; three Indigenous and local knowledge workshops; workshops with delegates from the different IPBES countries; and a final round of review by governments. All the inputs from these processes are considered in the design and development of the assessment itself and of the design and operationalization of the review processes in particular.
56	General	Three: I got the main message of this article “we historically focused too much on single values e.g., economic values. We now need to embrace and practice diverse types of valuations”, which is great, but it is not news to anybody. Before reading this article, I was hoping to see more quantitative evidence of such. Although this is claimed to be evidence-based, readers may not be quite sure what is evidence-based. It felt like authors’ opinions on the trends they observed in the literature.	We acknowledge the concerns of the reviewer. We have now further clarified, both in the main text and in a new introductory section to the supplementary materials (see Section A) how we conceptualised ‘evidence’ and analysed it in different ways in response to the specific topics and data sources. The previous version did not define clearly enough what types of evidence were considered. We now show that sources from many different disciplines and knowledge types were considered. We also show how different criteria of what is robust and legitimate data/information and what are the different analytical approaches to address the evidence were used. Having clarified this, we have screened the paper for opportunities to better highlight some of the quantitative evidence (not dismissing the qualitative one).
57	General	Four: This is related to my point 3. Although they seem to have some descriptive statistics from the 50,000 documents they reviewed for the study, their narrative does not include statistics. I note that Fig 2 has some stats, but these are not	Thank you. Please see responses to above comments on this.

		well integrated into the text; I also note “5%” is mentioned in the abstract). I suggest they use more precise language quoting % in the main text.	
58	General	Five: The repeated and excessive use of acronyms makes it impossible for me to read this article without going back and forth (e.g. NCP, IPLC, LK, PES). And this has prevented me from understanding many seemingly good points the authors are making.	We have reduced the use of acronyms to the minimum and kept them to only those that are widely known (e.g., IPBES, UN, CBD) and those that significantly reduce text, e.g. NCP, which is also already well known in the literature associated with IPBES.
59	General	Six: Relating to the point above, The MS used a lot of jargon and introduced a lot of concepts without explaining what they meant. A glossary may be helpful. Due to this and the excessive use of acronyms, the MS reads like impenetrable government reports or social science articles (therefore, I did not find it exciting or informative, at least in the current form, although this is an important study). I consider myself having a good understanding of general things, so for me to find it difficult to understand is a bad sign. Thus, it requires some re-writing and editing to make it a lot more accessible.	Thank you for raising this point. We have tried our best to reduce jargon. We appreciate that this is of course subjective, as readers from different disciplines or backgrounds will find some concepts as jargon, while others will find them useful or familiar. Given the intended interdisciplinary nature of the paper, we have tried our best to make the text accessible to a wide readership.
60	General	Seven: I was wondering why I found this article so difficult to read and understand apart from the points already mentioned. It is probably due to the complete lack of concrete examples (there are no concrete stories people can relate to, even though they must have so many examples). Yes, Fig 1 tries to do this to a certain extent, but a more concrete example would have done better (e.g., a specific river and real people there; I am not saying Fig 1 is bad, but I need some accompanying story to make it relatable). Also, concrete examples related to Fig 3 and 4 will make these concepts more tangible. Also, some recommendations and future directions are suggested, but again without seeing some success stories, it is impossible to see how these things would play out.	Throughout the paper, we have reinforced the link to specific examples, when possible, and also other ‘intermediate’ elements of practical action (e.g., highlighting that behaviour change can be affected by targeting motivation, capacity and ability, or noting that power dynamics involve discursive and structural dynamics). In this way, we seek to make the evidence ‘tangible,’ while still being broadly applicable. Regarding the specific examples, in particular we have now included the example of a relatively successful restoration experience of a globally-important wetland in India (the Chilika Lagoon). This case allows us to illustrate several key issues that are part of Figures 1, 3 and 4. The case is described in a new section C of the Supplementary Information and referred to in various sections, starting in section 1 and then echoed in subsequent sections. We hope that these approaches, including retaking the wetland example in section 5, provides a more tangible application of these issues and we

			hope that this also creates a more coherent and pleasant narrative throughline for the reader.
61	General	Eight: I was also surprised not to see any concrete suggestions for what different parties can do to increase uptake of different valuations in the literature, policy documents and practices (e.g. academics, NGO, politicians, local communities etc). In some sense, they have done this, but it is hard to see it in the current form.	A number of examples regarding uptake are available and show how different barriers have been overcome. These are case- and site-specific. However, the manuscript length does not permit delving into detailed examples. Figure 3, therefore, contains a 'high level' recommendation to purpose valuation to the specific needs of stakeholders at different points in the policy cycle. In addition, the new section 5 (merging previous sections 4 and 6) is more action-oriented with examples on how valuation can contribute to transformative change at different levels.
Reviewer 4			
62	General	A. "Diverse values of nature underpin just and sustainable futures" is a welcome review of and addition to the literature on sustainability because it attempts to deal with two relative neglected and complex realms that underlie human-environmental relations at the national and international levels: human values and knowledge diversity. The key results, based on a review of more than 50,000 scientific publications, policy documents and Indigenous and local knowledge sources by experts involved in the Intergovernmental Science-Policy Platform on Biodiversity and Ecosystem Services (IPBES) Values Assessment, include a basic typology and assessment of nature values and valuation methods and a framework for understanding and applying them within a more rigorous and inclusive sustainability science. The article also offers a (perhaps too-veiled) critique of national and UN Sustainable Development Goals (SDGs), which despite a broad nation-state consensus, "still prioritize a narrow subset of nature's values, ignoring many of the ways people interact with, care about, and benefit from nature." Although not a new critique, the case for what is lost by homogenizing nature values—especially as assets and services to humans—is well developed and compelling.	Thank you.

63	General	B. The paper is original and significant in that it provides a novel framework for conceptualizing and integrating values and valuation into nature assessments such as the IPBES. It posits that there is a “values crises...at the core of the intertwined crises of biodiversity loss and climate change, pandemic emergence, cultural erosion, and social and political polarization, and social-environmental injustice.” Therefore, values cannot be neglected or taken for granted (e.g., as homogenous, universal, or inherent in markets). The typology of values is also an original and useful heuristic but some categories/distinctions, such as “sociocultural” under “values indicators” beg for further subdivision or specification, as they cover just about everything (arguably even monetary values are sociocultural). Contrastingly, the term worldview, borrowed from anthropology, is a catchall term that seems over-essentialized when subdivided. In other words, cultures that are “cosmocentric” can still be anthropocentric in relation to the cosmos or aspects of it, at least at certain times (indeed this may unavoidable), or be transactional in “living with” nature. While there is value in these distinctions, perhaps, as ideal types (a la Weber), it important to caution the reader that they may not represent any particular culture or group but rather are points along a continuum.	We thank the reviewer for these important insights. The concepts used in the values typology were defined after a careful review of the literature from various disciplinary fields. Despite being a useful categorization of different concepts to identify and include multiple values of nature, we also recognize their complexity and overlaps. The text was revised to reflect these overlaps. Also, symbolically this is represented, for example, in Figure 1’s light beams to help the reader to understand that emphasis can be given to certain worldviews, values and indicators in valuation processes, but that these also interact and overlap.
64	General	C. The data and methodology are clear enough and follow an established standard for review or systematic review. The >50,000 sources consulted is impressive but how such things as coding the results actually worked, and to what extent it may have involved those with diverse knowledge systems, could have been further developed. The paper is largely conceptual synthesis based the existing literature (published and gray, including ILK of the environment), stressing not just major themes, but key gaps in the literature and how these gaps may be undermining efforts	We thank the reviewer for pointing this out. We have now produced a new section at the beginning of the supplementary materials (Section A) to further expand on the analytical procedures. We have also edited the methods section in the main text to better clarify protocols, but could not expand further on this due to word count limitations.

		such as those of the IPBES. This approach is laudable and the quality of presentation is generally excellent, especially as concerns the general critique and (counter) proposition and the figures used to support it. The underlying data is there in the copious sources cited, but the analytical procedures themselves could be slightly more elaborated in the body of the paper or notes.	
65	General	D. As noted above, the reader may wonder about coding procedures and whether statistical analysis of such things as how many cultural groups are classified as possessing a particular worldview in the values typology would be useful (if only to understand the likelihood of getting to a cosmocentric worldview). Nevertheless, the descriptive statistical analyses presented, particularly on what valuation methods toward nature have been used and where, were compelling—and, again, nicely supported with clear, nuanced figures.	Our literature review methods and analytical procedures are described in detail in the Methods section and supplementary material. The subset of papers retrieved for the Indigenous and local knowledge literature review was handled by various groups of experts, who were familiar with the concepts and codes used. A description of each code accompanied a coding sheet, in which coders could use more than one code, adding the specific part of the paper under review which represented that code (e.g., a statement about the interconnected nature of Indigenous spiritual connections with the land and the cosmos). A smaller group reviewed all the codes to verify alignment or discrepancies. Although there is always some degree of subjectivity in qualitative analysis, the results represent trends in the literature regarding the diverse worldviews and values represented in the academic literature portraying Indigenous and local knowledge tied to values of nature.
66	General	E. The conclusions are generally robust, though validity and reliability in some cases seem may rest on the replicability of the methods used to classify the literature. A “values turn” in sustainability and conservation science is overdue and critical to success in organizing the diversity of cultural knowledge systems and environmental values to support the diversity and sustainability of various types of environments and their constituent species. A weakness of the paper is that it does not say much about the causality (or cultural models) leading from values and knowledge to action (pro or anti-environmental in character), though the findings and conclusions do underscore key correlations such as that “Considering locally held or place-based	Thank you. We agree with the reviewer, but we refrained from evoking a particular theory of causality. There are many psychological theories that posit the connections between values and behaviour and also ‘gaps’ that complicate a single, determinative mechanistic explanation of this link. A full treatment of this topic requires more space and nuance, which can currently be found in the assessment itself. For the purposes of relating this topic to the other issues treated in this manuscript, we have clarified the prevalence of theories that highlight values-behaviour linkage but that also identify important mediating contextual factors (the key reference we use here is Michie’s synthesis in the ‘Behaviour Change Wheel’. We use an example of transport in which acting on a personal value to select a greener transport option is dependent on physical infrastructure that enables one to

		values, for instance through meaningful community involvement, can lead to more equitable and sustainable outcomes” as well as effective ways of framing engagement and communication with constituent groups about environmental sustainability/conservation aims and objectives. At the bottom of all this, too, lies power and dominance, and whose cultural model of the environment gets operationalized when competing models (and/or the worldviews that underlie them) may be incommensurate.	act on values. And we have proceeded to be more explicit about how value-action gaps can be addressed.
67	General	F. This last point comprises my main suggestion for revision, which is to consider causality beyond worldview a little more carefully. The cultural models framework may be useful in this respect because most of the world's biodiversity and cultural diversity is now subsumed under nation-states and their aggregates (the UN, the EU, etc.) which seem to be, with a few exceptions perhaps, driven by cultural model of sustainable development (epitomized by the SDGs) stemming from a Western notion of ‘progress’ that, through globalization, has been projected onto other diverse cultures. Elsewhere, one finds very different imagined futures under that globalized projection. Yet cultural models, as shared mental models, do not essentialize the way worldviews do; rather they aggregate mental models which show variation, if not significant divergence, both within and across populations, and can be calibrated as such. This work is also relevant to communication, for as Kempton et al.’s (1995) seminal work Environmental Values in American Culture showed that the unity and diversity of values within nation-states can be important to understand when developing and communicating about environmental change and policy. In fact Americans share important biocentric values which can unite people behind sustainability values, while in other respects they are diverse (see also Thornton, T. F. et al. 2019. Cultural models of and for urban sustainability... Climatic	Thank you. The reviewer raises an important point that multiple levels or layers of value exist in society. We agree, but in the Values Assessment, we did not distinguish between individualistic and hierarchical cultural values, as implied in this comment. Those differences have indeed been shown elsewhere (e.g., van Riper et al. 2019). Cultural models, including hierarchical models driven by Western paradigms of progress, are inferred in our discussion on institutions and power relations in the revised section 5 of this paper. Given the word limitations for this paper, we have decided not to explicitly treat this point in the revised version of the paper. Reference: Van Riper, C et al. 2019 Integrating multi-level values and pro-environmental behavior in a U.S. protected area, Sustainability Science, 14

		Change (https://doi.org/10.1007/s10584-019-02518-2) for a more recent review.	
68	General	G. Beyond those works mentioned above, there are also useful works that critique Western/“WEIRD” (Western Educated Industrialized Rich Democratic) models of ecosystem services (as in the Millennium Ecosystem Assessment) by interpreting/critiquing them through other cultural values systems (e.g., Comberti, et al. 2015. Ecosystem services or services to ecosystems? Valuing cultivation and reciprocal relationships between humans and ecosystems. Global Environmental Change, 34, pp.247-262, (https://doi.org/10.1016/j.gloenvcha.2015.07.007)). Other useful work on biocentric and biophilia values also exists (e.g., Kellert and Wilson’s 1993 The Biophilia Hypothesis) and seems worth referencing, especially as it considers the unity and diversity of environmental perception and values in ways not considered in this article. In addition, while the broadly anthropological (and philosophical) perspective on worldview and cultural-environmental values is welcome, including a few seminal references (e.g., Ingold 2000), there is much more that has been done in this field and environmental social science more generally, which might be constructively incorporated, including a number of recent works on Indigenous Knowledge Systems with lots of case examples and references (e.g., Thornton and Bhagwat, eds., Routledge Handbook on Indigenous Environmental Knowledge, 2020). Many of these show how different values systems lead to different human-environmental perspectives, relations, and outcomes.	We thank the reviewer for this comment. The literature review conducted for the Values Assessment included Kellert and Wilson’s widely cited work on Biophilia, as well as submissions by Indigenous peoples and local communities, regarding the diversity of worldviews and values of nature, and how these diversity drives relationships, perspectives and outcomes for both people and nature, as this reviewer noted. A more in-depth review and analysis of this relevant literature is presented in Chapter 2 of the IPBES Values Assessment, cited as Anderson et al. (2022), and it is beyond the scope of this article to include such treatment here. Our article presents, in a concise way, a synthesis of concepts, methods, and insights to advance the consideration and incorporation of the multiple values of nature in decision and policy-making.
69	General	H. Overall the paper is clearly written, contextualized (although could be broader and deeper in places as suggested in specific comments above), and lucid. The figures enhance the argument and presentation in	Thank you. We believe we have strengthened the paper throughout as a result of addressing the reviewer’s comments. To further balance this conclusion, we added additional sentences towards the end to keep the

		illuminating and useful ways. In addition to the above suggested revisions, I would like to see a slightly stronger conclusion, too, going beyond recognizing the diversity of environmental values and valuations, and prescribing how best to organize and balance the diversity in a way that supports both sustainable biodiversity and cultural diversity at present or restorative levels. That's a tall order given most nation-states' power and continuing preferences for growth and development in unsustainable ways, and requires much more social science and political work around values and diversity that could flow from this investigation.	spirit largely unchanged, but allowing us to better connect the evidence we have found with the description in the text.
Reviewer 5			
70	General	This paper applies a values framework to map out what the global sustainable futures might look like if a diverse range of values was considered in potential pathways towards sustainability. The paper has arisen from the work carried out under the aegis of IPBES and meets the criteria of rigour and global significance. As a review, it is not particularly novel, but the heuristic framework proposed is likely to generate interest and provoke conversations about sustainability pathways. The only area where the paper perhaps falls short is the 'how' question. The paper talks about what needs to be done and this is well argued and justified, but what steps need to be taken to get there in the eight years to 2030 could be covered more directly to trigger change in policy and practice	Thank you for the constructive comment. On the 'how' issue, we believe that providing step-by-step guidance on ways to trigger change in policy and practice exceeds the scope of this paper (and the Values Assessment). However, in this revised version of the manuscript, we strove to make the 'how' more tangible. First, we merged the previous sections 4 and 6 into a new section 5 that mostly focuses on the opportunities and challenges identified in the Values Assessment regarding 'how' to catalyse the four values-centred leverage points of transformative change. We have also provided new real world examples on 'how' such changes, including shallower and deeper leverage points, may be activated. Also, given the request to provide a specific example, we have included the illustrative example from the Chilika Lagoon in India across the paper (and section C of the supplementary information). All in all, we have edited the new version to more explicitly engage with the 'how' question, space permitting.
Reviewer 6			
71	General	The authors present a summary of the findings of the IPBES Values Assessment report. These results suggest	Thank you.

		(paraphrasing) that more diversity in valuation approaches, and a better incorporation of this diversity of approaches into policymaking, can leverage transformative change towards more just and sustainable planetary futures. I enjoyed reading the paper as it was well-written, engaging, and thought-provoking, and no doubt summarizing a massive volume of work in such a short space was an incredible challenge. The fact this is a summary of a pre-existing report does make it a difficult piece to review, since there isn't really the potential for making suggestions that could lead to additional data collection, analyses, etc. As such I focus my comments on how existing information could be better presented to make the piece more compelling. These detailed suggestions are below, but they boil down to two main areas:	
72	General	1. Empirical evidence for value systems. The authors suggest that there are a diverse range of values for nature found around the world, but that only a small subset of these values is typically represented in decision making and communication (paraphrasing: instrumental values, anthropocentric worldviews predominate). The authors suggest that this is due to asymmetries in power dynamics, and suggest or imply that many (most?) people hold other latent, pro-environmental values that are waiting to be unleashed. But no empirical evidence – at least here – is presented to justify this take? I completely agree that power dynamics are present and no doubt skewing policies and decision-making in this domain as they do in all others. But an alternative hypothesis is simply that the vast majority of humanity holds anthropocentric, instrumentalist values for	Thank you for this intriguing comment. We would like to point out that there is substantial evidence that in many communities worldwide, sustainability-aligned values are strongly prevalent. In the well-established World Values Survey, wave 7, comprising 64 surveys conducted between 2017-2022, it was, for example, found that post-industrial societies with high levels of security and autonomy prioritise self-expression values, including environmental protection and participation in decision-making in economic and political life (Haerpfer et al. 2022 - now cited in the paper). The relationship between individual and collective sustainability-aligned values and pro-environmental attitudes and behaviour also is well-established (Dietz et al, 2005; Marshall et al., 2019; Wang et al. 2021). Of course, it is not possible to map out all the existing values both in the Global North and Global South, but we believe there is sufficient evidence to suggest that pro-environmental values exist, being prevalent and expressed around the world (e.g., in multitude socio-environmental

		nature. And that the other worldviews/values are held by a tiny minority of people, meaning there isn't a great, untapped pool of pro-environmental values just waiting to be unleashed. Do you have empirical evidence to bolster your view that globally it is the former rather than the latter that explains the predominance of the anthropocentric/utilitarian worldview in decision-making?	conflicts). See also the Atlas of Environmental Justice (https://ejatlas.org) as evidence of residence by many local communities in the face of imposition of values or when values conflict between different stakeholders. In the revised paper, we had difficulty finding a place to integrate arguments on this increasing pool of environmental protection values in post-industrial countries. References: Dietz, T., et al. 2005. Environmental values 30, 335–372. Haerpfer, C., et al. (eds.). 2022. World Values Survey: Round Seven - Country-Pooled Datafile Version 5.0. Madrid, Spain & Vienna, Austria: JD Systems Institute & WWSA Secretariat. doi:10.14281/18241.20 Marshall NA, et al. (2019) Our Environmental Value Orientations Influence How We Respond to Climate Change. Front Psychol. 2019 Jun 18;10:938. doi: 10.3389/fpsyg.2019.00938. Wang, X., et al 2021. I am vs. we are: How biospheric values and environmental identity of individuals and groups can influence pro-environmental behaviour. Frontiers in Psychology 12.
73	General	2. Even if we take at face value the supposition that it is societal power asymmetries and flawed institutions that are excluding the full diversity of nature's values from decision making, and that correcting this can unleash transformational change, the question remains: how exactly can this be done? I understand and appreciate the leverage points that are described, but I believe that a fair reading of the text suggests very little specific actions or pathways are given that could provide a blueprint for how to achieve this. E.g., how exactly can the balance of institutions be changed? How can the influence of powerful, often malign stakeholders be blunted when 'transformational change' is	Please see also our response to Comments #4, 5, 7, 31, 32, 33 and 70. In short, we provide guidance regarding the 'how' when we delineate aspects of motivation, capacity and ability (i.e. the 'behaviour wheel' from psychology). Furthermore, also in consonance with responses to Reviewer 1, we highlight the role of property rights and rights-based approaches, which could partially attend this issue of political economy. Plus, the new section 5 is dedicated to explaining 'how' to engage the four values-centred leverage points identified by the assessment. While we cannot do an extensive treatment of the how in a general review of how values relate broadly to decision-making, we hope that these approaches have sufficiently addressed the reviewer's legitimate concerns to ensure this work contributes to existing disciplinary approaches and provides concrete

		not in their interest? See additional points in specific comments below. There are some key threads from political economy and behavioural psychology that could perhaps be drawn into this discussion, but as it stands, much of the 'how' is left undescribed. And yet this would be a very useful contribution of this piece: to dig into the 'how' in much more detail.	or tangible paths forward (see also responses regarding better case studies and illustrations of concepts throughout the manuscript).
74	425	Define 'valuation' here for readers.	We have now included the definition used in the Values Assessment at the start of section 2.
75	468	Perhaps I am misunderstanding what you mean, but I would have thought the motivation for valuation in many/most cases would have been academic exploration for knowledge generation and improved understanding of a particular system? Rather than having some desired end goal in sight?	Our review also identified studies with 'no reference to uptake', which are classified as 'explorative' in purpose. The proportion of papers documenting uptake is about the same as 'explorative'. Most papers actually make at least a cursory reference to uptake (Figure 4.12 in chapter 4 of the Values Assessment - Barton et al. 2022), showing that most papers at least mention possible applications, while a small minority actually document how valuation was used. We have, therefore, not changed the text since this is indeed an important finding.
76	477	Per above, is this simply because the large majority of the world holds these framings?	See response to Comments #72 and 73
77	503-504	Understandable, as this isn't typically the point of a valuation study itself; rather as you say above, studies can be harnessed by decision-makers for policy purposes.	Noted.
78	506	Documenting uptake of a particular valuation study is not necessarily the job of the scientists behind it, and in any case can only happen (much) after the original study is published...perhaps I am misunderstanding but suggests rewording/clarification is necessary.	Thank you for pointing this out. A valid reason for not expecting higher documented uptake than we observe (<5%) is that most actual uptake takes place after a study with an informative purpose is published. To observe uptake, valuation studies need to be iterated over the policy cycle. The policy cycle, as illustrated in Figure 3, draws attention to this recommendation.

79	516-518	Environmental economists will not be surprised at this result!	Noted. It is indeed expected that some findings will be surprises and others expected in an interdisciplinary synthesis prepared for a broad readership. In that sense, it is the overall assessment's output that needs to be evaluated on its rigour and relevance/novelty. This attempt to dialogue between disciplinary traditions entails both mutual surprises and comprehension, but what is sought is a new synthesis that provides greater understanding across and within traditions.
80	526-529	This starts to get at the crux of my point above, re: power dynamics / political economy.	Noted. See response to Comments #70 and 73.
81	545-549	Completely agree	Noted.
82	564-565	How does 'recognizing the diversity of values held across all actors through participatory assessments' actually address asymmetries in power dynamics? On the face of it, not at all. Please elaborate.	We agree that this sentence was confusing. We have rewritten this part of section 3. It is clear that recognition is a necessary, but insufficient first step. However, it is a step nonetheless.
83	589-590	What incentives do powerful decision makers have to 'acknowledge and respect' diverse values of nature when this may threaten existing power structures that favour them?	Thank you for pointing this out. This is a key question. We argue that it is the role of organised facets of society (e.g., via social movements, political processes) to catalyse transformative change by pressuring harmful actors (e.g., those with vested interests to maintain the status quo) or political systems (i.e. to adjust power relationships structurally via rule-making procedures, etc.). We have added new text on power to section 5 and to the conclusion. Given strict word limitations, we cannot go deeper into the topic of power, as it also would exceed the scope of this review. We are convinced, though, that the text on power is now stronger based on this constructive and useful comment.
84	610-613	Totally agree and this is the crux of the matter...but how exactly can this be done?	Please note that we have striven to address the 'how' issue more explicitly throughout the manuscript, especially by restructuring the paper (with this goal in mind). The original section 4 was merged with section 6 (now a new section 5), and consequently the previous text has largely been rewritten. The sentence referred to by this comment is no longer included in the paper.

85	615-617	And the balance of these two types of institutions is very clear given the current state of the world. How specifically can this balance be changed?	How to balance these types of institutions is a crucial question. This would in partly depend on the socio-political and historical context and the preferred political system at play (e.g., liberal democracy or other). This issue is incredibly important and germane, but outside the scope of the Values Assessment . Thus, we are unable to provide clear insights from our assessment's reviews presented in this paper. We hope, however, that these important points will be addressed in the ongoing IPBES Transformative Change Assessment . In our work, we provide some key insights as to the need for balance, but cannot go further without having carried out specific reviews on the topic.
86	618-623	Same question as above: how can these underlying goals and norms be changed? Particularly as they are deep and slow moving, which means they will take time to change...and time is of the essence.	We now provide some hints towards this aspect in the new merged section 5. However, a comprehensive treatment of this issue (e.g., including theories around activating positive social tipping interventions, Pascual et al. 2022) is out of the scope of the paper. Chapter 6 of the Values Assessment provides some food for thought in this regard, as well. We also point towards the reference Pascual et al (2022). which provides reflections from IPCC and IPBES about how certain behaviour, technology etc. can be positively triggered in non-linear ways, i.e. social tipping interventions.
87	665-667	Certainly not true over the past decade-plus of conservation. Take a look at any conservation organization and see how things are often framed around ecosystem services, local communities, indigenous peoples, etc. Or the way the post-2020 CBD agreement is wording things. Suggest this statement needs editing or indeed removal.	Point well taken. We have toned this statement (see first paragraph of the new section 5): "... conservation policies have frequently prioritised nature's intrinsic values, despite increasing advocacy regarding the instrumental and relational values held by those living within and around protected areas who rely on biodiversity for their livelihoods (Pascual et al 2021; Anderson et al 2022)." We have also changed the text in section 2 (right after figure 2): "...when countries monitor the values of biodiversity (Aichi Target 2 of the Convention on Biological Diversity), National Biodiversity Strategies and Action Plans generally use biophysical and to a lesser extent monetary indicators despite the general perception that policymaking favours economic approaches to the valuation of nature".

88	669-671	Yet surely this statement could be applied to any particular policy issue. I do not think you need a massive, multi-year values assessment by dozens of global experts to come to this conclusion; it's simply common sense (similarly to a point made above on how policies and interventions work better in areas when people who actually live in that area are engaged in their design). And yet despite this, most often this ideal is not achieved. Why not? It would be useful to draw lessons from other arenas and see what has worked in instances where this has in fact been achieved. If the authors have already done this, very useful to present here.	Thank you. We have striven to focus more on the 'how' issue. Given the enormity and the complexity of how to affect individual and collective decision-making changes, and given that other reviewers have also asked for more examples linked to the 'how' issue, we have restructured the original sections 4 and 6 into a new section 5. The specific text the reviewer notes is replaced and rewritten. It is now included in a specification of processes that could activate changes at the deepest level of transformative change - leverage point 4. Generally, the 'how' issue is now covered in a more developed way. However, the amount of material to be covered in the review and the word limitations of the journal format dictate the conditions regarding the depth we can go into this topic. Full justice to the issues surrounding the political economy (e.g., distribution of power, engagement of citizens, property rights, etc.) exceed what can be done in this paper, but we have now referenced these when possible and appropriate in a more explicit way.
----	---------	---	--

Reviewer Reports on the First Revision:

Referees' comments:

Referee #1 (Remarks to the Author):

Review of: Revised Nature manuscript 2022-07-11504B
Diverse values of nature underpin just and sustainable futures

General Comments

In my review of the initial manuscript one of my major general comments stated:

“The paper puts forward a series of frameworks for viewing the literature. However, that literature itself contains a prior series of frameworks and it is not obvious that those suggested here are clearly superior, nor that they would lead to an improvement in the incorporation of nature’s diverse values within decision making.”

This comment questions both the novel academic contribution of this paper and its contribution to improved decision making. The revised paper addresses many of the more detailed comments in my previous review, but not these central challenges.

Two further issues are outstanding. First, in my opinion, the response to my challenge on property rights is insufficient. I discuss this in detail below but, given the lack of response in this first revision I now strongly suggest you withdraw any suggestions that this framework is suitable for practical decision making. The failure to address this issue seriously is acceptable only if the authors acknowledge clearly that the contribution of the paper is in highlighting the diversity of issues regarding valuation, that the ms does not provide sufficient analysis for practical decision support, and that issues such as the incentivisation of property owners (either through economic instruments or adequately enforced legal frameworks) are crucial to real world decision making.

A second, equally important omission from this paper is its lack of integration through to the natural and physical sciences. There is very little here concerning the ways in which such science input should be articulated into the basis of any decision making concerning the natural environment. It is insufficient to point to ‘values’ without also understanding their quantification and connection through integration with the science underpinnings of understanding.

Values are constantly changing. In part this is a human phenomenon as populations, demographics, socio-economics and understanding change. But the environment is also changing, both from natural processes but now predominantly because of human induced change. The changing baseline for valuation is not considered here.

Even if changing baselines were assessed, the connections within and across the environment and economy and the trade-offs and co-benefits arising from new drivers of change are massively important in assessing the net effects of a given decision. Again similarly there is no consideration here of an approach to appraising multiple alternative uses of earth’s limited resources.

All of the above reservations refer back to the limitation of this paper as a guide to decision making; it simply is not. The authors should make this very clear from the outset and within each section.

Specific comments are as follows. Thank you for the opportunity to review your revised paper.

Line 293

This, the opening line of the revised ms., states that the paper is "Based on a review of >50,000 academic publications". However, the Supplementary Information shows that this impressive number is principally obtained by using keyword searches of the Web of Science. Such computerised keyword searches do not correspond to my conception of a 'review' of a paper – I think I am unlikely to be the only academic who feels this way so that such terminology may inadvertently mislead readers who do not study the extensive Supplementary Information so carefully. Therefore I feel some alternative, more transparent, terminology should be adopted. I suggest that the paper should open "Based on a keyword search of >50,000 academic publications".

The searches are conducted across 29 different topic areas, although the procedure used to choose these topics is not clear. The number of papers collected for each topic area varies very considerably. So, while the topic "Gaps and capacity needs to operationalize the diverse values of nature in decisions" identifies 5 papers, the topic "Valuation methods and approaches" includes some 48,781 papers. The highly unbalanced nature of such a dataset will yield biased results unless the consequent extreme variation in observation error across these topics is incorporated into the analysis (see de Leeuw, J. and Meijer, E. (2008) Handbook of Multilevel Analysis). It is not obvious from the paper that this has been incorporated into the analysis.

This issue resurfaces regarding lines 543-545

Line 313

The parenthesis opened here is never closed making the sense of the paragraph difficult to follow.

Line 353-356

In my review of the initial submission I raised objections to the way in which the authors use the term 'intrinsic value'. As I noted, this perpetuates the (admittedly widespread) misnomer that humans (i) understand and (ii) can articulate nature's intrinsic value.

By definition humans cannot understand or articulate this value precisely because it is intrinsic to nature such that we as humans can never know what that is. . It is definitionally impossible for humans to define intrinsic values. All we can ever articulate are human values for the environment.

So humans can and do decide that they want certain species to thrive while others should be reduced in number; or that a given environment should continue unchanged while others should be modified. But none of these are an intrinsic value for nature; they are expressions of human values – and human values change. What is now seen as pristine wetland was once viewed as an awful bog to be eradicated.

Just because this error is repeated frequently in the literature this does not mean it is correct. Aside from the fact that 'bringing nature's intrinsic value into decisions' mangles the dictionary and proposes an impossible action, such claims confer an entirely spurious moral superiority to such assessments, intended to trump any critique. If we abandon the scientific method in favour of unverifiable claims to being able to measure the unmeasurable (by definition I cannot measure a non-human value; nonhuman entities such as wild animals or even trees cannot articulate their values in ways we humans understand) then there is no rational basis left for decision making. The environment has to be brought into decision making if we are to avoid global collapse, but abandoning science is not the way to do that.

In my previous review I stated that "I would object to this paper being published without a very

clear statement that the true intrinsic value of nature is by definition unknowable. An honest line would be to accept that humans make the decisions which are dominating the planet and those same humans have very clear preferences for sustainability (and other objectives as well such as improving equity) which need to be respected in decision making – but we cannot know nature’s intrinsic value; we can only know humans’ value for nature.”

The authors’ response to this argument is to reject it. For this point alone I could not support publication – this has to stop somewhere and this seems as good a place as any. Come clean – humans cannot know the intrinsic value of nature – full stop. All they can know are their own values for different states of the world.

While the authors say some very conciliatory things in their response to me, their revision of the ms. is now simply confused as, in discussing the various layers of Figure 1, they state:

“The third layer’s specific values refer to judgments regarding the importance of nature and its contributions to people in ‘specific’ contexts. It is well established that nature’s specific values can be instrumental (i.e. means to a desired human end)¹⁵ or intrinsic (i.e. independent of humans as valuers)¹⁶.”

How can there be ‘judgements’ of values which are ‘independent of humans as valuers’? Who or what are making these judgements if they are not human? And if they are human then these values cannot, by definition, be intrinsic to nature. Instead they are human values for a state of nature.

I know the authors are well meaning and I entirely agree with their objectives of trying to improve the way in which we make policy so as to protect the environment. But making statements which can be so easily disproved is not the way to deliver that end.

Line 309-331

In my review of the initial submission I stated that

“little weight is placed on the importance of property ownership as a very significant barrier to such incorporation. If valuation does not result in real world change then it is of little practical use. We could greatly enhance the assessment of values and find that this has no impact upon the decision taken by government or the actions of businesses because of these property rights.”

In response the authors state:

“regarding property rights, this issue is part of a more general problem regarding the needed changes to goals, policies, and measures (now explicitly mentioned) at the outset of the manuscript (i.e. in the ‘bold’ paragraph). We include specific comments on rights and property rights, but have not modified our assessment to make this ‘the fundamental issue’, which one could interpret to be implied from this comment. To clarify, this is a paper on the fundamental role that a value focus could play in environmental scholarship and decision-making.”

I have read the bold paragraph three times but there is no mention of property rights or anything approximating to the fundamental problem that these cause for human relations with the natural environment.

The paper remains a pure academic piece concerning the variety of values relating to the environment but offers no insight into how this relates to real world decision making. At the risk of overstatement – it really doesn’t matter what the value of the environment is if property rights trump change in our relationship with nature. The basic cause of environmental degradation remains that those that hold property rights gain more from that degradation than they lose from

the destruction of the environment. This fundamental difference between private values and public values is paramount. The authors claim, in the final sentence above, that "this is a paper on the fundamental role that a value focus could play in environmental scholarship and decision-making." I do feel that they are right on the "environmental scholarship" point but I see little that the paper has to offer regarding "decision-making." This repeats a common lack of serious discussion of property rights and the fundamental difference in incentives between private resource owners and public beneficiaries of environmental enhancement. Such omission perpetuates an unwillingness to grapple with the realities of real world decisions which has characterised much of the conservation debate and blunted its effectiveness.

While addressing this problem would be preferable, to do this properly would require a major reorientation of the paper. Therefore the authors should very clearly note the nature and seriousness of the property rights issue and equally clearly state that they do not tackle this in this paper – but that it must be addressed if the values considered in the ms are to be brought in to real world decision making. The paper looks at one aspect of the problem of the relationship between people and the environment. That is acceptable but only if the authors very clearly state that the issue of private ownership and the differences in incentives and objectives between private resource owners and the public has to also be considered if we are to bring those values into decision making

Line 459-460

The ms. states:

"The choice of a valuation method itself influences the information made available for decision-making"

True – but very often the reverse also holds; the choice of valuation method is determined by the information available

Line 488-492

Figure 2 remains a very interesting analysis which is likely to be cited

Line 496-497

The ms. states:

"The evidence also suggests that the majority (62%) of valuation studies, especially nature-based valuations, do not involve stakeholders in the valuation²²"

This statement cannot be correct other than by an unclear definition of the (dreadfully malleable) term 'stakeholders'. Values are held by humans who gain some benefit from the good or service in question. By definition all those who benefit are stakeholders, a fact which renders the term meaningless (if commonly perpetuated).

The arbitrary definition of stakeholders as those who have a 'more important' benefit from a good or service

The self-citation to the source underpinning this submission does not help here.

Line 543-545

The ms. states:

“This extensive evidence base includes a global meta-analysis of 171 peer-reviewed studies and a re-analysis of >8,000 assessments from >3,000 global protected areas”

The highly unbalanced nature of such a dataset will yield biased results unless the consequent extreme variation in observation error across these topics is incorporated into the analysis (see de Leeuw, J. and Meijer, E. (2008) Handbook of Multilevel Analysis). It is not obvious from the paper that this has been incorporated into the analysis.

Line 605-613

Aside from the fact that it precedes Figure 4, I have some reservations regarding Figure 5 (“The diverse values of nature underpin multiple pathways towards sustainability”).

First it places the concept of intrinsic value as central to this diagram – please see comments on lines 353 – 356.

Second the ‘degrowth’ concept emphasised here is very difficult to defend as a viable future pathway for real world policy. Even its supporters, such as Kallis (2011), Kallis et al., (2011) and van den Bergh and Kallis (2012) and D’Alisa and Kallis (2020), recognise that simple degrowth on a planet of 8 billion people would be catastrophic and argue instead for socially and ecologically sustainable degrowth.

References:

D’Alisa, G. and Kallis, G. (2020) Degrowth and the State, *Ecological Economics*, 169, 106486, <https://doi.org/10.1016/j.ecolecon.2019.106486>.

van den Bergh, J.C.J.M. and Kallis, G. (2012) Growth, A-Growth or Degrowth to Stay within Planetary Boundaries?, *Journal of Economic Issues*, 46:4, 909-920, DOI: 10.2753/JEI00213624460404

Kallis, G. (2011) In defence of degrowth, *Ecological Economics*, 70(5): 873-880, <https://doi.org/10.1016/j.ecolecon.2010.12.007>.

Kallis, G., Kerschner, C. and Martinez-Alier, J. (2012) The economics of degrowth, *Ecological Economics*, 84: 172-180, <https://doi.org/10.1016/j.ecolecon.2012.08.017>.

Line 619-620

The ms. states:

“conservation policies have frequently prioritised nature’s intrinsic values”

No, they have not. Conservation policies prioritise human values for nature. including human desires to conserve certain aspects of nature. For centuries people have waged war on those wild species and aspects of nature that humans do not like. No one seems to care about the ‘intrinsic value’ of the common pigeon. Why? Because (i) there is no way of knowing what that intrinsic value is and (ii) humans have decided that individuals from Species A (e.g. ospreys) are more important, more valuable, than individuals from Species B (pigeons). Those are human values and I very much doubt those individuals from Species B would agree with this prioritisation.

The prioritisation of a policy according to something that is by its very definition unknown and unknowable to humans is impossible.

The authors could provide a very considerable service to the literature by clearly standing out

against this nonsense.

Please see comments on lines 353 – 356.

Line 615-709

Section 5 “Transformative change involves leveraging nature’s values” is interesting but I feel unbalanced.

An initial, minor, writing point is that the text needs to mirror the cumulative addition of levers shown in the Figure as the move from Shallower to Deeper levers.

A more important point is the apparent assertion that those levers designated as ‘Deeper’ will indeed be more effective than those labelled as ‘Shallower’. While this almost has to be true if levers are indeed added cumulatively (see above), this does not have to be the case if they are independent alternatives. So one could have policy change independent of a social norm change. In such cases is it necessarily the case that the former must generate less impact than the latter? I’m not sure and would need to see more extensive evidence regarding this.

A further, more substantial point is that the Figure and text strongly suggest that the movement from Shallower to Deeper levers is always desirable. However this ignores the feasibility and costs associated with such movement. It might be that changing policy is relatively costless but generates major improvement but changing social norms might be challenging and costly. In such a case the intuition of the diagram breaks down and becomes misleading.

I think the authors need to rethink this section. Given that Nature Figures need to work as stand alone items, and often get reproduced separate from their wider papers I feel that merely revising the text is unlikely to be sufficient here.

Line 827

In my previous review I stated the link to the following reference leads to a blank document and that this was an important element of the review as it presumably provides details of the methodology.

16. IPBES. Methodological assessment of the diverse values and valuation of nature of the Intergovernmental Science-Policy Platform on Biodiversity and Ecosystem Services. P. Balvanera, U. Pascual, M. Christie, B. Baptiste, D. González-Jiménez (eds.). IPBES secretariat, Bonn, Germany (2022). <https://doi.org/10.5281/zenodo.6522522>.

In response the authors said:

“The full report will appear in this link once its editing is finalised. In the meantime, we have requested the IPBES Secretariat add the links to all the chapters and Summary for Policy Makers within this web entry.”

This means that I am still unable to review this document.

Referee #2 (Remarks to the Author):

The authors have addressed all the points I had raised in my first review in a satisfactory way. They have also reworked the article - according to the other reviewers comments - in a very positive way.

I hence suggest the publication of the article and I do not see the need for further revisions.

Giulia Sajeve

Referee #3 (Remarks to the Author):

I think the authors for addressing my comments. Now it is much easier to read this MS and it would be more relevant to many. I have no further comments.

Referee #4 (Remarks to the Author):

I am satisfied that the revised manuscript has met the main concerns addressed in my previous review, clarifying some key points in relation to methods for identifying values, use of broad terms like worldview and xxx-centric as ideal types, explication of coding and other analytical procedures to capture diversity, and, finally, developing a more substantive conclusion regarding how such analyses of values and value types can be used to improve conservation, restoration, and sustainability policies. Space constraints mean that some of this, especially the finer methodological points, is limited in the main text or incorporated into the supplementary material, which is fine. That issues of power and hierarchy remain even once diverse values and value types are "recognized" and "integrated" is at least acknowledged in the conclusion.

The figures have been improved and better integrated in response to reviews.

I recommend this essay for publication.

Referee #5 (Remarks to the Author):

Thank you for addressing the reviewers' comments thoroughly. The illustrative example of Chilika helps to apply the heuristic framework proposed in the paper and makes it more tangible.

Referee #6 (Remarks to the Author):

The responses to the broad and specific points raised in my review have mostly been acceptable.

However, I was disappointed by the authors' response to my first overarching point (that no empirical evidence on the relative prevalence of values for nature held by people has been presented) and indeed, do not think the way they have dealt with it is acceptable. This could easily be remedied by including some or all of the text in their response to my comment into the actual manuscript (note the authors claim Haerpfer et al 2022 is now cited but i couldn't see that this was the case, the citation appears missing).

The authors say "we had difficulty finding a place to integrate arguments on this increasing pool of environmental protection values in post-industrial countries", but this would fit very obviously in section 1 where the paper states that "the values of nature are diverse". A key measure of diversity is not only the *number* of categories/things, but the *relative abundance* of each of those categories/things. As it stands the authors have documented the first aspect of diversity in values for nature, but not the second. As such, it remains impossible for this paper as written to refute the alternative hypothesis raised in my review, which is that "the vast majority of humanity holds anthropocentric, instrumentalist values for nature. And that the other worldviews/values are held by a tiny minority of people, meaning there isn't a great, untapped pool of pro-environmental values just waiting to be unleashed."

Failing to address this weakens the paper and will leave it open to criticism that the authors are advocating for the existence of a diversity in values for nature for ideological reasons, rather than demonstrating empirically that this diversity in fact exists.

This shouldn't be a particularly difficult area of the paper to improve given that the authors state in their response to me that such empirical evidence exists. Surely the authors can do better than simply saying we couldn't find a place to address this comment?

Author Rebuttals to First Revision:

Diverse values of nature underpin just and sustainable futures Responses to second round of comments by referee #1 and referee #6

Referee #1

General Comments

In my review of the initial manuscript one of my major general comments stated: “The paper puts forward a series of frameworks for viewing the literature. However, that literature itself contains a prior series of frameworks and it is not obvious that those suggested here are clearly superior, nor that they would lead to an improvement in the incorporation of nature’s diverse values within decision making.” This comment questions both the novel academic contribution of this paper and its contribution to improved decision making. The revised paper addresses many of the more detailed comments in my previous review, but not these central challenges.

Thank you for offering your insights which we have taken seriously. As to the selection of topics, frameworks in our synthesis, we would like to stress that the paper synthesizes very different types of evidence. Our choice of what topics and frameworks have been reviewed would undoubtedly have certain biases. However, we believe our approach for balanced reviews across disciplinary perspectives is valid and we have provided information transparently about which topics, disciplinary approaches and other knowledge traditions we have used in generating our synthesis of the evidence. Please also see our response about the way we have conducted 29 different separate reviews below.

Two further issues are outstanding. First, in my opinion, the response to my challenge on property rights is insufficient. I discuss this in detail below but, given the lack of response in this first revision I now strongly suggest you withdraw any suggestions that this framework is suitable for practical decision making. The failure to address this issue seriously is acceptable only if the authors acknowledge clearly that the contribution of the paper is in highlighting the diversity of issues regarding valuation, that the ms does not provide sufficient analysis for practical decision support, and that issues such the incentivisation of property owners (either through economic instruments or adequately enforced legal frameworks) are crucial to real world decision making.

We thank the reviewer for this comment. We have taken this point seriously. We have striven to reflect the importance of property rights as part of the institutional structure guiding society and as crucial to leverage needed transformative change. We have no discrepancy in this regard with the reviewer, and in the text we now reinforce this point more explicitly to further highlight the important role of property rights (including the incentivisation of property owners), which are inherently connected to the dominant values in society. For this end we have:

- Changed the final line of the abstract to clarify that valuation per se confronts barriers to implementation that need to be addressed. Revised text: “The assessment concludes that combinations of values-centred approaches, both to improve the quality of valuation and to address the barriers to uptake, can leverage transformative changes towards more just and sustainable futures.”
- Included the following sentence in the opening of the paper via the bolded paragraph: “Yet, addressing the ongoing global biodiversity crisis³ still implies confronting substantial barriers to the effective incorporation of plural values of nature into decision-making. These barriers are tied to powerful vested interests that are supported by current social institutions (i.e. norms and legal rules), which determine, among other things, the property rights that determine whose values and which values of nature are acted upon. A better understanding of how and why nature is (under)valued in decision-making is more urgent than ever⁴.”

- Section 5 on the values-approach to transformative change now opens with a new paragraph: “Achieving more just and sustainable futures calls for reforming societal structures to address asymmetric power relations underpinning the allocation of property rights, including legal decisions about who holds rights to degrade or be protected from environmental harm and who/what is a subject of rights (e.g. a river, Mother Earth, etc). These reforms need to be complemented by the use of policy instruments to internalise negative environmental externalities that arise from the rift between private and public values, reducing overconsumption and overproduction, and by applying measures of progress that include social and ecological sustainability criteria⁴. Achieving these actions also implies confronting the contradictions evidenced by the historical and current prioritisation of a narrow suite of nature’s values.”
- We have rewritten the following sentence: “The third leverage point involves reconfiguration of societal structures, especially with regard to the decision-making architecture to normalise and scale-up the incorporation of diverse values in decisions.”
- The caption of figure 5 now reads: “*Transformative change is more likely when interventions engage multiple leverage points. The leverage points are interdependent, whereby jointly activating them entails addressing feedbacks among them, adding them up (moving left to right across the lever) or cascading down (moving right to left across the lever)*⁷.”
- Section 5 now closes with the following paragraph which has been rewritten to make the point about the interconnectedness of institutions (e.g. property rights and policy instruments) with values and valuation. We take a “cascading down the lever” direction pointing at the effect of changing social norms and goals on institutional change: “Transformative change is, thus, a multi-faceted process involving engagement of the four values-centred leverage points. Fortunately, opportunities for synergies arise, as leverage points are not static categories; instead, there are interdependencies along the lever’s action gradient. Leverage points may be activated in a cumulative way (from left to right across the lever), such as when a policy change (e.g. introducing a green tax) triggers a change in social norms over time (e.g. recycling). Values-centred leverage points can also be triggered in the opposite direction (i.e. cascading down the lever). For example, in Europe a deep leverage point involved a shift in vision about the role of agriculture, driven by the wider societal goal of sustainability and epitomised through a political agreement underpinning the Common Agricultural Policy. In early 1990s, this involved a change from supporting the agricultural sector to ensure self-sufficiency to recognising the need for mitigating the negative externalities harming wildlife and people’s health (i.e. a new social norm and goal). Since that time, a series of reforms and the associated political effort has increased the environmental components of the agricultural policy framework (i.e. third leverage point). First policy instruments and tools were implemented towards compliance with minimum environmental standards to justify income support to farmers. More recently the reform has introduced environmentally targeted payments for adopting sustainable agricultural practices (i.e. second leverage point)⁷⁶. The design of these policy instruments are being aided by the valuation of the externalities for which different methods and decision support tools are used (e.g. shadow pricing, choice experiments, and cost benefit analysis)^{77,78}. This example illustrates how shifting societal norms and goals can trigger the activation of all other values-centred leverage points. Clearly, power relations must be confronted, such as between citizens and agri-business, that ultimately influence whose and what values get priority in decisions.”

A second, equally important omission from this paper is its lack of integration through to the natural and physical sciences. There is very little here concerning the ways in which such science input should be articulated into the basis of any decision making concerning the natural environment. It is insufficient to point to ‘values’ without also understanding their quantification and connection through integration with the science underpinnings of understanding.

Values are constantly changing. In part this is a human phenomenon as populations, demographics, socio-economics and understanding change. But the environment is also changing, both from natural processes but now predominantly because of human induced change. The changing baseline for valuation is not

considered here. Even if changing baselines were assessed, the connections within and across the environment and economy and the trade-offs and co-benefits arising from new drivers of change are massively important in assessing the net effects of a given decision. Again similarly there is no consideration here of an approach to appraising multiple alternative uses of earth's limited resources. All of the above reservations refer back to the limitation of this paper as a guide to decision making; it simply is not. The authors should make this very clear from the outset and within each section.

Thank you. This is also an important comment, and it is particularly well taken. We also agree that the paper did not explicitly include the point that valuation ought to be able to prioritise how people utilise scarce resources provided by a nature that is changing as people make decisions on its use and management. This is an 'evolutionary angle' that we now refer to in the new version of the manuscript.

First, we agree that valuation should help us understand important ecological/biophysical limits to achieve more sustainable outcomes in the long run. Understanding values of a natural resource as a scarcity indicator can help decision makers protect natural capital stocks that are at the risk of being over-exploited. This is akin to the environmental economics perspective that see values and valuation mostly from a "living from" perspective (see typology of values). Further, as humanity needs to protect the ecological systems upon which it depends (we live from nature and need to be able to keep living from nature in future) the living with frame is also key in this interpretation of the role of valuation that focuses on the importance of ecological process.

We make these points now in Section 2 as follows:

- In the opening paragraph we state that "Valuation generates information about nature's values that can be used to make values visible to decision-makers, such as a scarcity indicator to protect natural assets that are at the risk of being over-exploited (i.e. *living from* nature). Valuation can also be used to inform about the need to protect the ecological systems upon which humanity depends (i.e. *living with* nature), and to recognize other ways humanity relates to nature (i.e. *living in* and *living as* nature)."
- We also clarify better the way we have assessed "integrated valuation" methods designed to integrate value information. We now explain more clearly that this category of valuation method involves two different goals and approaches: Integrated valuation covers both a) methodological approaches to link different types of values, and b) how values and nature/environment vary over space and change over time due to changes in practices and impacts in nature/environment. Our review indicates that such approaches have not been widely used for nature valuation. They are mostly developed and used to deal with agricultural externalities and ecosystem services mapping and valuation. But generally, an important point that we now include in the paper is that our review shows that valuation as a field is not very integrated yet (understood as a dynamic integration of peoples' values, interaction through behaviours and the impact on biophysical indicators and then in turn the impact on people's behaviour and well-being). We also now include a reference to a review paper (Chan and Satterfield, 2020) which has highlighted this point, showing that very few valuation studies include dynamics *sensu* changes over time in biophysical stock to address potential feedback on values. We have included the following new text "4) *integrated* valuation brings together different types of values assessed with diverse information sources (e.g. multicriteria decision aid)³⁶ and also seeks to understand how values, behaviour, and environmental outcomes interact in dynamic ways (e.g. integrated modelling)³⁷. While the valuation field has advanced substantially regarding the first three method families, it has not yet reached maturity regarding its integration potential to understand the dynamic interactions and feedbacks between peoples' values, behaviours, and the impact on biophysical indicators³⁸."

Specific comments

Line 293

This, the opening line of the revised ms., states that the paper is "Based on a review of >50,000 academic

publications”. However, the Supplementary Information shows that this impressive number is principally obtained by using keyword searches of the Web of Science. Such computerised keyword searches do not correspond to my conception of a ‘review’ of a paper – I think I am unlikely to be the only academic who feels this way so that such terminology may inadvertently mislead readers who do not study the extensive Supplementary Information so carefully. Therefore I feel some alternative, more transparent, terminology should be adopted. I suggest that the paper should open “Based on a keyword search of >50,000 academic publications”.

The searches are conducted across 29 different topic areas, although the procedure used to choose these topics is not clear. The number of papers collected for each topic area varies very considerably. So, while the topic “Gaps and capacity needs to operationalize the diverse values of nature in decisions” identifies 5 papers, the topic “Valuation methods and approaches” includes some 48,781 papers. The highly unbalanced nature of such a dataset will yield biased results unless the consequent extreme variation in observation error across these topics is incorporated into the analysis (see de Leeuw, J. and Meijer, E. (2008) Handbook of Multilevel Analysis). It is not obvious from the paper that this has been incorporated into the analysis. This issue resurfaces regarding lines 543-545

Each of the 29 data search strategies were designed separately to address the questions that were posed across the IPBES Values Assessment. The results of each of them are found in the different sections and subsections of the current paper. In the supplementary information we expand on what were the questions that triggered each of these protocols. For each of these protocols we carefully described the process to identify the evidence to be assessed in depth, as well as the process through which they were analysed. We thank you for the suggestion of the book on multilevel analysis. We would like to emphasise that the methods we used included qualitative methods in which the content of the theories, of policy documents or of case studies, for instance, was coded and synthesised using approaches like content analysis, discourse analysis, identification of relationships and similarities, among others. Sample size is relevant for quantitative analyses and the role of different sample sizes is critical when a single analytical procedure is used to synthesise across the different data sources. In our case each data search strategy responded to a different question and followed a different analytical protocol, based on the disciplines and types of knowledge involved.

We would like to highlight that the assessment of the robustness of the evidence and of the analyses undertaken in each of these 29 different strategies vary considerably among the different data sources, types of knowledge and data sources used in this assessment. We expand in the supplementary information on how this was taken into consideration. In short, we would like to emphasise that depth of the analysis and triangulation of results using different approaches is more important than sample size for most of our qualitative analyses.

We understand that in the editing process the language was not clear. Some of these details were already in the supplementary information but have moved some of this into the main document (methods section). We have now clarified the different ways in which the evidence was identified (more detail in the supplementary information). We have also clarified that once the data sources were identified they were analysed in depth following a wide range of quantitative and qualitative approaches. We have rewritten part of the methods section as follows: “The reviews encompassed multiple evidence sources identified using diverse strategies including keyword searches, and natural language processing of 48,781 peer-reviewed papers on nature valuation. The evidence reviewed in depth included 1,163 valuation studies, 1,270 study-site units reporting on values-based outcomes for 217 case studies, 838 documents from the ‘grey literature’ of environmental and development policy (e.g. reports from governmental, non-governmental organisations, and valuation initiatives), 26 specific contributions from Indigenous and local knowledge holders and experts, 460 futures scenarios, 37 policy instruments, 217 country-specific datasets (e.g. Aichi target 2 progress and UN System of Environmental-Economic Accounting - Ecosystem Accounting implementation), and 134 values-based behavioural theories (SI: B). This evidence was analysed in-depth following quantitative and/or qualitative approaches, which were supported by discipline-specific standards.”

Please note we have changed the text to “Based on >50,000 pieces of evidence from scientific publications, policy documents, and Indigenous and local knowledge sources” both in the abstract and bolded paragraph.

Line 313

The parenthesis opened here is never closed making the sense of the paragraph difficult to follow.
Corrected.

Line 353-356

In my review of the initial submission I raised objections to the way in which the authors use the term ‘intrinsic value’. As I noted, this perpetuates the (admittedly widespread) misnomer that humans (i) understand and (ii) can articulate nature’s intrinsic value.

By definition humans cannot understand or articulate this value precisely because it is intrinsic to nature such that we as humans can never know what that is. It is definitionally impossible for humans to define intrinsic values. All we can ever articulate are human values for the environment. So humans can and do decide that they want certain species to thrive while others should be reduced in number; or that a given environment should continue unchanged while others should be modified. But none of these are an intrinsic value for nature; they are expressions of human values – and human values change. What is now seen as pristine wetland was once viewed as an awful bog to be eradicated.

Just because this error is repeated frequently in the literature this does not mean it is correct. Aside from the fact that ‘bringing nature’s intrinsic value into decisions’ mangles the dictionary and proposes an impossible action, such claims confer an entirely spurious moral superiority to such assessments, intended to trump any critique. If we abandon the scientific method in favour of unverifiable claims to being able to measure the unmeasurable (by definition I cannot measure a non-human value; nonhuman entities such as wild animals or even trees cannot articulate their values in ways we humans understand) then there is no rational basis left for decision making. The environment has to be brought into decision making if we are to avoid global collapse, but abandoning science is not the way to do that.

In my previous review I stated that “I would object to this paper being published without a very clear statement that the true intrinsic value of nature is by definition unknowable. An honest line would be to accept that humans make the decisions which are dominating the planet and those same humans have very clear preferences for sustainability (and other objectives as well such as improving equity) which need to be respected in decision making – but we cannot know nature’s intrinsic value; we can only know humans’ value for nature.” The authors’ response to this argument is to reject it. For this point alone I could not support publication – this has to stop somewhere and this seems as good a place as any. Come clean – humans cannot know the intrinsic value of nature – full stop. All they can know are their own values for different states of the world.

While the authors say some very conciliatory things in their response to me, their revision of the ms. is now simply confused as, in discussing the various layers of Figure 1, they state: “The third layer’s specific values refer to judgments regarding the importance of nature and its contributions to people in ‘specific’ contexts. It is well established that nature’s specific values can be instrumental (i.e. means to a desired human end)¹⁵ or intrinsic (i.e. independent of humans as valuers)¹⁶.” How can there be ‘judgements’ of values which are ‘independent of humans as valuers’? Who or what are making these judgements if they are not human? And if they are human then these values cannot, by definition, be intrinsic to nature. Instead they are human values for a state of nature.

I know the authors are well meaning and I entirely agree with their objectives of trying to improve the way in which we make policy so as to protect the environment. But making statements which can be so easily disproved is not the way to deliver that end.

Thank you for the elaborated opinion on intrinsic values. To address Reviewer #1’s comment, we have now clarified the overarching definition of intrinsic values to highlight that when people express intrinsic

values of nature, this cannot be independent of the humans' valuing processes (a point that the reviewer strongly stresses and that we agree with). This clarification was needed and we thank the reviewer for pointing this out. During the editing process of the previous version of the manuscript the short definition, we had originally written in brackets, had lost further context and explanations that were needed and indeed, as the reviewer points out, created confusion. We now make it explicit that intrinsic value is expressed by people, but in a way that any reference to themselves (as valuers), e.g. to their own well-being is not part of the justification for explaining why nature's entities are important to them. The use of the term in the rest of the paper reflects this meaning. We have also clarified that the three specific value types refer to how people justify the importance of nature and NCP to themselves. The text now reads: "Specific values, the third layer of the typology, refer to how judgments regarding the importance of nature and its contributions to people are justified in 'specific' contexts. It is well established that nature's specific values can be instrumental (i.e. nature as a means to a desired human end)¹⁵ or intrinsic (i.e. value of nature considered and expressed by people as an end-in-itself without reference to their own well-being)¹⁶. The relational category captures how people express the importance of meaningful relationships between people and nature and among people through nature (e.g. reciprocity, care)¹⁷."

Because our paper is based on the assessment of the literature stemming from different disciplines, including environmental ethics, among others, the definitions we propose aim at including the variety of meanings extracted from that literature in the most comprehensive, inclusive, and respectful way possible. We acknowledge the relevance of the notion of intrinsic value in the environmental literature, and especially associated with a highly influential debate in biodiversity conservation discourse and practice (e.g. McCauley, 2006; Batavia and Nelson, 2017). Furthermore, based on a comprehensive review of over 230 articles about theories on nature's values, we cannot merely obviate that intrinsic values are part of environmental discourse, research, and policy. It is undeniable that this concept plays an important role not only in academic traditions related to conservation (e.g. conservation biology), but also in policy documents (e.g. 'intrinsic value of biodiversity' is the first value enunciated in the preamble to the Convention on Biological Diversity (UN, 1992)).

Moreover, the language of intrinsic value has been shown to be relevant for many Indigenous peoples and local communities and is often used as justification of legal regulations of, for example, the treatment of animals or the current debates around the rights of nature (e.g. legal cases from New Zealand or the constitution of Ecuador). Intrinsic values can be assessed in valuation (imperfectly though), for example, by using proxies such as biophysical indicators or preferences regarding "existence value", included in the total economic value (TEV) framework, that expresses, as the reviewer rightly points out, how humans value nature. It is also a type of value that plays an important role in leveraging people's pro-environmental motivations (e.g. Batavia and Nelson 2017).

We understand that the reviewer rejects the interpretation of intrinsic value as subjective (i.e. people are the valuers *and* they value something intrinsically, i.e. for its own sake or as an end in itself and not as a means to their ends, as in the case of friendship) (see e.g. Sandler 2012; O'Neill 1992) and appreciate this, but in our work we try to synthesise a large body of literature that also includes the subjective meaning of intrinsic value. We cannot in the paper discuss whether this understanding of intrinsic value is right or wrong, but can describe the extent to which it is important in academia and in policy to assess the diverse expression of nature's value. Such importance can be summarised pragmatically (e.g. because it can help understand and reduce the level of environmental conflicts), and normatively (e.g. because of issues of environmental and epistemic justice). Last but not least, there are also efforts to bridge between the instrumental and intrinsic value discourses as the basis for biodiversity conservation, which again shows the need not to exclude this topic, but integrate it (e.g. Reyers et al., 2012).

Batavia, C., & Nelson, M. P. (2017). For goodness sake! What is intrinsic value and why should we care?. *Biological Conservation*, 209, 366-376.

McCauley, D. J., 2006, "Selling out on nature", *Nature*, 443(7107): 27.

O'Neill, J. (1992). 'The Varieties of Intrinsic Value'. *The Monist* 75 (2), 119-137.

Reyers, B., S. Polasky, H. Tallis, H. A. Mooney and A. Larigauderie, 2012, "Finding Common Ground for Biodiversity and Ecosystem Services", *BioScience*, 62(5): 503-507.

Sandler, R. (2012) *Intrinsic Value, Ecology, and Conservation*. *Nature Education Knowledge* 3(10):4
<https://www.nature.com/scitable/knowledge/library/intrinsic-value-ecology-and-conservation-25815400/>

United Nations, 1992. 'Convention on Biological Diversity', (05/06/1992).
<https://www.cbd.int/doc/legal/cbd-en.pdf>

Line 309-331

In my review of the initial submission I stated that "little weight is placed on the importance of property ownership as a very significant barrier to such incorporation. If valuation does not result in real world change then it is of little practical use. We could greatly enhance the assessment of values and find that this has no impact upon the decision taken by government or the actions of businesses because of these property rights."

In response the authors state: "regarding property rights, this issue is part of a more general problem regarding the needed changes to goals, policies, and measures (now explicitly mentioned) at the outset of the manuscript (i.e. in the 'bold' paragraph). We include specific comments on rights and property rights, but have not modified our assessment to make this 'the fundamental issue', which one could interpret to be implied from this comment. To clarify, this is a paper on the fundamental role that a value focus could play in environmental scholarship and decision-making." I have read the bold paragraph three times but there is no mention of property rights or anything approximating to the fundamental problem that these cause for human relations with the natural environment.

The paper remains a pure academic piece concerning the variety of values relating to the environment but offers no insight into how this relates to real world decision making. At the risk of overstatement – it really doesn't matter what the value of the environment is if property rights trump change in our relationship with nature. The basic cause of environmental degradation remains that those that hold property rights gain more from that degradation than they lose from the destruction of the environment. This fundamental difference between private values and public values is paramount. The authors claim, in the final sentence above, that "this is a paper on the fundamental role that a value focus could play in environmental scholarship and decision-making." I do feel that they are right on the "environmental scholarship" point but I see little that the paper has to offer regarding "decision-making." This repeats a common lack of serious discussion of property rights and the fundamental difference in incentives between private resource owners and public beneficiaries of environmental enhancement. Such omission perpetuates an unwillingness to grapple with the realities of real world decisions which has characterised much of the conservation debate and blunted its effectiveness.

While addressing this problem would be preferable, to do this properly would require a major reorientation of the paper. Therefore the authors should very clearly note the nature and seriousness of the property rights issue and equally clearly state that they do not tackle this in this paper – but that it must be addressed if the values considered in the ms are to be brought in to real world decision making. The paper looks at one aspect of the problem of the relationship between people and the environment. That is acceptable but only if the authors very clearly state that the issue of private ownership and the differences in incentives and objectives between private resource owners and the public has to also be considered if we are to bring those values into decision making.

Please see our replies to the first general comment on how we have addressed the issue of property rights within the revised manuscript.

Line 459-460

The ms. states: “The choice of a valuation method itself influences the information made available for decision-making”. True – but very often the reverse also holds; the choice of valuation method is determined by the information available.

Good point. We have added the following sentence. “The availability of information influences the valuation method to be used. Likewise, the choice of a valuation method itself influences the information made available for decision-making...”

Line 488-492

Figure 2 remains a very interesting analysis which is likely to be cited.

Thank you.

Line 496-497

The ms. states: “The evidence also suggests that the majority (62%) of valuation studies, especially nature-based valuations, do not involve stakeholders in the valuation”

This statement cannot be correct other than by an unclear definition of the (dreadfully malleable) term ‘stakeholders’. Values are held by humans who gain some benefit from the good or service in question. By definition all those who benefit are stakeholders, a fact which renders the term meaningless (if commonly perpetuated). The arbitrary definition of stakeholders as those who have a ‘more important’ benefit from a good or service. The self-citation to the source underpinning this submission does not help here.

We agree with the point by the reviewer that stakeholders are those who gain some benefit from the good or service in question. The review points at the lack of participation in valuation in some way or another (e.g. by being the subjects of surveys) of stakeholders (besides the ones carrying out the valuation itself). We have adapted the text to “The evidence also suggests that the majority (62%) of valuation studies, especially nature-based valuations, do not involve stakeholder participation in the valuation²²”.

Line 543-545

The ms. states: “This extensive evidence base includes a global meta-analysis of 171 peer-reviewed studies and a re-analysis of >8,000 assessments from >3,000 global protected areas”. The highly unbalanced nature of such a dataset will yield biased results unless the consequent extreme variation in observation error across these topics is incorporated into the analysis (see de Leeuw, J. and Meijer, E. (2008) Handbook of Multilevel Analysis). It is not obvious from the paper that this has been incorporated into the analysis.

In response to earlier review comments about the sources of evidence being unclear, we had added in specifics on sample size in some of the studies we reviewed. However, we recognize now that the way this was worded was misleading and left the impression that we in fact had conducted the meta-analysis on 171 studies and analysis of >8000 assessments. In fact, we were citing two studies that had compiled that substantial evidence base. We have now reworded the explanation of this evidence to make clear that we are citing other studies that have documented the protected area outcomes we are relating. We have revised the following sentences in section 3: “A review of impact evaluation studies on protected areas and an in-depth qualitative examination of case studies from around the world show that when local values like stewardship are integrated, decision-making delivers more just and sustainable outcomes, especially when these values have been traditionally marginalised^{12,46}. Studies have established that community involvement improves management effectiveness (based on an analysis of >8,000 assessments from >3,000 global protected areas⁵⁰), and that local empowerment and recognition of local values, especially for Indigenous communities, enhances win-wins between ecological and social outcomes of protected areas (demonstrated in a meta-analysis of 171 peer-reviewed studies⁵¹ and a systematic review of 169

publications⁵², as well as in-depth case studies of our own review (SI B: #16). Hence, the sample sizes refer to the references we are citing in the text and not to one of our 29 search strategies.

Line 605-613

Aside from the fact that it precedes Figure 4, I have some reservations regarding Figure 5 (“The diverse values of nature underpin multiple pathways towards sustainability”).

First it places the concept of intrinsic value as central to this diagram – please see comments on lines 353 – 356. Second the ‘degrowth’ concept emphasised here is very difficult to defend as a viable future pathway for real world policy. Even its supporters, such as Kallis (2011), Kallis et al., (2011) and van den Bergh and Kallis (2012) and D’Alisa and Kallis (2020), recognise that simple degrowth on a planet of 8 billion people would be catastrophic and argue instead for socially and ecologically sustainable degrowth.

References:

D’Alisa, G. and Kallis, G. (2020) Degrowth and the State, Ecological Economics, 169, 106486, <https://doi.org/10.1016/j.ecolecon.2019.106486>.

van den Bergh, J.C.J.M. and Kallis, G. (2012) Growth, A-Growth or Degrowth to Stay within Planetary Boundaries?, Journal of Economic Issues, 46:4, 909-920, DOI: 10.2753/JEI00213624460404

Kallis, G. (2011) In defence of degrowth, Ecological Economics, 70(5): 873-880, <https://doi.org/10.1016/j.ecolecon.2010.12.007>.

Kallis, G., Kerschner, C. and Martinez-Alier, J. (2012) The economics of degrowth, Ecological Economics, 84: 172-180, <https://doi.org/10.1016/j.ecolecon.2012.08.017>.

Thank you for this clear suggestion - We have adjusted the text in line with the reviewer's comment to reflect the pathway intention and focus on “socially and ecologically sustainable degrowth” (this is also now reflected in the caption of figure 5). We have also clarified that “‘social and ecological degrowth’ focuses on reducing overconsumption and overproduction, and redistributing wealth” which is in line with the review about degrowth literature in chapter 5 of the IPBES Values Assessment (Martin et al., 2022), where it is stated that degrowth references a long-standing concern with ecological limits to growth and subsequently planetary boundaries. Hence, the literature on degrowth does not promote it as a blanket reduction in production and consumption but as a redistribution of consumption so that the consumption needs and wellbeing of all can be met. According to degrowth scholars this would require a reduction in consumption among the wealthy due to insufficient progress delinking production from environmental impact. The paper cites this as one narrative of a more sustainable future but makes no judgement about whether this is the best pathway. We do not see any justification for not recognising the existence of this important strand of environmentalism. We have also added a new reference to a recent paper in Nature by Hickel et al (2022) that further supports this adjustment.

Hickel, J., Kallis, G., Jackson, T., O’Neill, D. W., Schor, J. B., Steinberger, J. K., ... & Ürge-Vorsatz, D. (2022). Degrowth can work—here’s how science can help. *Nature*, 612(7940), 400-403.

We have decided to keep the concept of intrinsic values in the figure as this is an important part of the assessment on sustainability pathways (see also our response to the comment by the reviewer on intrinsic values above).

Line 619-620

The ms. states: “conservation policies have frequently prioritised nature’s intrinsic values”

No, they have not. Conservation policies prioritise human values for nature. including human desires to conserve certain aspects of nature. For centuries people have waged war on those wild species and aspects

of nature that humans do not like. No one seems to care about the ‘intrinsic value’ of the common pigeon. Why? Because (i) there is no way of knowing what that intrinsic value is and (ii) humans have decided that individuals from Species A (e.g. ospreys) are more important, more valuable, than individuals from Species B (pigeons). Those are human values and I very much doubt those individuals from Species B would agree with this prioritisation. The prioritisation of a policy according to something that is by its very definition unknown and unknowable to humans is impossible. The authors could provide a very considerable service to the literature by clearly standing out against this nonsense. Please see comments on lines 353 – 356.

Please see our previous response to the general point made by the reviewer on intrinsic values, particularly the way we have reframed the notion of intrinsic value. As part of the extensive review we assessed how people, and in this case conservation biologists and conservation practitioners (including NGOs), express intrinsic values of biodiversity as their rationale for protecting nature. Our assessment echoes this literature which points towards the expression of intrinsic values in this regard and debates around it reflected in works such as those by McCauley (2006), Sandler (2012), Reyers et al (2012), Batavia and Nelson (2017), McCauley (2006), and Pascual et al (2021), as well as noting that this is endorsed in the preamble of the Convention on Biological Diversity (UN, 1992). Please note that we are not implying that we agree or disagree with the rationale of expressing intrinsic values to support biodiversity conservation.

Batavia, C., & Nelson, M. P. (2017). For goodness sake! What is intrinsic value and why should we care?. *Biological Conservation*, 209, 366-376.

McCauley, D. J., 2006, "Selling out on nature", *Nature*, 443(7107): 27.

Reyers, B., S. Polasky, H. Tallis, H. A. Mooney and A. Larigauderie, 2012, "Finding Common Ground for Biodiversity and Ecosystem Services", *BioScience*, 62(5): 503-507.

Sandler, R. (2012) Intrinsic Value, Ecology, and Conservation. *Nature Education Knowledge* 3(10):4
<https://www.nature.com/scitable/knowledge/library/intrinsic-value-ecology-and-conservation-25815400>

Pascual, U., Adams, W.A., Diaz, S., Lele, S., Mace, G., Turnhout, E. (2021). Biodiversity and the challenge of pluralism. *Nature Sustainability*. 4: 567-572/

United Nations, 1992. ‘Convention on Biological Diversity’, (05/06/1992).
<https://www.cbd.int/doc/legal/cbd-en.pdf>

Line 615-709

Section 5 “Transformative change involves leveraging nature’s values” is interesting but I feel unbalanced.

An initial, minor, writing point is that the text needs to mirror the cumulative addition of levers shown in the Figure as the move from Shallower to Deeper levers.

A more important point is the apparent assertion that those levers designated as ‘Deeper’ will indeed be more effective than those labelled as ‘Shallower’. While this almost has to be true if levers are indeed added cumulatively (see above), this does not have to be the case if they are independent alternatives. So one could have policy change independent of a social norm change. In such cases is it necessarily the case that the former must generate less impact than the latter? I’m not sure and would need to see more extensive evidence regarding this.

A further, more substantial point is that the Figure and text strongly suggest that the movement from Shallower to Deeper levers is always desirable. However, this ignores the feasibility and costs associated with such movement. It might be that changing policy is relatively costless but generates major improvement but changing social norms might be challenging and costly. In such a case the intuition of the

diagram breaks down and becomes misleading.

I think the authors need to rethink this section. Given that Nature Figures need to work as stand alone items, and often get reproduced separate from their wider papers I feel that merely revising the text is unlikely to be sufficient here.

Thank you. The figure on (values-centred) leverage points can be interpreted as leverage points being interdependent (e.g. being nested, etc). We have now made this explicit in the caption: “*Transformative change is more likely when interventions engage multiple leverage points. The leverage points are interdependent, whereby jointly activating them entails addressing feedbacks among them, adding them up (moving left to right across the lever) or cascading down (moving right to left across the lever)*”⁷. We have also added an additional paragraph at the end of the section to explain that it does not necessarily need to be read either as independent leverage points, but could be done from left to right (cumulatively) and from right to left (i.e. cascading down the lever), i.e. a change in the deeper leverage point can unleash the rest of the leverage points. The paragraph at the end of the section includes an example of the agri-environmental policy framework in Europe (to ensure global coverage of our case study and illustrative examples). We have thought of remaking the figure, but we have realised that it would become overly crowded with too many arrows, etc. and thus we prefer keeping the original figure with the updated caption.

Line 827

In my previous review I stated the link to the following reference leads to a blank document and that this was an important element of the review as it presumably provides details of the methodology.

16. IPBES. Methodological assessment of the diverse values and valuation of nature of the Intergovernmental Science-Policy Platform on Biodiversity and Ecosystem Services. P. Balvanera, U. Pascual, M. Christie, B. Baptiste, D. González-Jiménez (eds.). IPBES secretariat, Bonn, Germany (2022). <https://doi.org/10.5281/zenodo.6522522>

In response the authors said: “The full report will appear in this link once its editing is finalised. In the meantime, we have requested the IPBES Secretariat add the links to all the chapters and Summary for Policy Makers within this web entry.” This means that I am still unable to review this document.

Sorry. The report is now accessible following the provided link.

Referee #6

The responses to the broad and specific points raised in my review have mostly been acceptable. However, I was disappointed by the authors' response to my first overarching point (that no empirical evidence on the relative prevalence of values for nature held by people has been presented) and indeed, do not think the way they have dealt with it is acceptable. This could easily be remedied by including some or all of the text in their response to my comment into the actual manuscript (note the authors claim Haerpfer et al 2022 is now cited but i couldn't see that this was the case, the citation appears missing).

*The authors say "we had difficulty finding a place to integrate arguments on this increasing pool of environmental protection values in post-industrial countries", but this would fit very obviously in section 1 where the paper states that "the values of nature are diverse". A key measure of diversity is not only the *number* of categories/things, but the *relative abundance* of each of those categories/things. As it stands the authors have documented the first aspect of diversity in values for nature, but not the second. As such, it remains impossible for this paper as written to refute the alternative hypothesis raised in my review, which is that "the vast majority of humanity holds anthropocentric, instrumentalist values for nature. And that the other worldviews/values are held by a tiny minority of people, meaning there isn't a great, untapped pool of pro-environmental values just waiting to be unleashed." Failing to address this weakens the paper and will leave it open to criticism that the authors are advocating for the existence of a*

diversity in values for nature for ideological reasons, rather than demonstrating empirically that this diversity in fact exists. This shouldn't be a particularly difficult area of the paper to improve given that the authors state in their response to me that such empirical evidence exists. Surely the authors can do better than simply saying we couldn't find a place to address this comment?

Thank you for constructive comments and for highlighting the comment on prevalence on differ types of values around the world. While the IPBES *Values Assessment* was not tasked to conduct a review to quantify such prevalences (i.e. 'status and trends in values'), we think that this is an important point to be explicitly mentioned in the paper. We have thus included in section 1 the following new paragraph to support the point of the prevalence of non-instrumental values, with a selection of references.

“This plurality of values is culturally expressed around the world, including in post-industrial societies with high levels of material security (Haerpfer et al., 2022). For example, in the U.S. endorsement of relational and intrinsic values (e.g. seeing wildlife as part of one’s social community and deserving of rights) is increasing (Manfredo et al., 2021). In the Global South where lower levels of livelihood security favour instrumental values (Haerpfer et al., 2022), expressions of non-instrumental values associated with e.g. spirituality and cultural identity are prevalent (Cocks et al 2012; Roux et al 2022). Indigenous and local knowledge is also embodied in different philosophies of good living underpinned by relational values as the basis for collective human-nature well-being (Anderson et al 2023), including through concepts like *Buen vivir* in South America (Albó, 2018) and *Ubuntu* in sub-Saharan Africa (Chibvongodze, 2016), among others.”

Albó, X. (2018). Suma Qamaña or Living Well Together: A Contribution to Biocultural Conservation. In R. Rozzi, R. H. May, F. S. Chapin III, F. Massardo, M. C. Gavin, I. J. Klaver, A. Pauchard, M. A. Nuñez, & D. Simberloff (Eds.), *From Biocultural Homogenization to Biocultural Conservation* (Vol. 3, pp. 333-342). Springer International Publishing. https://doi.org/10.1007/978-3-319-99513-7_21

Chibvongodze, D. T. (2016). Ubuntu is not only about the human! An analysis of the role of African philosophy and ethics in environment management. *Journal of Human Ecology*, 53(2), 157-166. <https://doi.org/10.1080/09709274.2016.11906968>

Cocks, M. L., Dold, T., & Vetter, S. (2012). 'God is my forest'-Xhosa cultural values provide untapped opportunities for conservation. *South African Journal of Science*, 108(5), 1-8.)

Haerpfer, C., et al. (eds.). 2022. *World Values Survey: Round Seven - Country-Pooled Datafile Version 5.0*. Madrid, Spain & Vienna, Austria: JD Systems Institute & WWSA Secretariat. doi:10.14281/18241.20

Manfredo, M. J., Teel, T. L., Berl, R. E., Bruskotter, J. T., & Kitayama, S. (2021). Social value shift in favour of biodiversity conservation in the United States. *Nature Sustainability*, 4(4), 323-330.

Roux, J.L., Konczal, A., Bernasconi, A., Bhagwat, S., De Vreese, R., Doimo, I., Marini Govigli, V., Kašpar, J., Kohsaka, R., Pettenella, D. and Plieninger, T., 2022. Exploring evolving spiritual values of forests in Europe and Asia: a transition hypothesis toward re-spiritualizing forests. *Ecology and Society*, 27(4).

Reviewer Reports on the Second Revision:

Referees' comments:

Referee #1 (Remarks to the Author):

Review of: Re-revised Nature manuscript 2022-07-11504C

Ref: 430534_3

Diverse values of nature underpin just and sustainable futures

Line No. Comment

All I appreciated the clarity of changes both in the Ms. and rebuttal – thank you

344 I suggest deleting the phrase “a similar concept is sometimes referred to as cosmocentric).” You don’t use the term elsewhere and it reduces the profusion of labels. If you are really keen to retain this somewhere in the submission I suggest you move it to the SI

351-352 The definition of intrinsic value is now phrased as:

“intrinsic (i.e. value of nature considered and expressed by people as an end-in-itself without reference to their own well-being)”

While this does now acknowledge that this is a human value, when people state such values they are demonstrably not acting “without reference to their own well-being”. If I say that I believe the Amazon should be reserved for its native wild species and indigenous peoples then I am expressing my personal values – I prefer that state of affairs and I experience greater happiness and well-being if that outcome arises.

This is not a minor point. The label of “intrinsic value” is applied as an ace-card in arguments to negate opposing views. We have to acknowledge that these are indeed well-being based values.

I suggest phrasing the sentence as:

“It is well established that nature’s specific values can be instrumental (i.e. related to the direct use of nature for some desired human end)¹⁵ or intrinsic (i.e. value of nature considered and expressed by people as delivering well-being quite separate from the valuers’ use of nature)¹⁶.

368 Please rewrite

“This values plurality”

As

“This plurality of values”

385-387 The ms. states:

“a multiple-values perspective that balances these with people’s important connections to the wetland (e.g. relational values connected to fisheries).”

I presume the latter refers to people's use of fisheries as a source of food and income. This seems to contrast with the previous definition of relational values given in lines 352-354 as:

"The relational category captures how people express the importance of meaningful relationships between people and nature and among people through nature (e.g. reciprocity, care)¹⁷."

Could this be clarified please.

388-389 "monetary measures (e.g. income growth and distribution)"

The common confusion between accounting measures (which relate solely to financial transactions) and economic measures (which relate to welfare changes typically translated into monetary amounts with recognition of the difficulties of with translation) could very usefully be addressed in this paper. Most Nature readers are non-economists and this will be helpful in fostering trans-disciplinary understanding.

I suggest you separate 'financial' measures from 'economic' measures or, if you want to be more forthcoming you could separate 'accounting measures of financial transactions' from 'economic measures of welfare change'.

435-436 A minor suggestion: you could change

"Valuation generates information about nature's values that can be used to make values visible to decision-makers,"

To:

"Valuation generates information that can be used to make nature's values visible to decision-makers,"

Or even

"Valuation makes nature's values visible to decision-makers,"

445-446 The ms. states:

"1) nature-based valuation gathers information about the importance of nature and NCP through direct and indirect observation of nature (e.g. spatial ecosystem services mapping)³¹"

Two points here:

First, could a less controversial label be found for such assessments. I can see it appearing rather churlish to criticise "nature-based" valuations and such assessments would appear to have a misleading superiority over other measures.

Second, I don't really feel that mapping can be seen as a form of valuation – surely mapping is a precursor to valuation.

489-490 The ms. states:

"Valuations have mainly been performed at sub-national scales, rather than national and global scales."

This of course reflects the scale of most decision making.

513-525 The paragraph on "Key barriers to valuation uptake in public decision-making" fails to mention what is arguably the most important issue of private ownership. Public decision makers can only have complete control over those resources directly owned by the state. This is (by a very long margin) the minority of global resources be the renewable, non-renewable, land, water or other resources. The bulk of resources are in private ownership and here the public decision maker cannot simply value and implement change but instead must fall back on a set of measures for delivering change ranging from incentives to regulation. This needs to be clarified and the relationship between valuation and those policy instruments discussed.

586-613 My own experience of scenario/pathway analyses is that they are rarely underpinned by empirical, quantified assessment of the trade-offs which alternative futures entail. I feel this is future for scenario/pathway analyses and the authors may wish to support such a view.

616-625 If you adopt the changes suggested above regarding introduction of private property challenges to the section on "Key barriers to valuation uptake in public decision-making" then this would help further strengthen this section as you could refer back to the former.

678-681 The ms. states:

"Similarly, reforming macroeconomic indicators (e.g. Gross Domestic Product) to include values that encompass social and ecological well-being could change both the design and intent of the economic system⁴"

I don't think reform of GDP is the right approach here. GDP is a very specific measure of financial flows in an economy. This is useful information and there is nothing wrong with that measure if it is used as its creators advocated. As they made clear it is not a measure of wellbeing and should not be used as such. The fact that it is so misinterpreted is not the fault of the measure. As advocated by many commentators for many years now (e.g. the Dasgupta report), what is needed are a set of ancillary accounts estimating changes in wellbeing (including distribution) and sustainability. The Gross Ecosystem Product measure advocated by the UN is an obvious and relatively tractable approach to the latter requirement.

729-743 I thought that the first half of the Concluding section, with its focus on transferring property rights to nature was a rather unbalanced drawing out of the evidence presented prior to that point.

Referee #6 (Remarks to the Author):

Thank you for addressing my remaining comment in a substantive manner.

I believe the new paragraph mostly addresses my concern. The examples from the US and Global South provide evidence of the amount/abundance of these different value types (i.e., they are "increasing" or "prevalent"), as I'd suggested was necessary. The South America and SSA examples do not however, so I would further suggest that you characterize the prevalence of Buen vivir and Ubuntu, relative to utilitarian values, in the same way as the previous two examples.

Author Rebuttals to Second Revision:

Comments and responses (in green) to the Editor and Reviewer 1.

Editor	Comment	Response
E1	For publication in Nature, please could I ask that you modify your definition of 'intrinsic value' to take into account the comments of referee 1 on this point? This seems quite critical.	Thank you. We have modified it to capture a broader interpretation that would be acceptable for different academic traditions. For example, philosophers and economists, who tend to be in somewhat opposing camps regarding how the concept of intrinsic values ought to be interpreted (within variations under each field, as already clarified in our previous response letters to reviewer 1). The newly revised text now reads as: For example, while many philosophers interpret intrinsic value in ways that do not relate to the valuer’s well-being, economists tend to view intrinsic values as partly connected to a person’s well-being, but separate from their own use (i.e., a non-use value). We believe this clarifies to a wide readership how intrinsic values are understood (differently) across different disciplinary traditions (e.g. philosophy and economics)
E2	Also, we’ve discussed this a bit editorially, and we think it would be best to take out the word 'just' from the title and the abstract, just because this term is hard to define, and can have different connotations in different fields. Actually though, we would need you to change the title anyway, to something that better depicts what the paper shows. We suggest 'A typology of the multiple conceptualizations of the values of nature'.	We do not find that within the recent discourse regarding nature conservation and sustainability that the word “just” is somehow a challenge to interpret. Indeed, it is part of SDG 16 and Targets 18 and 22 in the Kunming-Montreal Global Biodiversity Framework. Furthermore, it has been used linked between just/justice and sustainability in various recent Nature family titles: https://www.nature.com/search?q=justice&journal= While we would very much like to keep the word “just”, which refers to justice in the title, we accept the Editors’ decision that the concept may be misinterpreted, therefore, we suggest any of the following three options:  1. Diverse values of nature for sustainability 2. Diverse values of nature underpin sustainability 3. Diverse values of nature underpin sustainable futures
E3	We also think it would be good to define the term 'just' on first use, presumably in line with the UNESCO definition (and maybe to acknowledge some of the difficulties in assessing whether an outcome is just or not). What would also be good is if you are slightly more explicit when referring to previous literature in this respect. That is, you could say whether you are using your definition of just (in this paper) or that which was used in the studies you cite and base your analysis on.	Thank you. We use the concept of justice in line with the Values Assessment (e.g. the Summary for Policymakers states that: “Justice is a broad value connected to the principle of fairness, i.e., the fair treatment of people and other-than-human nature, including inter- and intra-generational equity”. See end of the bold paragraph. We further explain that “achieving justice implies considering its various dimensions, including: (i) recognition justice, acknowledging and respecting different worldviews, knowledge systems and values; (ii) procedural justice, making decisions that are legitimate and inclusive for those holding different values; and (iii) distributional justice, ensuring the fair distribution of nature’s contributions to people”. See fourth paragraph of section 2.
E4	Also, please bear in mind we do not have a bold paragraph and abstract, as you have here, but	We have merged the two into a new bold paragraph.

	simply one opening bold paragraph. So please could you combine these two paragraphs as your bold opening paragraph? It would need to fit within our length limits for these paragraphs of course, but any surplus text can go in the introduction.	
--	---	--

Revi- ewer 1	Comment	Response
R1	All I appreciated the clarity of changes both in the Ms. and rebuttal – thank you	Thank you for your detailed review of the paper. We believe all the comments have helped to improve the paper.
R2	344. I suggest deleting the phrase “a similar concept is sometimes referred to as cosmocentric).” You don’t use the term elsewhere and it reduces the profusion of labels. If you are really keen to retain this somewhere in the submission I suggest you move it to the SI.	Done.
R3	351-352. The definition of intrinsic value is now phrased as: “intrinsic (i.e. value of nature considered and expressed by people as an end-in-itself without reference to their own well-being)” While this does now acknowledge that this is a human value, when people state such values they are demonstrably not acting “without reference to their own well-being”. If I say that I believe the Amazon should be reserved for its native wild species and indigenous peoples then I am expressing my personal values – I prefer that state of affairs and I experience greater happiness and well-being if that outcome arises. This is not a minor point. The label of “intrinsic value” is applied as an acecard in arguments to negate opposing views. We have to acknowledge that these are indeed well-being based values. I suggest phrasing the sentence as: “It is well established that nature’s specific values can be instrumental (i.e. related to the direct use of nature for some desired human end)¹⁵ or intrinsic (i.e. value of nature considered and expressed by people as delivering wellbeing quite separate from the valuers’ use of nature)¹⁶	Thank you. We have now included the following: For example, while many philosophers interpret intrinsic value in ways that do not relate to the valuer’s well-being, economists tend to view intrinsic values as partly connected to a person’s well-being, but separate from their own use (i.e., a non-use value).
R4	368. Please rewrite “This values plurality” As “This plurality of values”	Done.
R5	385-387. The ms. states: “a multiple-values perspective that balances these with people’s important connections to the wetland (e.g. relational values connected to fisheries).”I presume the latter refers to people’s use of fisheries as a source of food and income. This seems to contrast with the previous definition of relational values given in lines 352-354 as: “The relational category captures how people express the importance of meaningful	We use this example to clarify that the relational values of the fisheries are connected to fishers’ livelihoods, such as fishers’ cultural identify (i.e., fishing beyond a means to material well-being). The text now reads: “... important connections to the wetland (e.g., relational values connected to the cultural identify of being fishers)”.

	relationships between people and nature and among people through nature (e.g. reciprocity, care) ¹⁷ .” Could this be clarified please.	
R6	388-389. “monetary measures (e.g. income growth and distribution)”. The common confusion between accounting measures (which relate solely to financial transactions) and economic measures (which relate to welfare changes typically translated into monetary amounts with recognition of the difficulties of with translation) could very usefully be addressed in this paper. Most Nature readers are non-economists and this will be helpful in fostering transdisciplinary understanding. I suggest you separate ‘financial’ measures from ‘economic’ measures or, if you want to be more forthcoming you could separate ‘accounting measures of financial transactions’ from ‘economic measures of welfare change’.	We agree. Given space constraints, we have simply changed it to “economic and financial aspects through monetary measures”.
R7	435-436. A minor suggestion: you could change “Valuation generates information about nature’s values that can be used to make values visible to decision-makers,” To: “Valuation generates information that can be used to make nature’s values visible to decision-makers,” Or even “Valuation makes nature’s values visible to decision-makers,”	Done. We have changed to “Valuation generates information that can be used to make nature’s values more visible to decision-makers”
R8	445-446. The ms. states: “(1) nature-based valuation gathers information about the importance of nature and NCP through direct and indirect observation of nature (e.g. spatial ecosystem services mapping) ³¹ ” Two points here: First, could a less controversial label be found for such assessments. I can see it appearing rather churlish to criticise “nature-based” valuations and such assessments would appear to have a misleading superiority over other measures. Second, I don’t really feel that mapping can be seen as a form of valuation – surely mapping is a precursor to valuation.	We have chosen the term nature-based valuation to distinguish this type of valuation from, for example, preference-based valuation. Given that valuation is defined as stated above “Valuation generates information that can be used to make nature’s values visible to decision-makers”, this makes ecosystem service mapping and biodiversity hotspot mapping “valuation”. Furthermore, such valuation is used to communicate nature’s values to decision makers. We aim to characterise the different forms of valuation and argue that they entail different information to decision-makers. We have not been able to come up with a better term and would prefer to leave it as it is.
R9	489-490. The ms. states: “Valuations have mainly been performed at sub-national scales, rather than national and global scales.” This of course reflects the scale of most decision making.	True. Point now included in the ms.
R10	513-525. The paragraph on “Key barriers to valuation uptake in public decision-making” fails to mention what is arguably the most important issue of private ownership. Public decision makers can only have complete control over those resources directly owned by the state. This is (by a very long margin) the minority of global resources be the renewable, non-renewable, land, water or other resources. The bulk of resources are in private ownership and here the public decision maker cannot	Thank you. We agree that much of decision-making happens outside the public sphere. We have, thus, deleted the word “public” from the sentence. We prefer to keep the rest of the paragraph as it is given the space constraints.

	simply value and implement change but instead must fall back on a set of measures for delivering change ranging from incentives to regulation. This needs to be clarified and the relationship between valuation and those policy instruments discussed.	
R11	586-613. My own experience of scenario/pathway analyses is that they are rarely underpinned by empirical, quantified assessment of the trade-offs which alternative futures entail. I feel this is future for scenario/pathway analyses and the authors may wish to support such a view.	Good point. We have added the following to the end of the first paragraph of section 4: "... although generally scenarios are not co-developed by accounting for value trade-offs".
R12	616-625. If you adopt the changes suggested above regarding introduction of private property challenges to the section on "Key barriers to valuation uptake in public decision-making" then this would help further strengthen this section as you could refer back to the former.	No action taken here, as we have not gone into the discussion about private property. This would have entailed even more text, which we cannot afford given the word limit and also being true to the scope of this manuscript.
R13	678-681. The ms. states: "Similarly, reforming macroeconomic indicators (e.g. Gross Domestic Product) to include values that encompass social and ecological well-being could change both the design and intent of the economic system⁴" I don't think reform of GDP is the right approach here. GDP is a very specific measure of financial flows in an economy. This is useful information and there is nothing wrong with that measure if it is used as its creators advocated. As they made clear it is not a measure of wellbeing and should not be used as such. The fact that it is so misinterpreted is not the fault of the measure. As advocated by many commentators for many years now (e.g. the Dasgupta report), what is needed are a set of ancillary accounts estimating changes in wellbeing (including distribution) and sustainability. The Gross Ecosystem Product measure advocated by the UN is an obvious and relatively tractable approach to the latter requirement	We agree that the important point here is that the GDP is a poor measure of well-being, which is widely acknowledged. We have added "complementing" to reflect the widely argued point that multiple indicators are needed to measure well-being and sustainability. The sentence now reads "Similarly, reforming and complementing macroeconomic indicators (e.g., Gross Domestic Product) to include values that encompass social, economic and ecological well-being could change both the design and intent of the economic system⁴"
R14	729-743. I thought that the first half of the Concluding section, with its focus on transferring property rights to nature was a rather unbalanced drawing out of the evidence presented prior to that point.	Thank you. We have changed the expression "allocation of property rights to nature" to "over nature" as the former "to" could incorrectly conveys the idea that the conclusion of the ms should be geared towards the idea of transferring property rights to nature. This is part of the bigger picture, but the issue of property rights is bigger than this.